# The paradoxical evolution of runoff in pastoral Sahel: Analysis of the hydrological changes over the Agoufou watershed (Mali) using the KINEROS-2 model.

Laetitia Gal[1], Manuela Grippa[1], Pierre Hiernaux[1], Léa Pons[1] and Laurent Kergoat[1]

[1]Geosciences Environnement Toulouse, Toulouse, France

*Correspondence to*: L. Gal (gal.laetitia@gmail.com)

**Abstract**

In the last decades, the Sahel has witnessed a paradoxical increase in surface water despite a general precipitation decline. This phenomenon, commonly referred to as "the Sahelian paradox", is not completely understood yet. The role of cropland expansion due to the increasing food demand by a growing population has been often put forward to explain this situation for the cultivated Sahel. However, this hypothesis does not hold in pastoral areas where the same phenomenon is observed. Several other processes, such as the degradation of natural vegetation following the major droughts of the 70ies and the 80ies, the development of crusted top soils, the intensification of the rainfall regime and the development of the drainage network, have been suggested to account for this situation.

In this paper, a modeling approach is proposed to explore, quantify and rank different processes that could be at play in pastoral Sahel. The KINEmatic runoff and EROSion model (KINEROS-2) is applied to the Agoufou watershed (245 km²), in the Gourma region in Mali, which underwent a significant increase of surface runoff during the last 60 years. Two periods are simulated, the "past" case (1960−1975) preceding the Sahelian drought and the "present" case (2000−2015). Surface hydrology and land cover characteristics for these two periods are derived by the analysis of aerial photographs, available in 1956, and high resolution remote sensing images in 2011.The major changes identified are: 1) a partial crusting of isolated dunes, 2) an increase of drainage network density, 3) a marked decrease in vegetation with the non-recovery of tiger bush and vegetation growing on shallow sandy soils and 4) important changes in soil properties with the apparition of impervious soils instead of shallow sandy soil. The KINEROS-2 model was parameterized to simulate these changes in combination or independently. The results obtained by this model display a significant increase of annual discharge between the "past" and the "present" case (p value $< 0.001$), which is consistent with observations, despite a slight overestimation of the past discharge. Mean annual discharges are estimated at $0.51\times10^{6}$ m$^3$ (2.1 mm yr$^{-1}$) and $3.29\times10^{6}$ m$^3$ (13.4 mm yr$^{-1}$) for past and present respectively.

Changes in soil properties and vegetation cover (tiger bush thickets and grassland on shallow sandy soil) are found to be the main factors causing this increase of simulated runoff, with the drainage network development contributing to a lesser extent but with a positive feedback. These results shed a new light on the Sahelian paradox phenomenon in the absence of land use change, and call for further tests in other areas and/or with other models. The synergetic processes highlighted here could play a role in other Sahelian watersheds where runoff increase has been also observed.

**Keywords:** Sahelian paradox, Annual discharge, KINEROS-2, Agoufou watershed

## 1 Introduction

During the second half of the 20th century, the Sahel underwent a severe rainfall deficit, considered as the largest multi-decadal drought of the last century (Hulme, 2001; Nicholson et al., 1998), with extreme droughts in 1972-73 and again in 1983-84, that strongly impacted ecosystems, water availability, fodder resources, and populations living in these areas (Nicholson, 2005).

Responses induced by this deficit result in contrasted effects depending on the ecoclimatic zone considered. If the Sudano-Guinean zone displayed an expected decrease of surface runoff following the drought, the opposite situation was observed in the Sahelian zone (Descroix et al., 2009; Séguis et al., 2002, 2011). First reported for a small watersheds in Burkina Faso by Albergel (1987), this paradoxical situation was also diagnosed by Mahé and Olivry (1999) for several other watersheds in West African Sahel, then by Mahé et al. (2003) for the right bank tributaries of the Niger river and by Mahé et al. (2010) for the Nakambé watershed. This phenomenon was also observed as West as Mauritania (Mahé and Paturel, 2009) and as East as Nigeria (Mahé et al., 2011) and as north as in the Gourma region (Gardelle et al. 2010). This regional phenomenon is commonly referred to as "the Sahelian paradox" and its causes are still debated.

Whether this situation is man-made or mostly a response to climate variability is of great importance for planning and management of water resources and development. The leading role of increased cropped surface and land clearance has been put forward in several studies carried out in cultivated Sahel (Favreau et al., 2009; Leblanc et al., 2008; Mahé and Paturel, 2009). Population growth in the Sahel is rapid and associated with important Land Use Changes (LUC) since the 50s.

However, the LUC hypothesis does not hold for pastoral areas commonly found in central and northern Sahel. In northern Mali for instance, an important area extension and flood duration of ponds and lakes has been observed (Gardelle et al., 2010), which has a large impact on local population and economy since the installation of people and livestock often depends on the presence of surface water. A similar evolution is suspected for other ponds and lakes in pastoral areas in Niger and Mauritania also (Gal et al., 2016). Changes in Land Cover (LCC), particularly in vegetation and soil properties, have been put forward as a possible explanation. Gardelle et al. (2010) suggested that the non-recovery of some ecosystems after the major droughts could be responsible for the significant increase in the surface of ponds in northern Mali. Vegetation degradation favors surface runoff via the acceleration of the overland flow and the reduction of soil hydraulic conductivity. In addition, a reduction in vegetation cover can contribute to decreasing rainfall interception and soil protection against raindrop energy, favoring the top soil crusting which again limits infiltration and trigger rainfall excess overland flow. The role of top soil crusting has been pointed out in several studies. Sighomnou et al. (2013) suggested that vegetation degradation and land clearance in southwestern Niger have changed soil surface properties and infiltration capacity enough to increase Hortonian runoff. A general decline in vegetation cover generating increased soil erosion and crusting and in turn an increase of surface runoff has been put forward by Leblanc et al. (2007), Hiernaux et al. (2009a), Toure et al. (2010) or Aich et al. (2015).

Another possible factor cited in the literature is the development of the drainage network. Leblanc et al. (2008) analyzed time series of aerial photographs in southwestern Niger and reported a spectacular increase in drainage density, as it was also found by Massuel (2005).

It should be noted that interactions and feedbacks among these different drivers are quite common in dry lands. For instance, the development of impervious surfaces may favor rapid runoff, possibly gully erosion, which in turn may deprive vegetation from water, resulting in vegetation decay and more imperviousness. Last, a change in daily rainfall regime could be a possible cause of increased runoff. A slight increase in large daily rainfall has been suggested by Frappart et al. (2009) and demonstrated by Panthou et al. (2012, 2014). This signal is mostly observed since the 2000, and does not imply a change in rainfall intensity measured at shorter time scale.

Although hydrological modeling is a valuable tool to investigate the mechanisms responsible for the Sahelian paradox, few modeling studies have been carried out so far, mainly addressing the impact of land use change and land-clearing on surface runoff (Aich et al., 2015; D'Orgeval and Polcher, 2008; Favreau et al., 2009; Li et al., 2007; Mahé et al., 2005; Mahé and Paturel, 2009; Séguis et al., 2004). This is partly due to difficulties of modeling hydrological processes in semi-arid regions, for instance in endorheic areas, but also to the limited historical data available to calibrate and validate hydrological models (see for example Li et al., 2007; Mahé et al., 2005). Furthermore, Grippa et al., (2016) analyzed the hydrological behavior of 20 different land surface models (LSMs) over the Agoufou watershed and showed their inability to correctly differentiate among shallow or silty soils, generating runoff, and deep sandy soils with high infiltration capability that dominate non-runoff areas. Attribution studies inferring the impact of the different factors detailed above on surface runoff are therefore lacking.

The objectives of this study are: 1) to analyze the soil, land cover and hydrological changes that occurred over the Agoufou watershed since the 50ies and 2) to investigate how these changes impact surface runoff. In that purpose, the KINEmatic runoff and EROSion model (KINEROS-2) is used to simulate runoff over the past (1960-1975) and the present (2000-2015) periods.

# 2 Materials

## 2.1 Study site

The Agoufou watershed (**Fig. 1**) is located in the Gourma, a region of northern Mali delimited by the Niger river to the North and the border with Burkina-Faso to the South.This region has been extensively monitored by the AMMA-CATCH observatory (Analyse Multidisciplinaire de la Mousson Africaine - Couplage de l'Atmosphère Tropicale et du Cycle Hydrologique) and before by ILCA (International Livestock Centre for Africa) and IER (Institut d'Economie Rurale in Mali) providing historical data (Hiernaux et al., 2009b; Lebel et al., 2009; Mougin et al., 2009).

As elsewhere in the Sahel, the climate is tropical semi-arid with a unimodal precipitation regime. The rainy season extends from late June to September, and is followed by a long dry season. Precipitation comes from tropical convective events, 25 to 50 per year, brought by the West-African monsoon (Frappart et al., 2009; Vischel and Lebel, 2007). Its long-term evolution has been characterized by a wet period between 1950 and 1970 followed by a long dry period with extreme droughts in 1972-73 and again in 1983-84. The last 15 years have shown a partial recovery of rainfall, with large events seemingly occurring more often (Frappart et al., 2009; Panthou et al., 2012). Average rainfall was 345 mm $yr^{-1}$ over the 2000−2015 period and 382 mm $yr^{-1}$ over the 1960−1975 period. Potential evaporation was much higher than precipitation with averages of 3 235 mm $yr^{-1}$ and 2 930 mm $yr^{-1}$ for the two periods respectively.

The Agoufou watershed extends over 245 km² and ranges between latitude of 15.3 °N and 15.4 °N and longitude of 1.4 °W and 1.6 °W. The Gourma region is endorheic, which means that it is a mosaic of closed drainage watersheds that does not provide outflow to the Niger river and thus to the Atlantic Ocean. The Agoufou lake is the outlet of the watershed. As the majority of lakes and ponds in the region, it showed an important surface area increase over the last 50-60 years (Gal et al., 2016) and nowadays, typically reaches about 3 km² at the end of the rainy season.

Geology is characterized by Upper Precambrian schists and sandstones partially covered by staggered ferricrete surfaces (Grimaud et al., 2014), silt depositions and sand dunes. The study site has been extensively described in Gal et al. (2016). The northern part of the watershed (**Fig. 1**) consists of outcrops and shallow soils lying on sandstone, schist or iron pans. Some of these soils are fine textured soils (silt flats) which favor the frequent generation of runoff. The southern part is dominated by deep sandy soils with high infiltration capacity. The altitude range is 92 m; the average slope of the main reach is equal to 0.22 %.

The vegetation is typical Sahelian vegetation with an herbaceous layer almost exclusively composed of annual plants, among which grasses dominate, plus scattered bushes, shrubs and low trees (Boudet, 1972; Hiernaux et al., 2009a, 2009b). Almost continuous on sandy soils, except for a few deflation patches and bare dune crests, the herbaceous layer is highly discontinuous on shallow soils and clay plains, leaving large bare areas prone to runoff. The density and crown cover of woody plants are low in average, usually between 0 and 5 % (Hiernaux et al., 2009b). Woody plants concentrate along drainage lines, around ponds, in the inter-dune depressions and also sometimes on shallow soils, with a regular pattern of narrow linear thickets set perpendicular to the slope known as “tiger bush” (Hiernaux and Gérard, 1999; Leprun, 1992). These thickets live on the water and nutrients harvested on the impluvium made by the bare soil upstream, and their development efficiently limit runoff further downstream (D’Herbès and Valentin, 1997).

Casenave and Valentin (1989), among others, have demonstrated that the Sahelian hydrological processes are largely dependent on land surface conditions: soil properties, crusting, topography and vegetation cover (Albergel, 1987; Collinet, 1988; Dunne et al., 1991; Hernandez et al., 2000). Low soil infiltrability associated with the convective nature of the precipitation favors runoff generation by infiltration excess (Descroix et al., 2009, 2012; Leblanc et al., 2008; Peugeot et al., 2003) commonly known as Hortonian runoff.

The choice of this watershed has been motivated by tree characteristics: 1) This site is instrumented by the SO-AMMA-CATCH, which provides field measurements on vegetation and soil characteristics, meteorological variables, and lake’s height estimates over a long period of time (starting in 1984 for the long term ecological survey), as well as good field knowledge by co-authors. 2) the Agoufou lake has experienced a spectacular increase in inflow over the past 60 years despite the decrease in precipitation (Gal et al. 2016), which is a very good example of the Sahelian paradox and of the evolution of surface water observed more generally in the Gourma region (see Gardelle et al., 2010) and elsewhere in the Sahel (Mauritania and Nigeer, Gal et al., 2016). 3) This site is a pastoral watershed where agricultural activity is almost non-existent. It is thus different from the watersheds that were addressed by hydrological studies on the Sahelian paradox up to now. It can therefore shed a new light on the debate over land use versus land cover as possible explanation of the Sahelian paradox.

## 2.2 KINEROS2

Gal (2016) carried out a literature review of 20 different hydrological models (global, distributed and semi-distributed) in order to identify the most appropriate models to simulate hydrological processes in the study area and to meet the objectives of this study.

The KINematic runoff and EROSion model (KINEROS-2) was considered the most suited among those analyzed. KINEROS-2 (K2; Goodrich et al., 2011; Semmens et al., 2008; Smith et al., 1995) is the second version of KINEROS (Woolhiser et al., 1990). It is an event-oriented physically based model describing the processes of infiltration, surface runoff, interception and erosion for small watersheds and most of its applications concern arid and semi-arid areas (Hernandez et al., 2005; Kepner et al., 2008; Lajili-Ghezal, 2004; Mansouri et al., 2001; Miller et al., 2002). The surface runoff simulation is based on the numerical solution of the kinematic wave equations (Wooding, 1966), solved with a finite difference method. It assumes that runoff can be generated by exceeding the infiltration capacity (Hortonian mechanism) or by soil saturation depending on rainfall intensity and soil properties (infiltration capacity). The infiltration process is based on the Smith and Parlange equation (1978) defined by soil and land cover parameters: soil water capacity (the difference between soil saturation capacity and initial saturation), saturated hydraulic conductivity, soil porosity, net capillary drive, pore distribution, roughness coefficient and percent of canopy cover. Evapotranspiration and groundwater flow are neglected (Mansouri et al., 2001) but K2 takes into account canopy interception and storage. Soil water is redistributed during storm intervals (Corradini et al., 2000) based on the Brooks and Corey relationship, corresponding to an unsaturated permanent flow.

The watershed is treated as a cascading network of planes and channel elements. Channels receive flow from adjacent planes and/or upslope channel. Each element is assigned homogeneous parameter values that describe geometry and hydrological parameters (slope, vegetation cover, soil properties, initial conditions etc.) and control runoff generation (Goodrich et al., 2011).

Element definition is done with the Automated Geospatial Watershed Assessment tool (AGWA) which is the GIS-based interface (Miller et al., 2007). From the topography, AGWA discretizes the watershed into sub-watersheds (or planes) according to the Contributing Source Area (CSA) defined by the user. The CSA is the minimum area that is required for initiation of channel flow. The number of sub-watershed (or planes) and the density of the channel network increase with decreasing CSA. Each plane is considered as homogenous and its hydrological parameters are derived from soil surface characteristics maps based on soil texture (FAO classes) and vegetation properties.

## 2.3 Input data

K2 needs four input datasets: the digital elevation model (DEM), the soil map and the land cover map that are necessary to describe the watershed in term of hydrological and geometric parameters and the precipitation data that are needed at a small time step (5 minutes) to take into account the short and intense rainfall events, typical of the Sahelian monsoon. The input data used in this study are summarized in Table 1 and described below. Further details and analysis of in-situ data can be found in Frappart et al. (2009) Guichard et al. (2009), Timouk et al. (2009) and Gal et al. (2016).

#### 2.3.1 Digital elevation model (DEM)

Two DEMs, with a horizontal spatial resolution of 30 meters, are commonly used in hydrological studies: the Advanced Spaceborne Thermal Emission and Reflection Radiometer (ASTER) DEM and the Shuttle Radar Topography Mission (SRTM) DEM. Studying two Ghana watersheds, Forkuor and Maathuis (2012) found that SRTM had a higher vertical accuracy than ASTER even if both DEMs provided similar geomorphologic structures. Moreover, ASTER was found to suffer from artifacts, mainly peaks, particularly in flat terrain, which proved difficult to remove through filtering (Isioye and Yang, 2013). For these reasons, SRTM was retained for this study, although the DEM derived by ASTER was not markedly different in our case.

#### 2.3.2 Soil and land cover images

For the present period, a high-resolution GeoEye-1 satellite image (0.42 m) acquired on February 17, 2011 is available through Google Earth. It is supplemented by a SPOT satellite image (resolution of 5 m) to cover the whole watershed (5 % of the watershed is not covered by GeoEye). For the past period, a series of aerial photography is available from IGN Mali (ND30 XXIII 1956). Seven stereo pairs of images acquired in 1956 cover the whole watershed.

#### 2.3.3 Precipitation and meteorological data

Two sets of precipitation data are used for the Agoufou watershed:

- Daily precipitation (DP) from the Hombori SYNOP meteorological station available from 1930 to 2012 through the Direction Nationale de la Météorologie du Mali and completed until 2015 by the AMMA-CATCH observatory. This station is 15 km away from the Agoufou lake.

- Rainfall at a temporal resolution of 5minutes (5M) obtained from an automatic raingauge network operated over 2006-2010 by the AMMA-CATCH Observatory in the Gourma region (Frappart et al., 2009; Mougin et al., 2009).

Raingauges used in this study (Table 1and **Fig. 1**) were selected for their proximity to the study site and for the quality of the measurements series (few gaps).

In addition, relative humidity, air temperature, incoming short-wave radiation and wind speed derived from the Agoufou automatic meteorological station at a time scale of 15 minutes, are used as input to the grass layer sub-model (see Sect. 3.2.4).

#### 2.3.4 Hydrological data

An indirect method developed by Gal et al. (2016) estimates the water inflow to the Agoufou lake which corresponds to the watershed outflow. This method uses a water balance equation that takes into account precipitation over the lake, infiltration, open water evaporation and changes in lake water storage. This last term is obtained by combining open water surface area, derived by high-resolution remote sensing data (Landsat, SPOT and Sentinel2) or in-situ height measurements, and a relationship between area and volume. Annual and intra-annual watershed outflows are available for 17 years between 1965 and 2015, depending on the availability of the satellite data. A sensitivity analysis has been carried out to evaluate this methodology (described in details

in Gal et al., 2016): it was found that errors on volume estimation and evaporation estimation are the most important and can both lead to under/overestimation of water outflow of about 10 %.

# 3 Methods

## 3.1 Landscape units

Land cover and soils maps have been derived from satellite data for the present period (2011) and from aerial photographs for the past period (1956). For each period, four major groups of landscape units have been distinguished: sandy soils units (S), outcrops units (O), pediment units or "glacis" (P) and flooded zones units (F) which are further divided into subunits with different soil and land cover types, and hydrological properties (Table 2). This classification is based on long term ecosystem survey (Mougin et al., 2009) and studies carried out in the Sahel by Casenave and Valentin (1989), Valentin et Janeau (1988), Kergoat et al. (2015) and Diallo and Gjessing (1999). Each landscape unit has unique vegetation and groups soil types with similar hydrodynamic properties.

For the past period, photo interpretation of stereo-pairs is used so that the relative elevation of the different units can be derived from the three dimensional view, which is helpful for identifying units on panchromatic images. For the present period, the very high resolution of satellite images and the true color composites allows discriminating each unit rather easily. For both periods, units have been delimited independently and manually to maximize consistency. When photo-interpretation is not sufficient to discriminate some landscape units, changes between present and past are considered null.

## 3.2 Model setup and watershed representation

The objective of this work is to use the K2 model to analyze changes between the past and present periods and to employ the model as a diagnostic tool. For this reason, model parameters are prescribed as realistically as possible for these two periods and calibration is kept at minimum in order not to mask out or distort the impact of observed landscape evolution on past and present surface runoff. The only calibrated parameters are those for channels since these are the least known from literature and they are difficult to identify with precision from remote sensing. Therefore, the change attribution study is carried out for planes only, which are not calibrated. Details on the model setup and on the determination of the different parameters are given in the next subsections.

### 3.2.1 Rainfall temporal disaggregation

Simulation of the Hortonian runoff associated with Sahelian convective rainfall requires precipitation data at a small time scale, typically of the order of a few minutes or tens of minutes. For the majority of the Sahel meteorological stations, historical rainfall data are available on a daily time step only, which makes temporal disaggregation necessary.

The temporal downscaling precipitation method applied in this study consists in replacing each daily precipitation (DP) event by an existing 5-minute (5M) series having the same daily amount. To that end, a Look-Up-Table (LUT) of all 5M events from all automatic raingauges was built. It comprises 612 events spanning 0-144 mm per day.

To document the dispersion caused by temporal disaggregation, for each DP event, ten 5M events of the LUT are retained to compose ten ensemble members. These ten 5M events are randomly chosen among all events within

3 mm of the DP event total. If less than ten 5M events exist in the LUT, the interval is widened to -5/+5 mm or to -10/+10 mm and if necessary, the closest value is retained. Most of the time, intervals are less than 3 mm wide (76 % of events). The 5-minute rates are rescaled so that the daily total amounts are the exactly same.

The temporal disaggregation of daily data creates variability in the 5M precipitation forcing caused by the difference between the 5M events from the LUT and the rainfall actually seen by the watershed. Therefore, analyses are carried out on the ensemble mean (annual means and 15-year average), which smoothed out this noise.

Before the temporal disaggregation, the rainfall time series are split into events delimited by at least two days without rain. Events are considered independent, implying that the soil recovers its initial moisture conditions at the beginning of each event. The time required to delimitate independent events has been determined using soil moisture data available via the AMMA-CATCH observatory (see De Rosnay et al., 2009).

We further assumed that the rainfall cells are large enough to be considered uniform over the entire watershed. The analysis of the 5-minute rainfall events densities for the whole stations available, Gal (2016), shows that the probability density functions are similar among the different stations close to the watershed (see **Fig. 1**), especially for the high intensities that are the major contributors to runoff. In addition, the cloud top temperature, derived by MSG remote sensing data, was also analyzed, confirming that the rainfall cells in this region are generally larger than the watershed area (see Gal, 2016 for more details). However, a small temporal time lag (<30min) can occur between the eastern and western parts of the watershed since the cells are propagating westward. Planes are small enough to be considered as homogeneous in terms of precipitation but such a time lag impacts the water level and infiltration in channels. However, this effect is compensated for during the simulation setup and the derivation of channel parameters (see below). Last, the spatial variability in Sahelian rain fields at the event scale, as observed by Le Barbe et al. (2002), tends to smoothed out at the annual time step even if it still persists and constitutes a source of uncertainty.

#### 3.2.2 Watershed complexity

As mentioned in Section 2.2, the CSA controls the level of geometric complexity in the discretization of the watershed and the density of the channel network (Thieken et al., 1999). Ideally, the complexity of the simulated watershed is consistent with the watershed soil heterogeneity as well as with the spatial resolution of the simulated processes (Canfield and Goodrich, 2006; Kalin et al., 2003; Lane et al., 1975). According to Helmlinger et al. (1993), the optimal CSA depends on the case study, but a value smaller than 2.5 % of the total watershed area is commonly selected. For this study, a CSA corresponding to 1 % of the total watershed was selected, so that the drainage network development corresponds to the drainage network common to 1956 and 2011, and a good compromise between simulation time, watershed complexity and homogeneity of the planes is reached (**Fig. 2**). The flow direction and the flow accumulation are then derived from the SRTM DEM, leading to 174 planes with a mean area of 1.4 km² and a “DEM-derived network”.

The same CSA have been retained for the present and the past cases, assuming that the broad features of the DEM did not change between the two periods (topography and the slopes of cascading planes). However, the drainage network has changed between these periods. To account for this, the DEM-derived network, which corresponds to the common network between 1956 and 2011, has been modified in some sub-watersheds to

match the network development observed by remote sensing in 2011. To that end, the aspect ratio of the planes in these sub-watersheds is adjusted to increase the channel length and keep the plane area constant, by multiplying plane width and dividing plane length by the same number. This number corresponds to the ratio of observed network length to DEM-derived network length for each sub-watershed. The alternative method of changing CSA to change the network between the past and the present cases was not retained, since it would also change planes size, location and properties in an uncontrolled way, which could complicate the interpretation of the results.

### 3.2.3 Derivation of soil characteristics

FAO codes used by AGWA are assigned to all landscape units defined in Table 2 to match as closely as possible the soil texture and depth, known from field survey and previous knowledge of the study region. Table 3 summarizes the hydrological parameters assigned to each landscape unit by AGWA (based on laboratory analysis from soil textures throughout the world and pedotransfer function) and, used in K2 simulations. The initial saturation, expressed as a fraction of the pore space, is estimated for each plane to be 20 % of the maximum soil saturation but no less than 0.001 $m^3$ $m^{-3}$ (the minimum required by K2).

### 3.2.4 Derivation of vegetation characteristics

Landscape units bear six different forms of vegetation: grassland and trees (GT), grassland (G), sparse trees (T), tiger bush tickets (TB), woody plant (W) and no vegetation (R), with different combination of herbaceous and woody plants (Table 4). Herbaceous plants are dominated by annual grasses and forbs, which grow rapidly during the rainy season and decay rapidly after the last rains. The Manning's roughness coefficient (Man) is particularly sensitive to vegetation cover (Table 4). The saturated hydraulic conductivity (Ks) is also increased when plants are present following K2 equations. Interception is considered negligible because of the nature of the precipitation (high intensity and high winds during convective storms) and because of the usually low values of Leaf Area Index (LAI) found at the study site (Carlyle-Moses, 2004; Mougin et al., 2014).

The seasonal dynamics of the grass canopy cover (CC) has been simulated with the STEP vegetation model (Mougin et al., 1995; Pierre et al., 2016). It is driven by historical daily precipitation recorded at the Hombori station and meteorological data (short-wave incoming radiation, air temperature, relative humidity and wind speed) recorded every 15 minutes. For the latter, a mean annual climatology is obtained using data from the Agoufou automatic weather station operating from 2002 and 2010. STEP being also dependent on soil texture and depth, it is run for deep soils and shallow soils (sand sheet 3 cm deep) separately to provide canopy cover over these different soils (CCd and CCs respectively). The relation between the Man and the percent of canopy cover (CC) is derived from several LUT, including NALC (North American Landscape Characterization) and MRLC (Multi-Resolution Land Characterization) provided in AGWA and reads as follows (Eq. (1)):

$$Man = 0.008 \times CC \quad (1)$$

For land cover types other than grasslands, constant values for Man and CC are attributed based on ecosystem survey, GeoEye-1 imagery and K2 literature.

The saturated hydraulic conductivity (Ks) value based on the soil texture is modified (Ksnew; mm $hr^{-1}$) to take into account the effects of plants (Stone et al., 1992) as follows (Eq. (2)):

$$Ksnew = Ks \times \exp^{(0.0105 \times CC)} \quad (2)$$

### 3.2.5 Derivation of channel parameters

Channel properties are less documented than the plane parameters. Considering the material in the channels (mostly a fine texture sandy loam, no gravels, few bedrock outcrops), the rather simple geometry (few braided channels) and the low number of scattered trees (found in the downstream part), a Man between 0.025 s $m^{-1/3}$ and 0.032 s $m^{-1/3}$ would be the best guess (Barnes, 1987). The best guess for Ks, assuming the material is a mixture of silty soils and sandy soils, which are eroded in the upper watershed, gives a somewhat larger interval, ranging from 25 mm $hr^{-1}$ (silty soil dominating, using Cosby pedotransfer function or the AGWA scheme) to 50 mm $hr^{-1}$ (sandy soil dominating). Instead of taking the mean values of these intervals, which would give Man = 0.0285 s $m^{-1/3}$ and Ks = 37.5 mm $hr^{-1}$, we preferred to use an optimization method using the annual discharge over the 2011−2015 period (n=5). This period benefits from numerous and accurate observations (named thereafter channel setup period). The benefit of the optimization is to check that the runoff simulated over the planes matches the observed flow at the outlet with reasonable values of the channel parameters.

A total of thirty sets of Ks and Man parameters values were used to sample the 10−50 mm $hr^{-1}$ and 0.01−0.05 s $m^{-1/3}$ domain, corresponding to large intervals derived from general literature for semi-arid zones (Chow, 1959; Estèves, 1995; Peugeot et al., 2007). Assessment of the channel parameters is not fully automated and requires a large number of simulations and post processing, hence the limited number of parameter values tested. Each set of parameters is used to run the ten simulations corresponding to a disaggregated precipitation ensemble (see Sect. 3.2.1). Parameters leading to the lowest bias (Eq. (3)) as well as a reasonably low Root Mean Square Error (RMSE) value (Eq. (4)) on the annual discharge are retained and used for all simulations (past and present periods).

$$Bias = \left| \left( \frac{\sum_{i=1}^{n} sim_i - obs_i}{\sum_{i=1}^{n} obs_i} \right) \right| \quad (3)$$

$$RMSE = \sqrt{\frac{1}{n} \sum_{i=1}^{n} (obs_i - sim_i)^2} \quad (4)$$

Where *sim* is the simulated annual discharge, *obs* is the observed annual discharge at the time *i* and *n* is the number of data available.

### 3.2.6 Model evaluation

K2 is evaluated in terms of bias and RMSE for all years with available discharge observations during the 2000−2010 period (2000, 2001, 2002, 2007, 2009 and 2010; n=6, named validation period) as well as for all years with available discharge observations over the past period (1965, 1966, 1973, 1975, n=4). The years of the channel setup period (2011−2015) are not considered in the evaluation.

### 3.3 Reference and attribution simulations of the Agoufou watershed

Two reference simulation cases are designed together with a suite of academic simulations to quantify and rank the effects of the landscape and meteorological changes observed over time.

The first reference simulation is the "present case", which builds on the soil and vegetation map of 2011, with a simulation period extending from 2000 to 2015 (n=15). The present case, which has the highest number of observations available, combining the channel setup and evaluation period, is considered as the "baseline" simulation. The second reference case is the "past case", which builds on the soil and vegetation map of 1956, with a simulation period extending from 1960 to 1975 (n=15).

From the present case, a suite of landscape changes, identified through the comparison of the Agoufou watershed in 1956 and 2011 (see Sect. 4.1), lead to simulations C, D, V and S, and a meteorological change leads to simulation P. These changes are implemented in the model first independently, then in combination. The simulation setup is summarized in Table 5 together with the associated forcing.

The impact of the different factors considered in the different simulations is expressed as a fraction of the difference between present and past mean annual discharge (*Ex* in %, Eq. (5)), with 100 % corresponding to the past discharge and 0 % to the present.

$$Ex = \frac{(AQpr - AQx) \times 100}{(AQpr - AQpa)} \tag{5}$$

Where *AQpa* is the past annual discharge, averaged over 1960−1975, *AQpr* is the present annual discharge averaged over 2000−2015 and *AQx* the annual discharge of each simulation. The different factors can therefore be ranked according to their effect on runoff. Additional simulations (CD, VS, and CDVS) also address the effects of factors combination.

### 3.4 Sensitivity analysis and spatial evolution

A sensitivity analysis was carried out to assess the robustness of the model in ranking the factors responsible for the increase of surface runoff, considering the uncertainties associated to planes and channel parameters.

According to sensitivity studies previously carried out for K2 in semi-arid area, the Ks and the Man are the most important parameters affecting the simulated surface runoff (e.g. Al-Qurashi et al., 2008; Smith et al., 1999). A first sensitivity test was carried out on planes Ks and Man. The range of variability in plane parameters was based on data compiled by Casenave and Valentin (1989) for Sahelian soils, resulting in a factor of 2.5 for Ks and 1.75 for Man intervals. A second sensitivity analysis was carried out for the channel parameters, to compare the parameters giving the lowest RMSE and the lower bias during the channel setup period (2011-2015).

To represent the spatial evolution of these two sensitive hydrological parameters (Ks and Man) and surface runoff (Q) within the watershed, watershed maps were constructed for the monsoon period over the past and the present period.

## 4 Results

### 4.1 Soil and land cover maps derived for 1956 and 2011

The 1956 and 2011 land cover maps are presented in **Fig. 3a** and **Fig. 3b**, together with the corresponding drainage networks. For each landscape unit, the difference between these two periods has been computed (**Fig.** 3**c**).

***Drainage network and flooded zones (F):*** The drainage network significantly increased between the two periods, with a total channel length of 71 km in 1956 against 104 km in 2011, corresponding to a drainage density increased by a factor of 1.5. Four zones (Z1, Z2, Z3, and Z4) underwent a particularly strong development of the drainage network (Table 6 and **Fig. 4**). Furthermore, a fraction of the watershed, located in the western region, has become a contributing area of the watershed in 2011, as can be seen by the active drainage network development. Floodplains (F1) have also expanded from 6.5 km² in 1956 to 12 km² in 2011. This change coincides with an important increase of the open water area (F2), especially marked for the watershed outlet (the Agoufou lake).

***Sandy soils (S):*** Sandy dunes (S2) and deep sandy soils (S3) exhibit limited changes. The total surface of these two units is 54 % of the total watershed in 2011 against 60 % in 1956. The conversion of S2 into agriculture enclosure (S4) explains most of this change, since enclosure occupy 10 km² in 2011 against 2.5 km² in 1956. Isolated dunes (S1) are found at the same location for both periods, but have been eroded and partially encrusted. Today, approximately 30 % of their surface is covered by crusts (i.e. 30 % of S1 in 1956 correspond to S1c in 2011). Overall, the sandy soils represented 63 % of the total watershed in 1956 and 60 % in 2011. Their hydrological properties are similar for the present and the past periods, except for crusted isolated dunes, which represent 0.36 % of the total watershed in 2011 and were not detected in 1956.

***Outcrops (O):*** Conversely, outcrops markedly developed in the northern part of the watershed. For instance, large areas in the northeastern part changed from pediment to O2 outcrops. Overall, the surface of the outcrop classes has increased from 18 km² in 1956 to 27 km² in 2011.

***Pediment (P):***

Although the overall proportion of Pediment class (P) on the watershed has not really changed between 1956 and 2011 (24% and 26% respectively), this unit underwent great changes within the Pediment class. Indeed, all tiger bush units (P3) have completely disappeared leaving impervious denudated soils (P4), sometime with some rare trees or bushes, witnesses of the old tiger bush. In addition, the silt layer (P2) has increased from 7 km² to 11 km². The watershed map of 1956 also shows a large central area occupied by shallow sandy soils (P1v) that has largely disappeared and has been replaced by mostly impervious rocky Pediments (P1). This last landscape unit occupied 10 % of the total watershed in 1956 against 19 % in 2011.

These changes strongly impact the watershed hydrological properties, since P1 and P4 favor surface runoff compared to P1v and P3 (Table 3). In addition, the silt layer (P2) has increased from 7 km² to 11 km², mainly in areas where the drainage network highly developed, reflecting the transition from sheet runoff to concentrated runoff and/or the increase of overland flow.

### 4.2 Model evaluation

The best agreement between observed and simulated annual discharges for the channel setup period is obtained with a channel Man of 0.03 s $m^{-1/3}$ and a channel Ks of 30 mm $hr^{-1}$ (Table 7), which is close to the best guess for these parameters. The corresponding bias is 2.7 % of the averaged discharge and the RMSE is equal to $6.4\times10^5$ $m^3$ (corresponding to 2.6 mm $yr^{-1}$ over the total watershed, n=5). As expected, several combinations of Man and Ks give close results, higher Man compensating lower Ks. The optimized values of Man and Ks correspond to rather impervious channels, which infiltrate much less than what is found in the literature for Sahelian watersheds, which reports a Man close to 0.03 s $m^{-1/3}$, but a Ks commonly reaching 150 to 250 mm $hr^{-1}$. These studies however concern particularly sandy areas where channels are several meters deep and are sometimes preferential infiltration sites (Chow, 1959; Estèves, 1995; Peugeot et al., 2007; Séguis et al., 2004). Conversely, the Agoufou watershed is characterized by silted very shallow soils or outcrops (northern part of the watershed) and silted channels, which is consistent with lower values of Ks.

For the validation period (6 years with available observation during the period 2000 to 2010), the bias on the annual discharge is -1.2 % and the RMSE is $1.1\times10^6$ $m^3$ (4.5 mm $yr^{-1}$; n=6) showing that the model performs reasonably well. For both the channel setup and validation periods, the inter-annual variability of the simulations is slightly greater than the observed one, with an under estimation for 2000-2001 (mid to low discharge years) and an over estimation for 2010 and 2013 (high discharge year, **Fig. 5**). The intra-annual variability of the simulated discharge is also reasonably close to the observations, considering the significant scatter of the simulated ensembles due the statistical rainfall disaggregation.

Overall, the annual cumulated discharge are close to the observations, with simulated mean annual discharges of $3.72\times10^6$ $m^3$ (15.2 mm $yr^{-1}$; n=5) and $3.85\times10^6$ $m^3$ (15.7 mm $yr^{-1}$; n=6) for the channel setup and validation period respectively against $3.42\times10^6$ $m^3$ (14 mm $yr^{-1}$; n=5) and $3.47\times10^6$ $m^3$ (14.1 mm $yr^{-1}$; n=6) for the observations. The mean relative bias between observed and simulated discharge during the whole period is 0.5 % (n=11) with a RMSE of $9.35\times10^5$ $m^3$ (3.8 mm $yr^{-1}$; n=11).

### 4.3 Long term evolution and attribution of changes

#### 4.3.1 Long term evolution

Results from the past and present periods are compared to all available observations of annual discharge (**Fig. 6**), namely 4 years for the past and 11 years for the present. Both observed and simulated discharges showed an important increase over time, with a mean observed discharge of $0.02\times10^6$ $m^3$ (0.08 mm $yr^{-1}$; n=4) and $3.43.10^6$ $m^3$ (14 mm $yr^{-1}$; n=11) for the past and present periods respectively to be compared with $0.5\times10^6$ $m^3$ (2 mm $yr^{-1}$; n=15) and $3.29\times10^6$ $m^3$ (13.4 mm $yr^{-1}$; n=15) for the simulations. For the past period, simulations overestimate the annual discharge for the four years with observations. The observed runoff coefficient over the whole watershed (ratio between annual discharge and the precipitation over the total watershed area) is estimated at 0.02 % (n=4) and 4.0 % (n=11) for the past and the present periods respectively against 0.55 % (n=15) and 3.87 % (n=15) for the simulations.

Despite the past simulations being overestimated and modeled variability being slightly larger than observed variability, both observations and simulations indicate a marked change in the watershed behavior between the

past and present periods, with a discharge increase of an order of magnitude. Precipitation during the present period averages 347 mm and displays a significant inter-annual variability, with extreme dry (2004, 2014) and wet years (2010, 2011). During the past period the precipitation average is equal to 382 mm, which is slightly above current value, and displays a smaller inter annual variability, in line with what is commonly observed in the Sahel (Lebel and Ali, 2009). The relation between event precipitation and discharge (**Fig. 7**) highlights important differences between the past and the present. First, for the same precipitation amount, the discharge is twice as large in the present as in the past. Second, for the present period, rainfall events larger than ~18.8 mm contribute to the discharge whereas in the past, rainfall events larger than ~30 mm were required.

#### 4.3.2 Attribution of changes

Results from the K2 simulations outlined in Table 5 are summarized in **Fig. 8** and **Table. 8** and are discussed in details below. The mean discharge (*mQ*) and the standard deviation (*Sd*) for each simulation are calculated for the ten members. The reference simulation corresponds to the simulation of the present case and the impact of the different factors tested by the different simulations is assessed through the *Ex* values (see Eq. 5).

***Dune Crusting (C):*** This simulation corresponds to present characteristics without dune crusting. Replacing 30 % of the crusted dune area by dune without crusting has two effects on the land surface: first, soil Ks increases (more infiltrability), and, second, the growth of herbaceous vegetation is made possible. These two effects favor infiltration and limit surface runoff generation. The overall effect of removing dune crust on the annual discharge is minor as it only explains **1 %** (*Ex*) of the past to present evolution.

***Drainage network Development (D):*** The present drainage network is replaced by the past network, meaning that the network development of the four sub-basins is deactivated and the contribution of the western part sub-watershed is forced to zero (by adding deep sandy channels that mimic the sand dunes interrupting the water flow). Overall, this factor explains **22 %** of the surface runoff increase over time. The western part of the watershed (Z1) is currently connected to the principal drainage network, and produces a runoff of $3.3\times10^4$ $m^3$ (0.1 mm $yr^{-1}$) while the contribution of the network development over the Z2, Z3 and Z4 is equal to $3.6\times10^5$ $m^3$ (1.5 mm $yr^{-1}$) $1.5\times10^4$ $m^3$ (0.06 mm $yr^{-1}$) and $1.2\times10^4$ $m^3$ (0.05 mm $yr^{-1}$) respectively. Overall, changes of the network drainage in the northern areas, where shallow soils and outcrops are found, have the largest impact on the simulated discharge.

***Vegetation changes (V)***: This simulation tests the impact of the herbaceous vegetation expansion on the annual discharge. It is implemented independently from soil type changes (see below), which is mostly academic since vegetation and soil type are most often tightly related. Nevertheless, such a simulation is useful for guiding for instance future model development and helps decipher the physical factors impacting runoff. For each plane, the soil texture is kept at present values and the fraction occupied by herbaceous vegetation depends on the past maps (**Fig. 3a** and Table 4). This past map is characterized by the presence of shallow sandy soil (P1v) and deep sandy soils (S1, S2, S3 and S4), totaling 75 % of the watershed area, over which annual herbaceous plants can grow. The seasonal growth and inter-annual variability is forced by present day precipitations not to interfere with simulation “P”. Vegetation efficiently slows surface runoff and increases infiltration capacity. As a result, simulation “V” produces a discharge of $2.1\times10^6$ $m^3$ (8.6 mm $yr^{-1}$) and herbaceous vegetation changes explain **42 %** of the difference in surface runoff between past and present.

***Change in soil properties (S):*** This simulation tests the impact of soil changes on annual discharge without changing the herbaceous cover fraction, which is also an academic simulation. The fraction of all landscape units in each planes is defined by the past land cover map (**Fig. 3a**). The increase in some pediment units and outcrops over time results in a very strong impact on soil hydrological properties and then on the annual runoff. The simulation "S" produces a discharge of $0.67\times10^6$ m$^3$ (2.8 mm yr$^{-1}$) and explains **95 %** of the change in annual discharge. Note that some landscape units, like tiger bush, comprise a fixed fraction of thickets, so that the simulation "S" accounts for changes in woody vegetation and thickets in addition to soil texture.

***Precipitation (P):*** Precipitation impact is investigated by running K2 using past daily precipitations (1960-1975) and the watershed characteristics of the present period. As opposed to the previous attribution simulations, the simulation "P" produces a discharge of $4.09\times10^6$ m$^3$ (16.7 mm yr$^{-1}$), or an increase of $0.8\times10^6$ m$^3$ (3.3 mm yr$^{-1}$) compared with the present, at odds with its observed reduction. This is an expected result since the precipitation average is slightly higher in the past period. Therefore, this factor results in a negative value of *Ex* (**-29 %**).

***Crusting and Drainage network combination (CD):*** This simulation combines the first two attribution cases (dune crusting and drainage network development). The combination of these two factors explains only **23 %** of the difference of annual discharge between the two periods, which is equal to the sum of the two factors taken separately (1 % and 22 %).

***Vegetation and Soil combination (VS):*** This simulation combines the effects of herbaceous vegetation map and soil type changes. Taken together, these two effects explain **101 %** of the difference in annual discharge between the two periods. The two factors do not impact runoff additively (101 % to be compared to 95 % plus 42 %), but are clearly strong enough to account for the observed changes in the watershed outflow.

***Crusting, Drainage network, Vegetation and Soil combination (CDVS):*** Last, this simulation combines four factors. It corresponds to the past case fed with the present precipitations. All these cumulated changes explain **105 %** of the difference of mean annual discharge between the two periods.

#### 4.3.3 Spatial evolution over the watershed

The impact of landscape changes on the spatial distribution of Man and Ks within the watershed are presented in **Fig. 9a** and **Fig. 9b** respectively. Ks is predominantly related to the changes in the soil units while Man corresponds to the vegetation cover. The replacement of planes where shallow sandy soils dominated by planes with less pervious pediment units (see **Fig. 3**) led to important changes in Ks (**Fig. 9**). Planes subject to a decrease in vegetation cover display a decrease of the Man values (typically of the order of 0.1 s m$^{-1/3}$ for the past period and smaller than 0.05 m s$^{-1/3}$ for the present period). These local changes have induced an important and spectacular increase of the surface runoff generated over these planes (**Fig. 9c**): the contributing part of the watershed almost doubled with 20 % of the watershed contributing in the past against 37 % in the present period.

### 4.4 Model robustness

The sensitivity analysis results are assessed by a comparison between the *Ex* values obtained by the attribution simulations for a reference case and the values obtained by the two sensitivity tests (Table 8) on planes and channel parameters. The reference case and the sensitivity tests are run using only one precipitation member,

namely the closest to the ensemble average. This indeed leads to *Ex* values (Table 8, Column 3) very close to those reported for the attribution simulations on the 10 members ensemble (section 4.3.2). Regarding the first test, multiplying the Ks of all planes by 2.5 and Man by 1.75 decreases runoff by a factor of 3 (3.3 mm $yr^{-1}$ against initially 13.4 mm $yr^{-1}$ for the "PRES Ref" test), but it does not change the ranking of the different factors (Table 8, Column 3, 6 and 9), which is the primarily goal of these attribution simulations. The main difference in the *Ex* values is found for simulation "C", with a more important contribution of dune crusting to the runoff increase, for planes with higher Ks and Man. The second sensitivity test uses Ks equal to 40 mm $hr^{-1}$ (which gives the lowest RMSE for the channel setup period) instead of 30 mm $hr^{-1}$ (which gives lowest bias). This results in a small decrease in total runoff in all simulations but it does not change the ranking of the different factors. The impact of the drainage network development is however sensibly lower than in the reference simulations, which is consistent with channels with higher infiltration capacity. Overall, these two sensitivity tests show the robustness of our results concerning the ranking of the different factors contributing to runoff changes between the past and the present. In particular, the impact of vegetation changes and the evolution of soils properties, which alone are sufficient to simulate the past-present difference, is a robust feature.

# 5 Discussion

## 5.1 Watershed evolution

The maps of landscape units were derived from different data (aerial photography and satellite images) of various spatial and spectral resolutions. Delimitation and identification of the landscape units proved easier for the present than for the past. Panchromatic aerial photographs give limited information and in many occasions, the 3D visualization is necessary to clearly identify the units. Photo-interpretation for the past aimed to be conservative, meaning that obvious changes only were retained while ambiguous cases were considered as "no change". Overall, despite the uncertainties related to the photo-interpretation and mapping of landscape units, that are not easily estimated, the land surface changes of the Agoufou watershed are important and clearly observable.

The results reported in **Fig. 3** highlight the increase in drainage density, which reaches a factor of 1.5 over the whole watershed and is accompanied by an expansion of open water surface (F2) and alluvial plains (F1). Similar changes were also observed by Massuel (2005) and by Leblanc et al. (2008) in southwestern Niger, where the drainage density increases by a factor of more than 2.5 between 1950 and 1992, as well as in a small watershed in northern Mali by Kergoat et al. (2015), who reported a factor of 2.8 between 1956 and 2008. Considering that few changes are observed on the southern sandy part of the basin, the evolution of the drainage network for Agoufou is consistent with the values found in the literature.

Woody vegetation, and especially the thickets of the tiger bush unit, and some of the shallow sandy soils have completely disappeared and have been replaced by hard pan outcrops and silt surfaces which is consistent with several studies (Hiernaux and Gérard, 1999; Leblanc et al., 2007; Touré et al., 2010; Trichon et al., 2012). Hiernaux et al. (2009b) have observed that the woody vegetation of the Gourma region has declined since the 1950s and particularly from 1975 to 1992 over shallow soils. Given that tiger bush thickets grow perpendicularly to the water flow and therefore protect the soil against erosion, capture runoff, and favor infiltration (Valentin et al., 1999 among others), disruption of thickets leads to runoff concentration and changes the overall hydrological

properties. Sighomnou et al. (2013) in Niger and Kergoat et al. (2015) in Mali have also noted a significant decrease of vegetation over shallow soils and a corresponding increase in denudated surfaces. In a watershed in Niger, Touré et al. (2010) estimated that the tiger bush occupied 69 % in the 70s and has disappeared in the 2000s. Man-driven deforestation has been put forward as a cause for thickets clearing in southwestern Niger. For Agoufou, such activity is not reported and remains of dead trees can be observed, testifying natural death of vegetation.

The decay of shallow soil vegetation is not at odd with the Sahelian regreening that is observed all over the Sahel and in the Gourma since the 80s (Anyamba et al., 2003; Dardel et al., 2014a, 2014b; Heumann et al., 2007; Olsson et al., 2005). Dardel et al (2014b) suggest that the resilience of herbaceous vegetation allows rapid regrowth over most soils in response to rainfall recovery, but that a fraction of the shallow soils may undergo long term vegetation decay, in a way that impacts runoff but not region-average greening. In the Agoufou watershed, the vegetation changes affecting the northern part of the watershed would not be easily detected by coarse resolution satellite datasets, as opposed to the herbaceous vegetation growing over the sandy soils of the southern part of the watershed. In addition, the greening trend is obvious since the 80s, because of the maximal drought of 83 and 84, but the longer term trend is likely to be different (e.g. Pierre et al., 2016).

As far as land use change is concerned, a few additional enclosures (made of dead-wood fences whose prime function is to delineate land rights) are present nowadays in the Agoufou watershed, as a result of an easier access to water year-round since the lake became permanent. Located on deep sandy soils not contributing sensibly to runoff, these enclosures do not impact the overall characteristics of the watershed. In that respect, Agoufou differs from most of the watersheds studied in the Sahel so far (Niger, Burkina-Faso).

### 5.2 A significant discharge increase found despite the simulation limitations

If simulations and observations are in good agreement for the present period, simulated discharge for the past period is overestimated ($0.51\times10^6$ m$^3$ and $0.02\times10^6$ m$^3$ for simulation and observation respectively or 2.1 mm yr$^{-1}$ and 0.08 mm yr$^{-1}$). Different reasons could explain this. First, the error bars in **Fig. 8**, which represent the standard deviation of the ten members used for the ensemble simulations, illustrate the high sensitivity of the model to precipitation intensity. Moreover, the intra-annual dynamics (**Fig. 6**) also reflects the sensitivity of the model to a limited number of rainfall events each year. By assumption, the 5M rainfall intensity is supposed to be the same (i.e. to have the same distribution) for the past and the present periods. Lower 5M intensities in the past than in the present could then lead to lower simulated discharge values, which could come closer to observations. However, there is no evidence of changes in precipitation intensity at this short time-scale (Panthou et al., 2014). Second, the model was evaluated with all available observation data over the 2000−2015 period, when data are the most accurate and numerous. During the past period, only few data are available (four years) and the estimation of the annual discharge is less precise (see Gal et al., 2016) which could also account for part of the discrepancies between simulated and observed mean discharge in the past.

As highlighted in section 3.2.1, a possible time lag in precipitation falling within the watershed would have an impact on the instantaneous amount of water ending up into channels. In our simulation setup, channels are calibrated, so a possible time lag in precipitation would have as consequence an overestimation of channel

conductivity during calibration. However, this would not sensibly impact our results, which have been shown to be robust to changes in channel characteristics.

Moreover, all channels were considered to have the same characteristics over time and space but field observations suggest that channel properties vary according to their geographical position, channels being possibly more permeable in the southern part than in the northern part, where they are also shallower. In addition, increasing surface runoff over time could contribute to erode the soil surface and to increase sediments transport along channels downstream. If the sediment texture were mostly clay and silt, channels may have become more impervious, thus increasing runoff at the outlet. Less impervious channels in the past may therefore explain model overestimation. However, literature reports the reverse situation in the Sahel (Séguis et al., 2002) with materiel particularly rich in sand being transported. As for gully depth and soils, it is not clear whether the different Sahelian watersheds studied so far are comparable, given the importance of shallow soils and silt in the northern part of the Agoufou watershed.

Despite the possible sources of uncertainty previously identified, the difference between observations and simulations ($0.14\times10^6$ m$^3$ and $0.49\times10^6$ m$^3$ or 0.5 mm yr$^{-1}$ and 2 mm yr$^{-1}$) is largely below the difference between the past and the present period ($3.41\times10^6$ m$^3$ and $2.78\times10^6$ m$^3$ for the observation and simulation discharge respectively or 13.9 mm yr$^{-1}$ and 11.3 mm yr$^{-1}$, see **Fig. 7**). Simulated mean discharges for the present and the past periods are significantly different (t-test for means equality) as it is the case for observations (Gal et al., 2016).

### 5.3 Attribution of the Sahelian paradox

Changes in vegetation and soil properties alone are sufficient to simulate the observed increase in watershed runoff over time. Previous studies of Sahelian hydrology agree on the major role of surface conditions on erosion and runoff generation (Casenave and Valentin, 1990; D'Herbès and Valentin, 1997; HilleRisLambers et al., 2001; Peugeot et al., 1997; Rietkerk et al., 2002). For the Agoufou watershed, the comparison of past and present land cover maps indicates that vegetation, mainly the dense thickets but more generally vegetation growing on shallow soils, has decayed after the severe droughts in 1972-73 and again in 1983-84 (Dardel et al., 2014b; Hiernaux et al., 2009b). The lack of vegetation recovery during the long drought period combined to erosion of shallow soils and runoff shift from sheet runoff to concentrated runoff is in agreement with findings by Séguis et al. (2004) who estimated, using hydrological modeling, that changes in land cover on the Wankama watershed had multiplied the mean annual runoff by a factor close to three for the 1950–1992 period. Valentin et al. (2004) have also shown that a general decrease in vegetation cover modified the hydraulic properties of the topsoil and led to an increase in Hortonian runoff collected in numerous gullies and ponds. Our study highlights the predominant role of land cover changes in a pastoral area as opposed to several studies conducted in cropland dominated areas, which pointed to the leading role of the land use changes on surface runoff changes (Albergel, 1987; Favreau et al., 2009; Leblanc et al., 2008; Mahé et al., 2005).

Drainage network development is a key marker of ecosystem degradation (Descroix and Diedhiou, 2012; San Emeterio et al., 2013). Studies of the direct impact of this phenomenon on surface runoff are scarce. The changes in drainage density shown in this study could be explained by the acceleration and/or concentration of surface runoff (due to vegetation decay) allowing gully erosion to develop in susceptible areas like on silty or sandy soil

for example (Leblanc et al., 2008; Marzolff et al., 2011; Poesen et al., 2003; Valentin et al., 2005). An increase in both the number and the length of channels reduces the travel time for water to reach the drainage network (which increases the total water flow when channels are more permeable than to planes, as in our case). An increase of drainage density was also reported by Leblanc et al. (2008) in Niger. In semi-arid regions, gullies tend to enhance drainage and to decrease the water supply for the vegetation growing on planes and slopes (as opposed as along gullies or downstream), providing therefore a positive feedback to the vegetation decay /enhanced runoff system (Leblanc et al., 2008; Valentin et al., 2005).

Crusts are frequently cited as a possible explanation of the Sahelian paradox (Favreau et al., 2009; Leblanc et al., 2008; Mahé and Paturel, 2009). Our results suggest that the impact of crusted sandy dune on the surface runoff is limited. This is not necessarily the case further south like in southwestern Niger, where some soils have a higher percentage of clay. Moreover, soil crusting in the Agoufou landscape may be slightly underestimated given the low resolution of aerial photographs in 1956. Trampling by livestock, not considered here, has an unclear impact on soil crusting: according to the work by Hiernaux et al. (1999) on sandy soils in Niger, the soil infiltration capacity slightly increases with moderate grazing, but decreases at higher stocking rates. Moreover, the evolution of the stocking rates is poorly known over 1956-2011, although an increase cannot be excluded. Besides, it should be noted that in the literature, vegetation degradation is sometimes classified as “increase in surface crusting”, while in this study changes from tiger bush vegetation into impervious soil, which are crusted, are simulated as part of vegetation and soil changes (“VS” simulation).

Finally, the increase in the occurrence of extreme rainy events in daily precipitation suggested by Frappart et al. (2009) and demonstrated by Panthou et al. (2012, 2014) is intrinsically taken into account by the use of daily precipitation series used to force the model in our study. The results suggest that changes in daily precipitation regime do not explain runoff changes between the past and the present. If this variable is only taken into account (simulation “P”), simulated surface runoff decreases rather than increasing over time. This is in line with Descroix et al. (2012), Cassé et al. (2015) and Aich et al. (2015), who found that the modest increase in large rainfall amount (events > 40 mm) observed during the 2000s cannot alone explain the Sahelian paradox. However, this should be taken with caution because changes in precipitation not statistically detectable here may have occurred elsewhere, due to the high natural variability, and further studies are required to address this question into more details.

## 6 Conclusions

In this study, a modeling approach was applied to investigate the paradoxical evolution of surface hydrology in the Sahel since the 60s. Landscape changes between 1956 and 2011 over the Agoufou watershed display four major features: 1) a partial crusting of isolated dunes, 2) an increase of drainage network density, with the connection of the western part of the watershed, 3) a marked evolution of the vegetation with the non-recovery of tiger bush and vegetation growing on shallow soils after the drought, 4) a marked evolution of soil properties with some shallow soils being replaced by impervious soils (hard pans, outcrops or silt flats) probably following erosion.

These changes were implemented independently and in combination in the KINematic runoff and EROSion model (K2) to quantify and rank their impact on mean annual discharge. According to the model, changes in soil

properties and vegetation (grassland and tiger bush thickets) are large enough to reproduce the increase of surface runoff observed between the past (1960−1975) and the present period (2000−2015), with the drainage network density also contributing to this effect. The non-recovery of vegetation (woody and herbaceous) growing on shallow soils resulted in enhanced runoff, erosion, and drainage network development, in turn depriving vegetation from nutrient and water resources. According to our modeling results, these synergistic processes drive the Sahelian paradox in the absence of land use changes.

The results reported here provide new perspectives towards better understanding the Sahelian paradox through hydrological modeling. Our study points out the need of taking into account all these processes in models aiming at representing hydrological past, present and future evolution in this region. In addition, the important landscape changes observed in this area highlight the interest of long-term monitoring of vegetation and hydrological variables in this region at a fine spatial and temporal scale.

**Acknowledgement**

We thank Nogmana Soumaguel, Ali Maïga, Hamma Maïga and Mamadou Diawara for collecting data and Eric Mougin for managing the Gourma site and providing the STEP model. We also acknowledge Shea Burns and Carl Unkrich for their feedbacks on AGWA and K2. This research was based on data from the AMMA-CATCH observatory and partially funded by the ESCAPE ANR-project (ANR-10-CEPL-005).

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

| *Datasets* | *Type* | *Acquisition date* | *Sources* |
|---|---|---|---|
| DEM | SRTM (30 m) | 23 September 2014 | NASA |
| Satellite images and aerial photographs | SPOT (5m) | 19 March 2004 | CNES through Google Earth |
| | GeoEye-1 | 7 February 2011 | DigitalGlobe through Google Earth |
| | Aerial photographs | November 1956 | IGM: Instititut Géographique du Mali |
| Water outflow from the Agoufou watershed | Annual and intra annual | 1965, 1966, 1973, 1975, 1984, 1990, 2000-2002, 2007, 2009-2015 | Gal et al., 2016 |
| Precipitation data | Daily | 1920-2015 | Hombori (Mali), Direction Nationale de la Météorologie, (DNM) and AMMA-CATCH |
| | 5 min | 2006-2010 | Bangui mallam, Bilantao, Agoufou, Belia, Taylallelt, Nessouma, Hombori automatic raingauges (AMMA-CATCH network) |

**Table 1: Data available for the Agoufou watershed and used in input K2 model.**

| | | |
|---|---|---|
| **Sandy soils (S)** | **S1:**<br>Isolated dunes | Oval shaped isolated dune, often elongated in the direction of the prevailing northeasterly winds. Soil deflation and crusting may occurs creating patches prone to runoff (**S1c**). |
| | **S2:**<br>Dunes system | Large sandy areas with succession of dunes and inter-dunes where the soil is deeper than 200 cm and has a very high infiltration capacity. |
| | **S3:**<br>Deep sandy soil over bedrock | Sandy sheets typically 30 to 200 cm deep topping bedrock. Hydrological characteristics are close to those of the dune systems (S2) as the soil retention capacity is seldom exceeded. |
| | **S4:**<br>Enclosures | Enclosures sometimes cropped with millet located on sandy soil near water reaches. Hydrodynamic characteristics are close to those of the dune systems although land use is different. |
| **Outcrops (O)** | **O1**:<br>Rocky outcrops | Rocky outcrops correspond to schist or sandstone and are mostly devoid of vegetation. Infiltration is limited and most rainfall runs off. See also P1. |
| | **O2**:<br>Hard pan outcrop | Hard pan outcrop largely devoid of vegetation. Infiltration is very low. See also P1. |
| **Pediments (P)** | **P1:**<br>Rocky Pediments | These Pediments (or "glacis") combine hard pan outcrops and rocky outcrops interspersed with shallow sand-loam bars and sand-silt linear shaped deposits. They are the consequence of water and wind erosion and deposition responsible for deflation and silting and they produce important runoff except where shallow sandy soils (< 30 cm) are dominant. In this case, an herbaceous vegetation layer may be present (**P1v).** |
| | **P2:**<br>Silt layer | These Pediments consist of a silt-clayed texture layer typically 30 to 100 cm deep laying on bedrock or hard pan, probably resulting from peri-desert silt. These soils are largely impervious and are a privileged area of runoff. |
| | **P3:**<br>Hard pan surface with tiger bush | Succession of bare surfaces and linear thickets made of a dense shrub population and a sparse herbaceous layer. This vegetation is often called “tiger bush”. Thickets are perpendicular to the slope and stop the runoff from the upstream bare patch. Banded vegetation grows on sandy-loam soil. The hydrological properties of the bare surface between thickets are those of impervious soils whereas thickets areas have high infiltration capacity. When not degraded, tiger bush systems produce little runoff (downstream) overall. |

| | | |
|---|---|---|
| | **P4:** P3 eroded | Degradation of the tiger bush results in eroded and crusted soils, which are largely impervious and produce important surface runoff. Traces of past woody vegetation can be observed (isolated thickets, dead logs). |
| **Flooded zones (F)** | **F1:** Alluvial plains | Floodplains are inundated during the largest rainfall events. This unit is characterized by alluvial sandy-loam or silty-clay soils. Large trees commonly grow along the channels. |
| | **F2:** Open Water | Ponds formed in depressions during the rainy season and permanent lakes (Agoufou lake in the study area). |

**Table 2: Characteristics of the landscape units (soil and land cover type, and hydrological properties).**

| Infiltration | Landscape units | Ks | G | DIST | POR | FR (S-S-C) | THICK | SMAX |
|---|---|---|---|---|---|---|---|---|
| | | $mm\ hr^{-1}$ | mm | - | $cm^3\ cm^{-3}$ | % | mm | % |
| - | O1-O2-P1-S1c | 0 | 0 | 0 | 0 | 0-0-0 | n/a | 0 |
| | P2 | 5.82 | 224.2 | 0.38 | 0.414 | 5-17-28 | n/a | 86 |
| | P4-F1 | 11.82 | 108 | 0.25 | 0.463 | 36-41-22 | n/a | 94 |
| | P1v -P3 | 142.98 | 83.2 | 0.59 | 0.435 | 80-9-11 | 300 | 96 |
| + | S1-S2-S3-S4 | 192.13 | 46 | 0.69 | 0.437 | 92-3-5 | n/a | 96 |

**Table 3: Summary of hydrological parameters for each landscape unit, sorted by increasing infiltration (Ks: saturated hydraulic conductivity, POR: soil porosity, G: capillary charge saturation, DIST: pore distribution, FR: fraction of sand, silt and clay, and THICK: upper soil thickness).**

| code | Vegetation type | Landscape unit | Man (s $m^{-1/3}$) | CC (%) | Ksnew (mm $hr^{-1}$) |
|---|---|---|---|---|---|
| GT | Grassland + Trees | S1-S2-S3-S4 | 0.008×CCd | CCd | Ks×exp $^{(0.0105\times CCd)}$ |
| G | Grassland | P1v | 0.008×CCs | CCs | Ks×exp $^{(0.0105\times CCs)}$ |
| T | Sparse trees | P4 | 0.05 | 3 | Ks×exp $^{0.0315}$ |
| TB | Tiger bush thickets | P3 | 0.6 | 30 | Ks×exp $^{0.315}$ |
| W | Woody plant | F1 | 0.05 | 20 | Ks×exp $^{0.21}$ |
| R | No vegetation | O1-O2-P1-P2-S1c | 0.001 | 0 | 0 |

**Table 4: Summary of the different land cover types and their hydrological parameters (Man: Manning's roughness coefficient, CC: canopy cover and Ksnew: saturated hydraulic conductivity modified).**

| Simulations | Crusted dune | Drainage Network | Vegetation | Soil | Precipitations |
|---|---|---|---|---|---|
| PRES | Present | Present | Present | Present | Present |
| C (Crusted dunes) | Past | Present | Present | Present | Present |
| D (Drainage network) | Present | Past | Present | Present | Present |
| V (Vegetation) | Present | Present | Past | Present | Present |
| S (Soil) | Present | Present | Present | Past | Present |
| P (Precipitation) | Present | Present | Present | Present | Past |
| CD | Past | Past | Present | Present | Present |
| VS | Present | Present | Past | Past | Present |
| CDVS | Past | Past | Past | Past | Present |
| PAST | Past | Past | Past | Past | Past |

**Table 5: Description of the simulations (1st column) and associated forcing (2nd to 5th column) for: the crusted dunes, the development of the drainage network, the evolution of the vegetation cover and soils and the change in the daily precipitation regime.**

| Zones | Area ($km^2$) | Total network length (km) | | Increase factor |
|---|---|---|---|---|
| | | 1956 | 2011 | |
| Z1 | 20.72 | 6.1 | 17.3 | 2.85 |
| Z2 | 30.17 | 10.1 | 23.4 | 1.93 |
| Z3 | 12.48 | 6.4 | 11.1 | 1.75 |
| Z4 | 3.5 | 0.2 | 3.4 | 14.01 |

**Table 6: For the four sub-watersheds, area, total drainage network length in 1956 and 2011 and the factor of increase between these two periods are given.**

| Ks ($mm\ hr^{-1}$) | Bias (%)<br>RMSE ($mm\ yr^{-1}$) | Man ($s\ m^{-1/3}$) 0. 01 | 0. 02 | 0.03 | 0.04 | 0.05 |
|---|---|---|---|---|---|---|
| **10** | Bias (%) | 75.30 | 50.40 | 29.80 | 9.90 | -9.20 |
| | RMSE | 11.32 | 8.00 | 5.34 | 3.17 | 2.54 |
| **20** | Bias (%) | 60.00 | 33.80 | 14.10 | -4.60 | -22.10 |
| | RMSE | 9.26 | 5.85 | 3.54 | 2.46 | 3.49 |
| **30** | Bias (%) | 48.40 | 21.80 | 2.70 | -15.10 | -31.30 |
| | RMSE | 7.71 | 4.36 | 2.61 | 2.88 | 4.50 |
| **40** | Bias (%) | 38.90 | 12.10 | -6.50 | -23.60 | -38.70 |
| | RMSE | 6.47 | 3.32 | 2.44 | 3.65 | 5.41 |
| **50** | Bias (%) | 30.70 | 3.90 | -14.20 | -30.70 | -44.90 |
| | RMSE | 5.43 | 2.67 | 2.80 | 4.44 | 6.20 |

**Table 7: Percent bias and RMSE on annual discharge over 2011-2015 for 25 sets of channels Ks and Man parameters. The minimum value for the percent bias and RMSE is indicated by the square box (red and black respectively).**

| | **Ref*** | | | **S-PL**** | | | **S-CH***** | | |
|---|---|---|---|---|---|---|---|---|---|
| | **mQ ($10^6\ m^3$)** | **mQ (mm)** | ***Ex* (%)** | **mQ ($10^6\ m^3$)** | **mQ (mm)** | ***Ex* (%)** | **mQ ($10^6\ m^3$)** | **mQ (mm)** | ***Ex* (%)** |
| **PRES** | 3.3 | 13.4 | 0.0 | 0.8 | 3.3 | 0.0 | 3.0 | 12.3 | 0.0 |
| **C** | 3.2 | 13.3 | 2.7 | 0.7 | 2.8 | 14.9 | 3.0 | 12.2 | 1.0 |
| **D** | 2.7 | 10.9 | 22.1 | 0.7 | 2.7 | 19.6 | 2.8 | 11.4 | 8.9 |
| **V** | 2.1 | 8.7 | 41.9 | 0.3 | 1.3 | 63.7 | 1.9 | 7.9 | 43.3 |
| **S** | 0.7 | 2.6 | 94.0 | 0.1 | 0.2 | 97.2 | 0.5 | 2.2 | 98.1 |
| **P** | 4.1 | 16.7 | -28.6 | 1.0 | 3.9 | -21.5 | 3.7 | 15.3 | -29.3 |
| **PAST** | 0.5 | 2.1 | 100.0 | 0.0 | 0.1 | 100.0 | 0.5 | 2.0 | 120.0 |

*: Initial simulation with default parameters presented in **Fig.8**
**: Simulation with Ks (×2.5) and MAN (×1.75) modified for all planes
***: Simulation with Ks (40 $mm\ hr^{-1}$) and MAN (0.03 $s\ m^{-1/3}$) modified for all channels

**Table 8: Mean annual discharge (mQ) and *Ex* values obtained for the initial simulation with default parameters presented in Fig.8 (Ref*) and the sensitivity test for plane (S-PL**) and channel (S-CH***) parameters.**

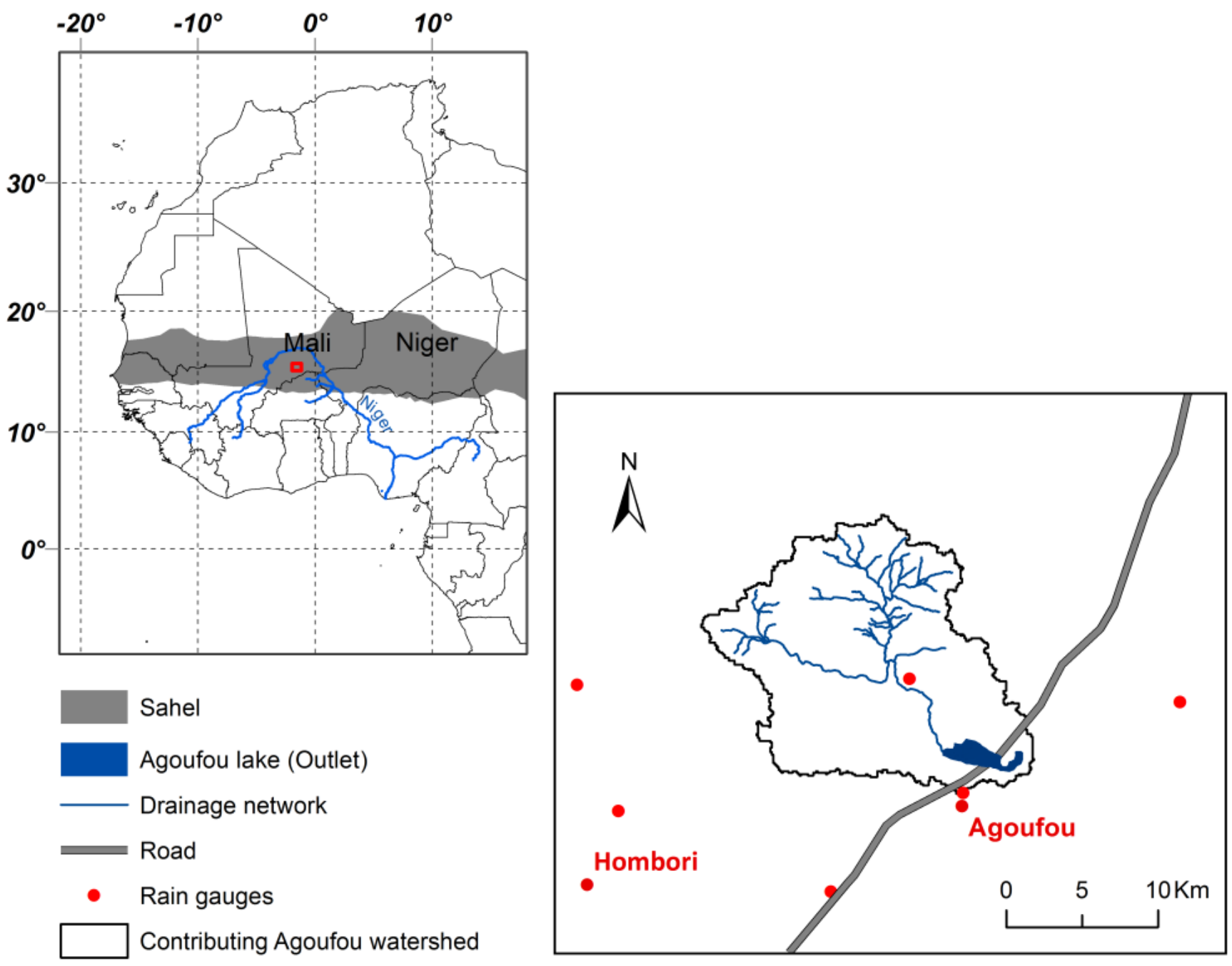

**Fig. 1: The Agoufou watershed (245 km²) located in western Sahel (Mali) with drainage network and available rain gauges (map source: http://www.diva-gis.org/gdata).**

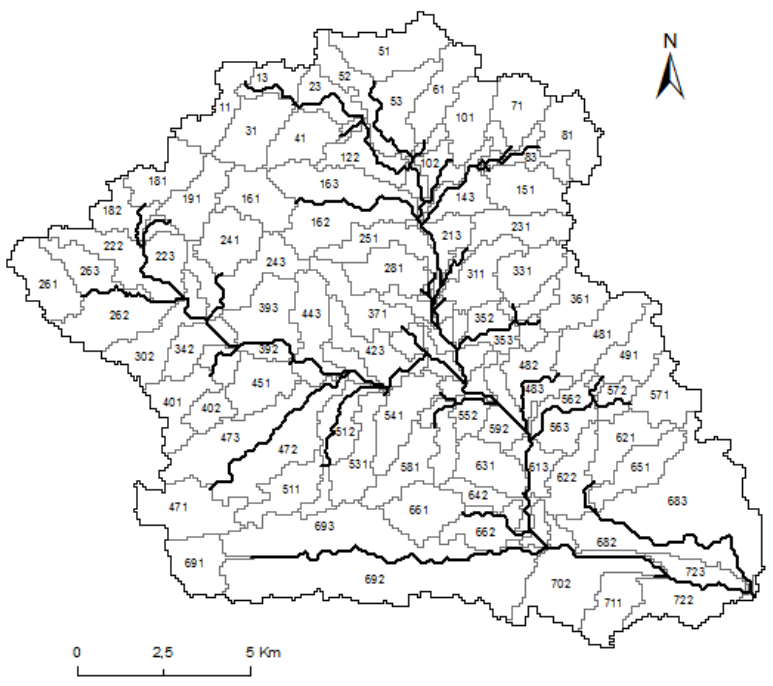

**Fig. 2: Planes for the Agoufou watershed with the DEM-derived drainage network.**

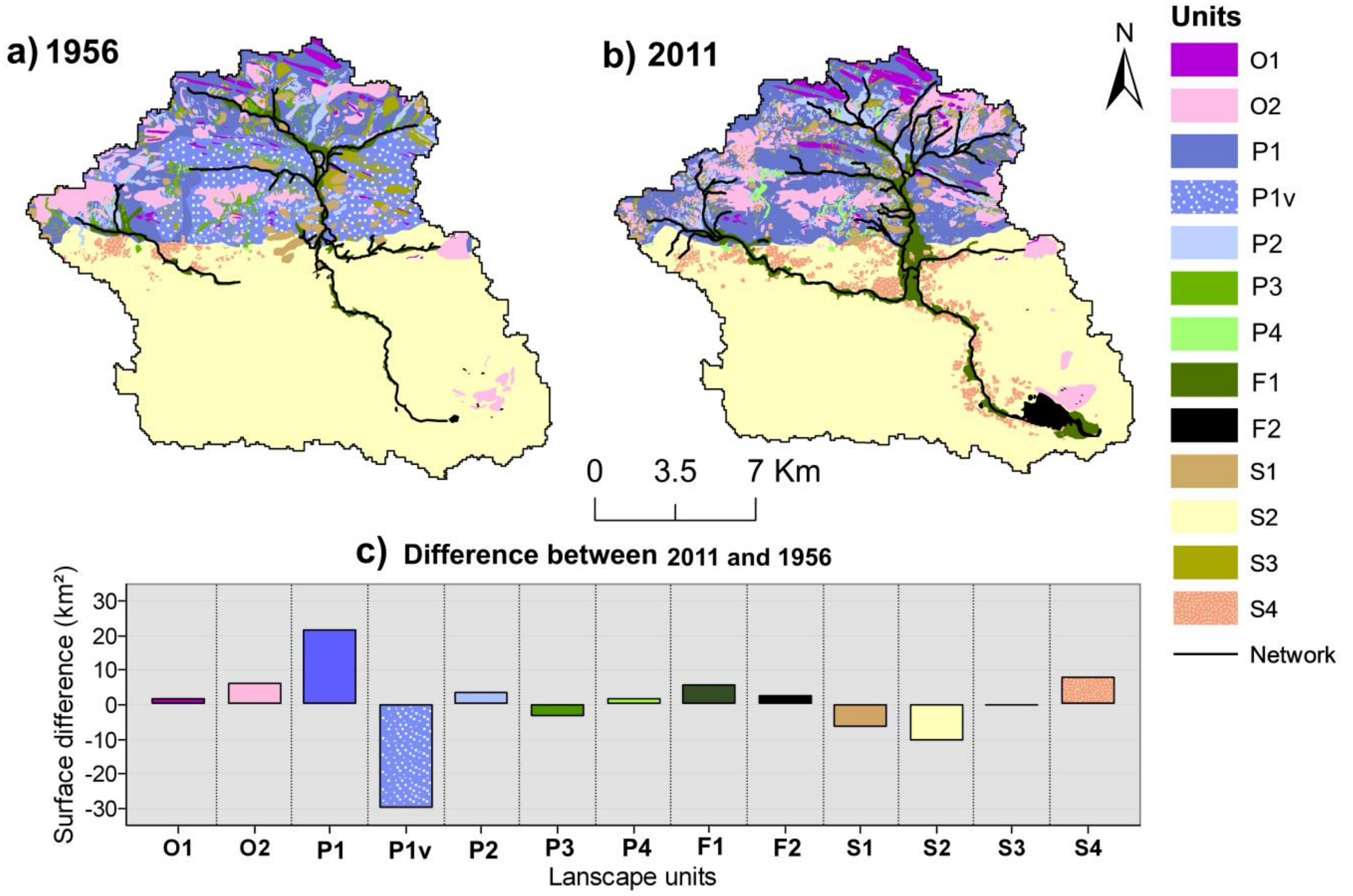

**Fig. 3: Land cover maps of the Agoufou watershed for (a) 1956, (b) 2011 and (c) gives the surface difference (in km²) for each landscape unit, between 2011 and 1956.**

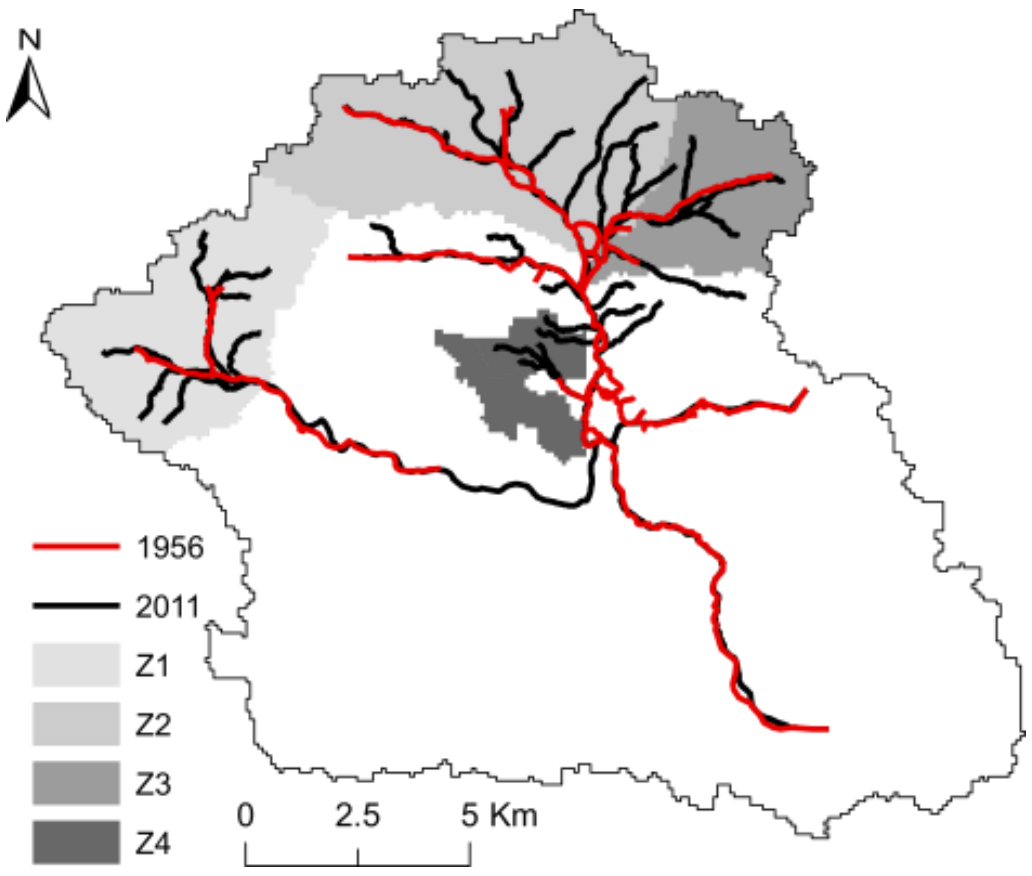

**Fig. 4: Identification of the four sub-watersheds (grey shades) which display the largest changes between the drainage network in 1956 (red line) and in 2011 (black line).**

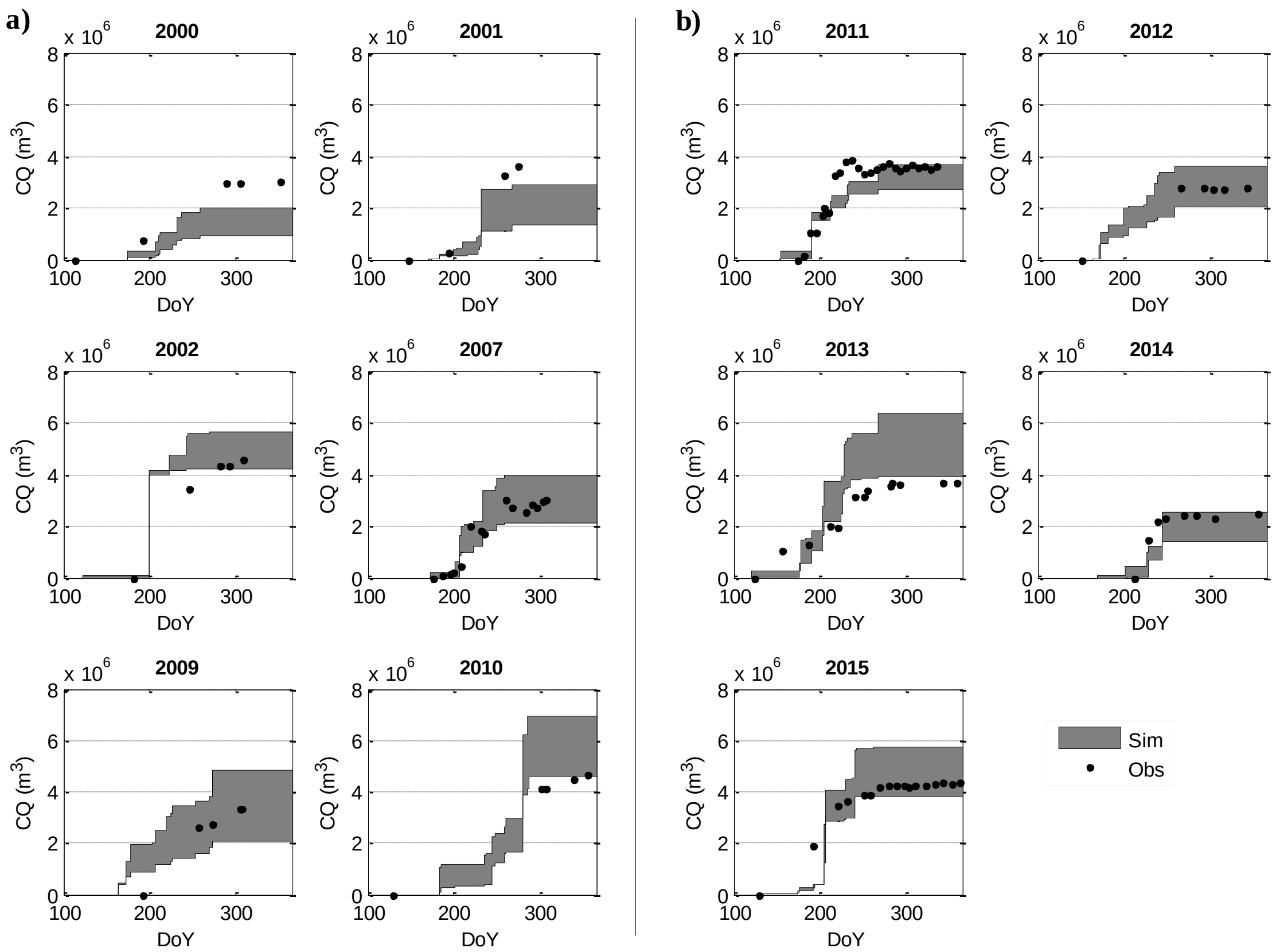

**Fig. 5: Cumulative discharge (CQ) for years with observation data over 2000−2015. For each year, black dots are for the observations and the gray-shaded envelop represents the maximum and minimum of the ten members of the ensemble simulations. (a) is for the validation period and (b) is for the channel set-up period.**

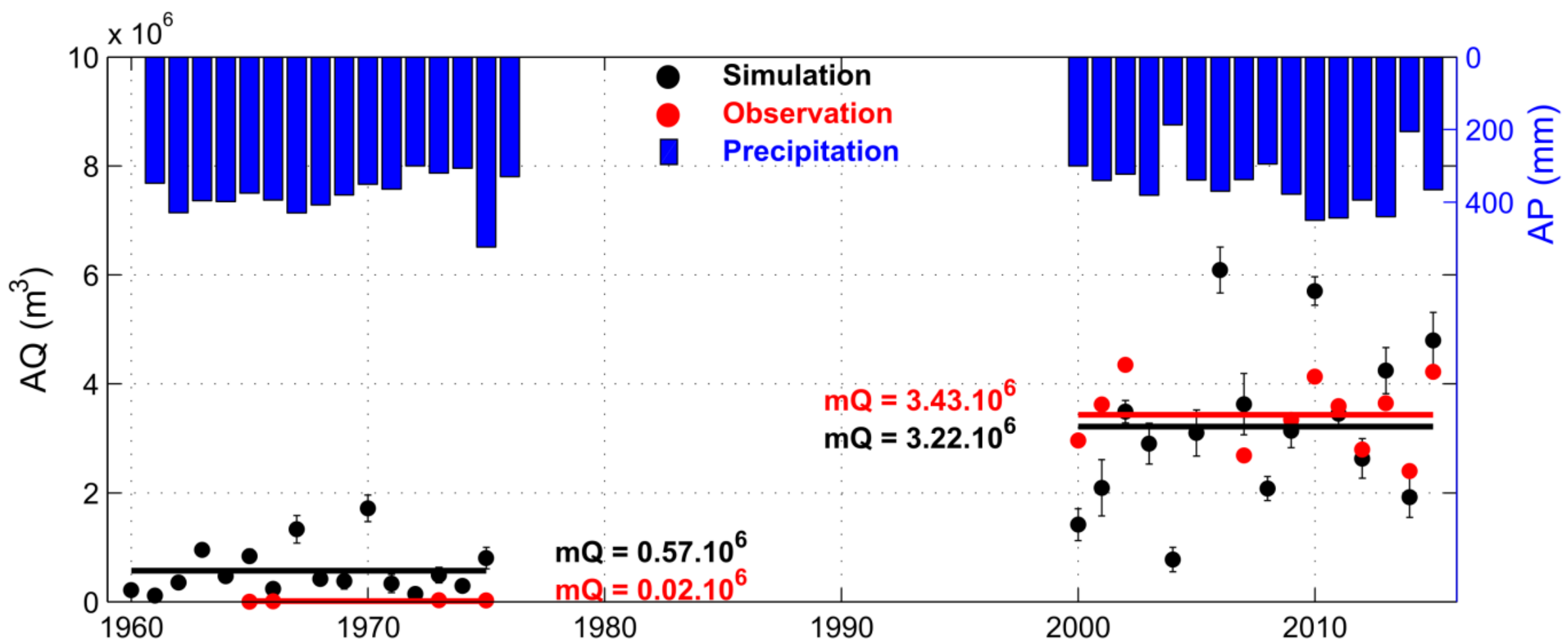

**Fig. 6: Evolution of annual discharge (AQ in m$^3$) between 1960 and 2015: simulations with standard deviation of the ten members (black dots with error bar) and observations (red dots) together with annual precipitation (AP in mm, blue bars).**

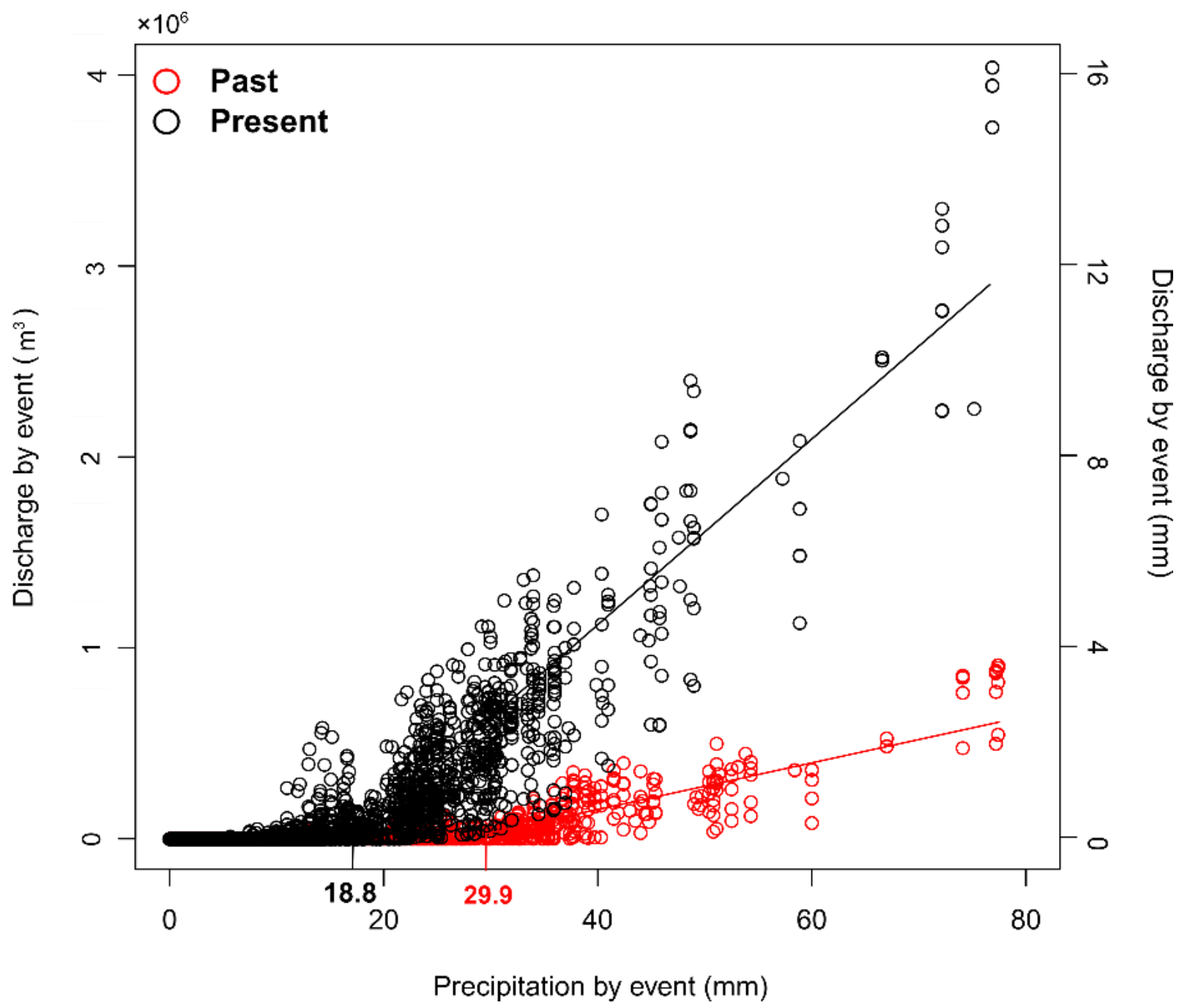

**Fig. 7: Relation between precipitation and discharge for all events in the 15 years period in the past (red points) and the present (black points).**

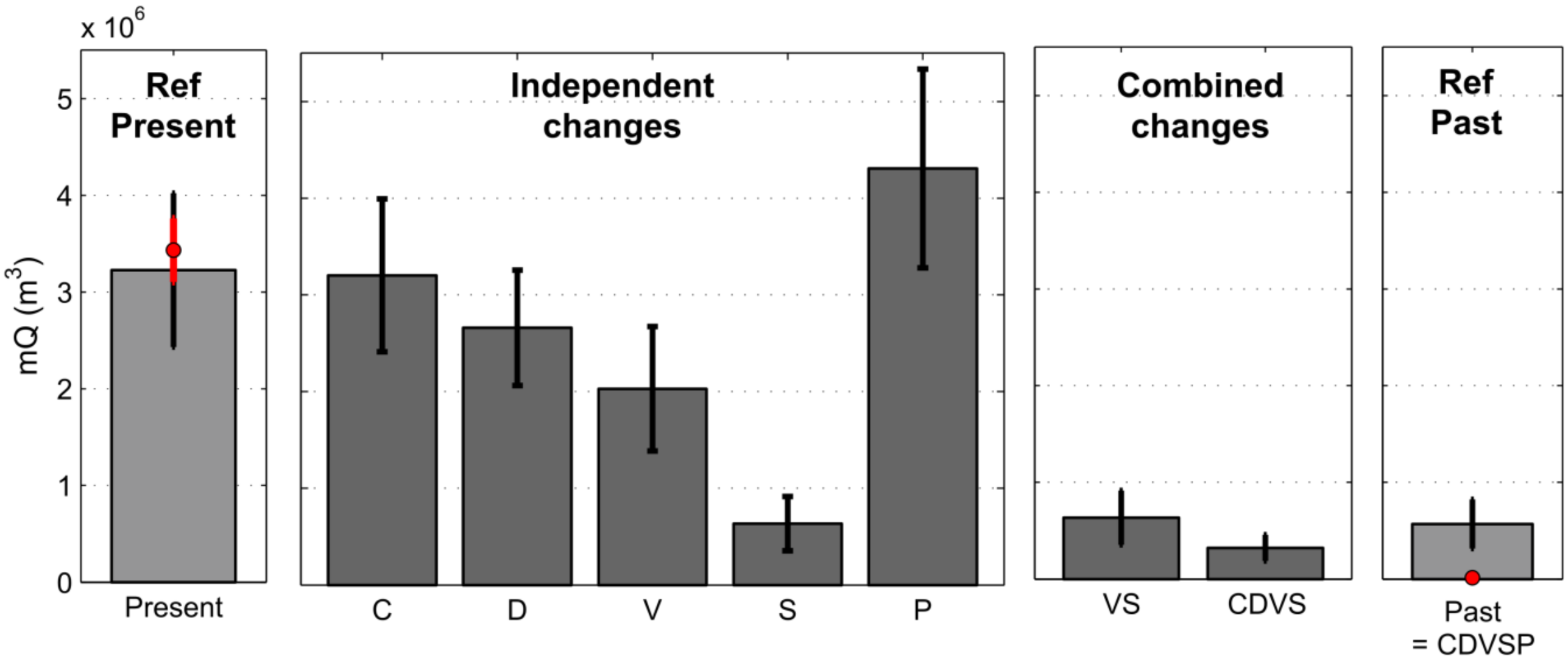

**Fig. 8: Mean discharge (mQ) for present and past reference cases and for the attribution simulations described in Table 5, with either independent or combined factors. For the references cases, observation data are added (red points). Errors bars indicate the standard deviation of the ten member and the fifteen years of simulation.**

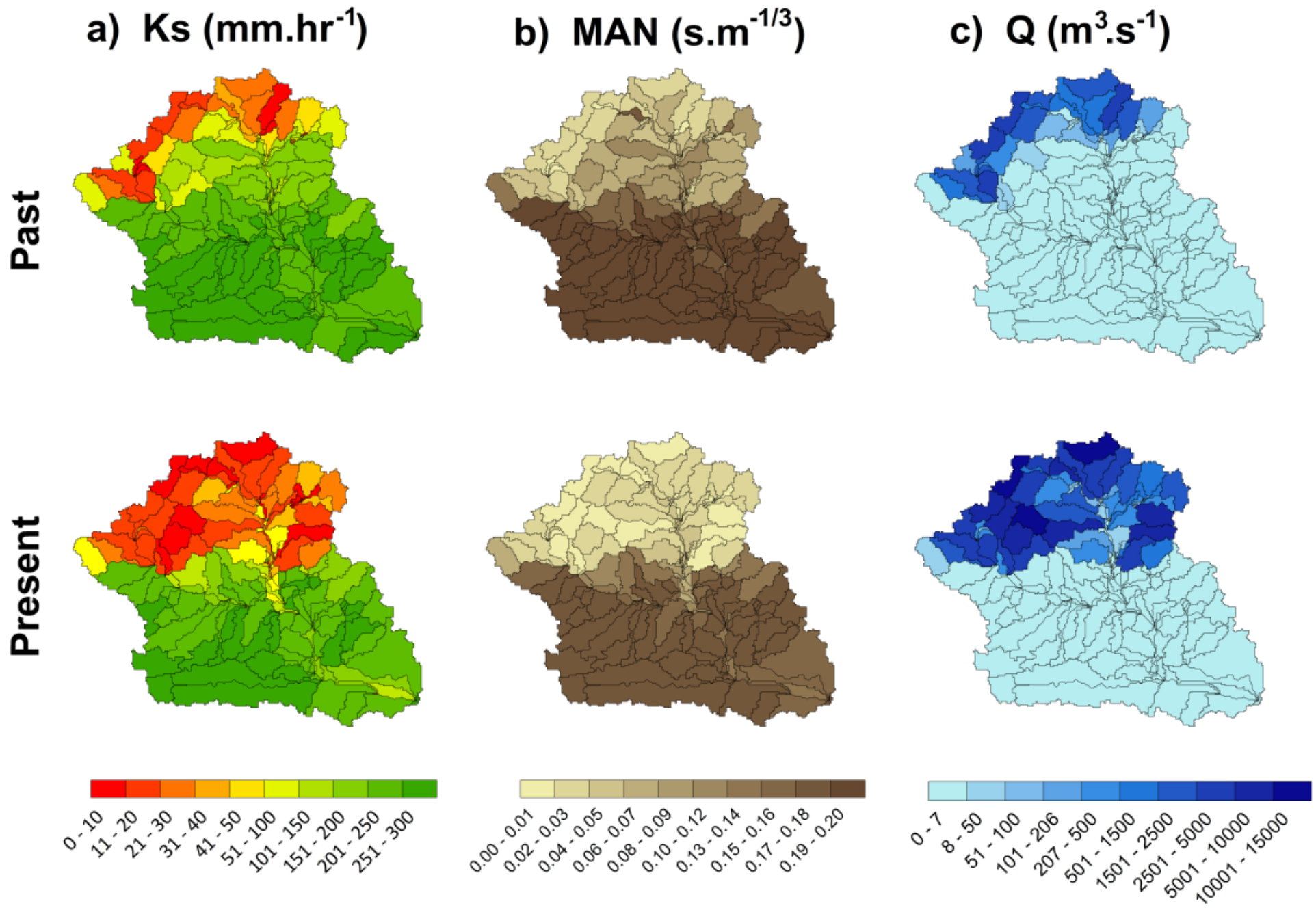

**Fig. 9: Spatial patterns of the (a) saturated hydraulic conductivity (Ks), (b) Manning's roughness coefficient (Man) and (c) Discharge (Q) between the past and the period for the monsoon season (JAS).**