# Peer review of "The paradoxical evolution of runoff in pastoral Sahel: Analysis of the hydrological changes over the Agoufou watershed (Mali) using the KINEROS-2 model."

_Hydrology and Earth System Sciences, 2016_

## Referee Comment (RC1) · J.-E. PATUREL (Referee) · 19 Dec 2016

Gal – Modeling the paradoxical evolution of runoff in pastoral Sahel. The case of the Agoufou watershed, Mali

**General observations**

This paper deals with the Sahelian paradox: despite the decline of the annual precipitation, the Sahelian region is paradoxically subject to an increase in runoff associated with an increase of the runoff coefficient. The causes of this phenomenon, commonly known as the "Sahelian paradox", are not yet clear. Based on an event-based and physical-based hydrological model, Kineros2, the authors model the runoff on a small basin located in the Gourma under the Niger River loop. The model allows them to prioritize the different factors that lead to this paradox. The title of their article is moreover incomplete since this last part of prioritization does not appear in the title.

This question of the Sahelian paradox questioned some researchers but many hypotheses have not been validated.

My first observation concerns the approach: a model is not the reality, a model is only an impoverished image of the reality, even a physical-based model: there is always a process of calibration of parameters to be launched; a model parameter have never a physical meaning. Therefore, we can not rely exclusively on a model, however excellent it may be, to determine all processes involved in rainfall-runoff transformation, and even to prioritize them. Now, this paper gives the impression that the authors seek to validate their knowledge they have of the problematic by means of a model. I preach for my part for incessant back and forth between observation and simulation ... I'm therefore a bit dissatisfied ...

My second observation concerns the very numerous approximations made by the authors: we do not know what are their simulation impacts, because the authors did not discuss the subject. They present mean or median results that ultimately smooth the response of the basin.

However, I congratulate the authors for all the data that they were able to collect and process (it is not simple in these environments) and which was the basis of this work.

**Specific observations**

The study material is very simple: a single watershed, which does not make it possible to give a universal character to the results obtained.

I would appreciate that the authors use at least one other model and compare the results of these different models and compare them to their observations and their knowledge of the terrain.

Pleas, give ranges of uncertainties of your treatments/process

I'm not native-english, so I can not evaluate the quality of the English.

**Technical observations**

Page 3, lines 30 and after : It also means that as a result of important rainfall events, these ponds may be temporarily interconnected for a more or less long period. Is this type of interconnections possible at Agoufou pond?

Page 4, lines 18 and after : the problem of such a model (event-based model) is to fix the initial conditions for each simulation : how do you proceed?

Page 5, "Precipitation and meteorological data…" : you need to more detailed your data. Some analyses are needed

Page 6, "landscape…" : did you discuss of your results with the local population? Did they validate your maps of landscape/.drainage network evolution?

Page 6, "Rainfall…" : astute approach but you have to validate it. That's why I asked before to more analyse your climatic data.

Page 6, lines 35 and after : can we have an idea of how many times you have to widen your intervals?

Page 7, line 4 and after : can't you validate this assumption with the synop station and the stations network of Amma-Catch program?

Page 8, "model calibration…" : why don't use an automatic calibration? Why these intervals : you tell later that some values found in the literature are higher? Can we have the dispersion of your ten simulation for an event?

Page 9, "reference …" : it is a too simplistic assumption which have an impact on your results… Isn't it possible to use "interpolated situtaions"?

Page 9, "soil land…" : give ranges of the uncertainties of your process. How can you say that in 2011, the western area of the basin contribute to the Agofou pond?  In the Inner Delta of Niger, there is the same phenomenon of interconnected lakes during some strong events; these interconnections are not necessarily permanent ant can disappear for a while; It's certainly the same here. How can you be sure that it does not happen before, between 1960 and 1975?

Page 13, line 37 : Pierre at al., 2016 is not referenced

Page 14, line 32 : "erreur…" ????

Page 15, line 34 : what are the "stocking rates"?

---

## Referee Comment (RC2) · Anonymous Referee #2 · 20 Dec 2016

The paper focuses on a very important topic of runoff generation processes and their changes through time and in identifying the main causes of such changes in an area of sparse data. The objectives and the general approach that was taken in the study to achieve these objectives are scientifically sound and a good and insightful data analysis is presented.

I have however main concerns of the modeling strategy, assumptions and application. They include: a large gap between the model complexity and the data used for its application; sensitivity analyses are essential to justify many of the modeling decisions made, such as which model parameters to calibrate (leaving out probably the most sensitive parameters); the uniform rainfall assumption is very problematic and should be justified; and, model calibration was made against a single number of mean annual runoff volume (although annual volumes estimation do available). See more details below:

1) The authors rightly present the need of high temporal resolution rainfall and apply a disaggregation procedure. However, in space they assume a uniform distribution of rainfall over the catchment. This assumption is very problematic. Even if the storm cell is more or less at the same size of the basin or somewhat larger, the cell location in space will be often thus that only a partial coverage is achieved. Given the very high sensitivity of runoff to the partial coverage the exclusion of this factor from analysis might add a large uncertainty. Sensitivity analysis of this of catchment runoff to partial storm coverage should at least be examined.

2) The change in the channel network density between the two periods was presented by a change in planes aspect ratio rather than the CSA elements. Altering the aspect ratio generates a more or a less elongated catchment shape but the drainage density is changed only a little. Why not to utilize the derived channel network maps and identify for each period its own CSA configuration?

3) Kineros2 has many parameters. The authors have chosen to calibrate the channel Ks and Manning coefficient, while the other parameters were determined using from data and using different functions. This is a very problematic decision – why these parameters were selected for calibration? What do we know about the accuracy of the other model parameters that are not calibrated? There are two necessary steps that are essential to justify the authors decision: 1) sensitivity analysis that will show the parameters that are most important for calibration (i.e., that model output most sensitive to them), 2) for the pre-defined parameters, assess their uncertainty and examine how this is translated into small uncertainty in model output.

4) I believe that total runoff is much more sensitive parameters associated with the infiltration process in the plans, e.g., to plans Ks rather than to channel Manning coefficient and probably to channel Ks. The decision to calibrate the two channel parameters, MAN and Ks is not clear and must be supported.

5) Furthermore, a main impact on annual runoff was found to be the modification of soil properties and vegetation cover, but the model parameters associated with the hydraulic properties of these units were not determined in such a way we have a high certainty in those parameters. Obviously, modeled runoff is very sensitive to these parameters, but they were not calibrated or even examined for their sensitivity.

6) The calibration strategy seems to me not appropriate. The authors use the bias of the annual runoff as the objective function for calibration; however, Bias does not account for the year by year variations

but integrates all year data into a single value, so possibly a large overestimation of modeled runoff in one year can be compensated by a large underestimation in another year. Instead, an objective function that accounts for the yearly residuals, such as the most popular RMSD objective function is much preferred. It should be emphasized that using the Bias ignores the annual runoff values estimated in the authors previous work and just uses their integration.

7) Furthermore, a calibration strategy that is based on Bias of runoff volumes, implies that a single number is used for calibration (one data!). This seems not reasonable given the very complex model used and the hard work done to produce very high resolution data.

8) The authors utilize a very detailed and high resolution hydrological model (which I am not sure is the most appropriate given the very limited data they have), but they do not really take advantage of the detailed simulations. For example, they could try to understand why the change of soil and vegetation properties increased runoff, for which type of rain events it is more pronounced? Are the change manifested in higher peak discharge or in more streamflow events, etc.

9) As rainfall is so highly variable, conclusions about the effect of its possible change should be done with a caution. For example, the authors state that "The results show that changes in daily precipitation regime do not explain runoff changes between the past and the present." (P. 15, L. 31), but even if such changes do occur in reality it is most likely that they are not statistically detectable due to the high natural variability.

Specific comments

1) A climatic description of the area is lacking: mean annual rainfall, potential/actual ET, etc.

2) Only one station is used for rainfall data (and few others are used for the temporal disaggregation); clearly a poor coverage of the catchment, as seen in Figure 1. Have the authors examined the option of remotely sensed precipitation? At least to examine the storm coverage area (which is assumed here to fully cover the catchment.

3) Assumption of soil recovers its initial conditions in two days – this assumption can be reasonable for arid regions. What about the deep soils in the south?

4) Add the "absolute value" sign to Eq. 3

5) The optimization of MAN and KS should be at a higher resolution in my opinion

6) Please clarify how did you identify "Isolated dunes (S1) are found at the same location for both periods, but have been eroded and partially encrusted" (P. 10).

7) Please represent the RMSE value also in percent from the mean (P. 10 L. 22).

8) I recommend to present runoff ratios for each year and to show an example of flood hydrograph.

9) As rainfall is so highly variable, conclusions about the effect of its possible change should be done with a caution. For example, the authors state that "The results show that changes in daily precipitation regime do not explain runoff changes between the past and the present." (P. 15, L. 31), but even if such changes do occur in reality it is most likely that they are not statistically detectable due to the high natural variability.

---

## Referee Comment (RC3) · Anonymous Referee #3 · 20 Dec 2016

**General comments**

This manuscript presents a modelling exercise made for investigating the causes for the so-called 'Sahelian paradox' that consists of a runoff increase in the last decades after the catastrophical drought of the seventies, in spite of a decrease in annual precipitation. The subject is of interest for the HESS readers, uses a known rainfall-runoff and erosion model as well as rainfall input data derived from networks and new information on land cover, soil types and catchment runoff derived from remote sensing and photointerpretation. The paper is mostly well written and its overall formal quality is good.

Nevertheless, the manuscript handles the issues related to temporal and spatial scales as well as parameterisation in too simplistic ways for the results being sufficiently sound to 'explain' or to 'understand' the Sahelian paradox. Using a 5-minute step event model designed for small agricultural catchments for simulating the annual discharges of a 250 $Km^2$ basin, assuming that the relationship between daily and 5-minute rainfall intensity did not change on time is not a conventional research approach. More essentially, as it is well known among the hydrological community that any hydrological model can give good results for the wrong reasons, when the purpose is not to obtain good results but to investigate the reasons, the researcher must be particularly cautious to take into account the likely equifinality of diverse possible parameterisations.

The paper might be accepted for publication if both the model parameters and results were more investigated. The analysis of the contribution of the diverse factors to the change of the catchment response is a strength of the paper, but it assumes just a 'correct' parameterisation; an uncertainty analysis should be made or, at least, a sensitivity analysis of the model response to parameter variation. Annual catchment discharges should not be the unique focus of model output, but some model results at the event scale (extreme events, annual number of runoff producing events, rainfall threshold values...) and at the landscape unit scale (identification of contribution areas for diverse type of events...) should also be shown and discussed. Yet, the authors must make a rigorous distinction between model and reality, simulations and observations, as well as more clearly separate drivers, processes and model parameters.

**Detailed comments:**

Page 1, line 16: KINEROS is not a water balance model but a rainfall-excess runoff model. Water balance, which is the main challenge in the Sahelian paradox, is therefore indirectly simulated, as runoff water is subtracted from infiltration and subsequent evapotranspiration. This is a relevant aspect to be stated in order to assist the understanding of the paper by readers not familiar with KINEROS.

Page 1, line 17: it is necessary to state the catchment area

Page 1, line 23: "shallow soil being eroded and giving place to impervious soils" please rewrite in a less literary way

Page 1, line 24: the converse is more rigorous: "The KINEROS-2 model was parameterized in order to simulate these changes in combination or independently".

Page 1, line 26: Catchment flows shown in volume ($m^3$) cannot be related to rainfall rate and this is not usual in the hydrological literature because volume depends on catchment area, it is more adequate to show them in runoff units (mm per year). Showing only the simulated results is not informative at all (are those simulated with KINEROS2?).

Page 1, line 29: "Modification" refers to the action of changing something and should not be used for a natural change. This is unclear that vegetation cover was modified in the parameterisation. Please describe more precisely the modification of parameters made for simulating the landscape changes.

Page 2, Line5: During or after the drought?

Page 2, line 25: Reduction of vegetation cover and topsoil crusting are factors of too different nature to be cited together. Reduction of vegetation cover can (directly) decrease rainfall interception and plant transpiration, or decrease the sol protection against raindrop energy, so (indirectly) favour soil crusting. Soil crusting usually decreases infiltration rates and favours rainfall-excess overland flow. Please, describe more precisely the drivers and mechanisms that have been pointed out for explanation of the 'Sahelian paradox'.

Page 4, line 2: "and runoff is frequently generated over them."

Page 4, line 21: KINEROS was not designed explicitly for arid and semiarid areas.

Page5, lines 18-22. These are not soil and land cover data but just images.

Page 6, line 13 (and below): "Erosion surface" seems to refer to a geomorphic unit (land form), but here is not a good denomination for a landscape unit because it is not clear to most readers and it is equivocal with the soil erosion processes. "Pediment" is a geomorphic English term equivalent to the French "glacis" term that could be used instead.

Page 6, line 30 and subsequent: This is one of the main weak points of the paper, as the method used assumes that there is no change in the fine temporal structure of rainfall events. In the lack of data to improve the approach, some sensitivity exercise should be made to test the role of changing this structure on runoff generation. This may be made using a range of 'ensemble' 5-minutes series with higher, average and lower 5-minute intensity within reasonable bounds.

Page 7, line 19 and subsequent: In fact, there is a terminological confusion in the paper respect to the changes in the drainage network: the changes observed are really changes in the stream channel network; in the old period runoff was too slight or infrequent in the thalwegs to form distinguishable channels that were cut after the drought period (see another comment below). The subsequent paragraph describes how the DEM derived drainage network was adapted to the network observed in 2011, but not clearly how the 'old' network was parameterized.

Page 7, line 34: Thickets

Page 8, line 1: Low LAI is a reason but high winds favour the evaporation of intercepted rainfall. Check the rainfall interception literature in semiarid areas (e.g. Llorens & Domingo Journal of Hydrology, 2007)

 Page 9, line 22 and subsequent: In the writing of the following paragraphs there is sometimes confusion between the changes of the extension of the mapped landscape units, the changes of the properties of these units and the related hydrological processes.

Page 10, line 9 and subsequent: Please, change the "Erosion surface" term.

Page 10, line 11: "an important erosion of the underlying soil has occurred": do you have evidences of this phenomenon? Where are the eroded soils deposited? "Impervious bare soils have replaced most of these areas": this is not a rigorous description of a landscape change. In all this paragraph there is confusion between changes in the map units, the characteristics of these units and the processes related to these changes (as causes or consequences).

Page 10, line 15: The development of a drainage network (in fact this seems to mean that new channels are observed in previously unchannelled thalwegs) may be attributed to the increase of overland flow, but not necessarily to the change from sheet runoff to concentrated runoff on the hillslopes, unless new rills and shallow gullies are observed throughout. The entrenchment of channels in semiarid conditions has been attributed to increased runoff or the decay of valley bottom vegetation (e.g. Nogueras et al. Catena 2000 and cited references).

Page 11 line 5 and subsequent. This sub-section is not well written. The parameterisation of the channels is not a result. Please, describe changes in precipitation before changes in discharge and use a chronological order of the periods when possible. Reporting discharges in volume is really difficult to follow, please use runoff units (mm/year). Please, state observations before simulations throughout.

Page 11, lines 16-18: this paragraph is unnecessary unless the behaviour of the catchment is better described, as proposed above.

Page 11, line 24: Please, include a sentence recalling that the reference run is the recent period and that changes are evaluated using equation (4).

Page 11, line 26 and subsequent: "... present characteristics except dune crusting ...". "... has two effects on the parameterisation of the land surface ...". "... dune crust on the simulated annual discharge...". Reporting volumes for the sub-basins is confusing, please report percentages of the total runoff and clearly state that these are simulated values for the recent/older periods.

Page 12, lines 3 and subsequent: Please, state the (indirect) effect of the vegetation changes on model parameters, this is to help understanding the runoff slowing and infiltration increase. Please, report discharge in mm.

Page 12, lines 14 and subsequent: "Modification" and "erosion surfaces" are not appropriate terms here, as discussed above. The "increase in erosion surfaces" is contradictory respect to the small changes in these units as described in Page 10 line 9. Here there is confusion between soil properties and landscape units, please be more explicit.

Page 12, lines 20 and subsequent: " the result is an increase of xxx mm/year of the discharge for the past period..."

Page 12, line 28: mind the confusion between soil and landscape unit

Page 14, line 36: "... and soil properties may largely explain..."

Page 15, lines 3-4: "The lack...concentrated runoff": There is a melange of causes and consequences, yet, the change from sheet to concentrated runoff must be demonstrated.

Page 15, lines 13-14: See the note above on channel entrenchment.

Page 15, line 14 and subsequent: "Our work has shown that enhanced and concentrated runoff results in an increase in both the number and the length of channels, therefore increasing the drainage density and diminishing the travel time for water to reach the drainage network" : This is not shown in the results above.

Page 15, line 19: "Our results suggest that..."

Page 15, line 28: "are simulated as part of vegetation..."

Page 15, line 33: "surface runoff is observed and simulated to decrease..."

Page 15, lines 36-27: a preliminary test should be made changing the fine temporal structure of rainfall, as suggested above.

Conclusions: this section should be rewritten after the revision of the manuscript, but it is important to bear in mind that in this case the model approach may be useful to "investigate" or to "shed some light on" the paradoxical evolution in the Sahel, but not to "understand" it.

---

## Referee Comment (RC4) · Anonymous Referee #4 · 27 Dec 2016

The paper deals with the "Sahelian paradox" phenomenon where despite a decrease of the precipitation in the Sahel during the last 50 years, an increase of runoff was observed. The Agoufou catchment (245 km$^2$) is taken as a study case, and the spatially distributed model KINEROS2 used to prioritize the different factors (dune crusting, drainage network development, vegetation changes, modification of soil properties, or a combination of some of these factors) that lead to this paradox.

My first feeling is that the title of the paper does not reflect its real content because the paper remains an application of a model on a catchment. Neither the catchment nor the model was chosen to demonstrate a hypothesis related to the "Sahelian paradox". Moreover the use of the model to prioritize the factors causing the increase of runoff remains a numerical modelling exercise without any validation using hydrological data. The model can give good results for the bad reasons. Consequently the conclusions of

the paper on the main factors causing the "Sahelian paradox" may be correct, but may also be not correct.

My comments concern:

i) The choice of the studied basin: The paper doesn't justify the basin choice in comparison to other catchments in the same region. Does the land use change and the consequences on the "Sahelian Paradox" observed on the neighboring catchments. In order to demonstrate (or not) the hypotheses and quantify the role of the different factors which lead to the "Sahelian Paradox", it would be preferable to choose the catchments with internal stream-gauges and piezometers. An increase of runoff will be accompanied with a modification of the other hydrological processes especially the water table level and extension, and evapotranspiration. However, the paper doesn't deal with these two main hydrological processes due to the lack of data. Consequently, the available data are not sufficient to validate or not a given hypothesis.

ii) What we learn from data: The paper doesn't present a detailed analysis of the spatio-temporal data and do not discuss the evolution of the components of the water balance. The authors must first discuss what we learn from the data only, and in a second time what is the added value using the model.

iii) The available data: The main problem is that only "annual" water outflow is available, reconstructed by the author for some years (see Table 1)! Moreover, one rain-gauge is available on the catchment, and data at a fine time step (5 min) are only available for given periods. The lack of analysis of the spatio-temporal structure of rainfall at 5-min time step, and the use of an empirical method for temporal disaggregation is a weak point of the study. Moreover, the paper limits the analysis at the annual water balance and no information is given on flood events characteristics on 5-min time step: evolution from the 50th until now of the rainfall intensities, runoff coefficient, peakflows, lag time, etc. The data are not coherent: a fine DEM resolution (30m) vs annual flow and daily rainfall! I'm not sure that the available hydrological data are sufficient to give

responses to the important questions raised in the introduction!

iv) The uncertainty on data: The authors must discuss the uncertainty on the hydrological data (e.g. spatial distribution of rainfall, the annual runoff reconstructed) and the consequences on modelling results.

v) The choice of the hydrological model: The spatially distributed model KINEROS2 is used without any justification. I'm not sure that this model is the most appropriate for the available data (only one rain-gauge, and total annual runoff). The paper doesn't demonstrate that the prioritization of factors causing the "Sahelian paradox" are independent of the model choice. Comparing different models will give more arguments for the discussion.

vi) The calibration procedure: an important number of the model parameters were arbitrarily fixed and only two parameters were calibrated. The values of the calibrated parameters will depend on the values chosen for the fixed ones. The authors must justify the choice of the parameters to be calibrated, and discuss how a modification of the fixed parameters will impact the conclusions of the paper.

vii) The criteria function: Only the "Bias" (Eq 3), at the annual time step, was chosen as a criteria function. The paper doesn't present any simulated hydrographs, neither other values of the criteria function (especially criteria related to peakflows) in order to evaluate the performance of the model. Different criteria functions must be used.

viii) In order to study the "Sahelian paradox", it will be interesting to compare the components of the water balance on a large number of basins (and more especially embedded ones). Before undertaken a modelling exercise, an analysis of data is necessary in order to link (or not) the evolution of "hydrological" processes to the evolution of land use.

Other comments:

- Abstract: please indicate the catchment area in the abstract.

- P3, L11-15: The objectives of the paper are reduced to an application of a model on a given basin, and don't give responses to the main question related to the "Sahelian paradox".

- The title of the paper must be in accordance with the objectives announced.

- Section 2.1 "Study site": The hydrological data 'rainfall, runoff) and the spatial data (land use, soil, geology, DEM, etc.) must be presented in this section and nor as input to the KINEROS2 model. The authors must discuss what we learn from data before undertaking a modelling exercise.

- What is the uncertainty on the delimitation of units on maps (from Table 2) and consequently on the area of each class of land use in space and in time (Table 6)?

- How the drainage network was defined on Figures 3 and 4? How the channel network was interpolated in time between 1956 and 2011?

- The Manning coefficient MAN has a unit (($m^{1/3}$ / s)

- Table 3: How these parameters were fixed? What is the sensitivity of the mode l results of these values are taken different?

- Figure 5: The grid used must be refined?

---

## Referee Comment (RC5) · Anonymous Referee #5 · 3 Jan 2017

**Referee comment on "Modeling the paradoxical evolution of runoff in pastoral Sahel. The case of the Agoufou watershed, Mali" by Laetitia Gal et al. (Hydrol. Earth Syst. Sci. Discuss., doi:10.5194/hess-2016-623, 2016.)**

This reviewer largely agrees with many of the comments already expressed by reviewers RC2 and RC4. Given the numerous issues expressed I feel the paper should be reframed and possibility retitled along the lines expressed by reviewer RC3 whose last suggestion was "Conclusions: this section should be rewritten after the revision of the manuscript, but it is important to bear in mind that in this case the model approach may be useful to "investigate" or to "shed some light on" the paradoxical evolution in the Sahel, but not to "understand" it." A new title might be something like "Exploration of the paradoxical evolution of runoff in the pastoral Sahel – Agoufou Watershed using available data and a watershed model."

The author could then stress that the model selected could be one of many for this investigation, but K2 was selected for x, y, and z reasons as a tool to investigate possible reasons for the paradoxical evolution of runoff in the Agoufou watershed. Within the constructs of the model, its structure, and the assumptions inherent in the model it was felt it could be used to conduct a relative ranking of various factors and watershed attribute changes contributing to the paradox. Using other models one might come to different conclusions or attributions but the authors could encourage others to conduct comparable "detective" investigations to better understand factor contributing to the paradox.

As pointed out by the other reviewers the uniform precipitation assumption for a basin this size constitutes a major simplification and calls into question the ability to carry our a defensible model calibration and validation. Al-Qurashi et al. (2008) applied K2 to a 734 km$^2$ arid watershed with 7 rain gauges where a "parameter set which gave best calibration performance over any combination of 26 events did not generally produce acceptable performance (defined as within 30% of observed) when used to predict the 27$^{th}$ event". In this and similar situations, the authors noted that "data sets typically used for distributed (or semi-distributed) rainfall-runoff modeling in arid regions cannot provide an accuracy which justifies the effort and expense of this (K2) modeling approach. The limitations imposed by relatively sparse observations of rainfall are of particular concern" (Al-Qurashi et al., 2008, p. 104).

To remove this major limitation and use K2 as a tool to explore causes of the runoff increase this reviewer suggests taking a relative change approach as advanced by Goodrich et al. (2012) and Sidman et al. (2015) for post-fire watershed assessment in watersheds that often do not have any rainfall-runoff data available for calibration and validation. In this approach a pre-fire land cover map is used to parameterize the watershed and conduct a simulation with a spatially uniform design storm. The burn severity map is then used to alter model parameters based

on prior research and analysis of the effects of burns on cover and soil hydraulic properties. A second post-fire simulation is then conducted using the same rain storm. The results can then be spatially differenced to analyze changes. For the author's study the present and past model parameterizations based on analysis of historic and current land cover and soils data are analogous to the pre- and post-fire conditions. The authors could pick one of their most trusted rainfall data sets (perhaps when they had high temporal resolution measurements) and use that rainfall input data set for both the past and present watershed model parameterizations. Given one of their conclusions (last paragraph) was that climatic and precipitation changes from past to present appeared to little or not impact on the findings this would further justify the approach noted above. By doing so the authors would isolate the analysis on watershed changes as they would be using identical input drivers. This would still be directly in line with the stated objectives of their study.

Technical Comments:

The authors have confused the meaning of the CSA or contributing source area. This is the drainage area required it initiate the head of a first order channel and effectively defines the level of geometric complexity of the watershed with a smaller CSA (percent of drainage area or absolute area) resulting in more watershed modeling elements. The channel source area modeling elements are those that drain to the head of a first order element. The remaining upland or hillslope modeling elements (planes – they can be curvilinear as well) contribute laterally to channel modeling elements.

Regarding the questions by other reviewers of K2 model sensitivity the author's should review and cite Yatheendradas et al. (2008) who conducted a thorough analysis of variance. In their analysis, model prediction uncertainties are dominated by precipitation input uncertainties (another reason in suggesting the approach noted above). For K2 model parameters a multiplier on the Ksat of overland flow model elements and the Manning's roughness multiplier on overland flow model elements were the most sensitive parameters while the channel roughness multiplier was also quite important.

Given this information it is odd that the authors selected the Ksat of the channels and not of the overland flow planes for calibration. Note that Ksat of channels and Ksat of hillslope elements do interact. The relatively low calibrated Ksat channel value that the authors found could easily be the result of a higher Ksat on the hillslope elements resulting in lower lateral runoff inflow into the channels.

Given the author's finding of the importance in the change drainage density and channel characteristics two items are suggested:
1.   Did the author's use the default values for channel cross-sectional geometry? If so these value were derived from regressions relating X-S

measurements to easily derived variables from GIS operation on watershed data as obtained at the Walnut Gulch Experimental Watershed in SE Arizona, USA (Miller et al., 2000).  The Walnut Gulch relationships are likely to be a poor representation of the channel cross sectional characteristics of the Agoufou watershed.  It is suggested that the authors gather some field measurements from the Agoufou watershed.  At least from several stream orders so they might be scaled across all the channels in the study watershed.

2.  Instead of altering the aspect ratio of the overland flow (plane) hillslope elements a watershed discretization can be derived from for each (past and present) channel network (contact Shea Burns for details).

*References*

Al-Qurashi, A., McIntyre, N., Wheater, H., Unkrich, C.L. 2008. Application of the Kineros2 rainfall-runoff model to an arid catchment in Oman. J. of Hydrology. 355:91-105.

Goodrich, D.C., Burns, I.S., Unkrich, C.L., Semmens, D., Guertin, D.P., Hernandez, M., Yatheendradas, S., Kennedy, J.R., Levick, L. 2012. KINEROS2/AGWA: Model Use, Calibration, and Validation. Transactions of the ASABE. 55(4): 1561-1574.

Miller, S.N., Youberg, A., Guertin, D.P., Goodrich, D.C. 2000. Channel morphology investigations using geographic information systems and field research. Proc. Land Stewardship in the 21st Century: The Contributions of Watershed Manage., USDA Forest Service RMRS-P-13, pp. 415-419.

Sidman, G., Guertin, D.P., Goodrich, D.C., Unkrich, C.L., Burns, I.S. 2015. Risk-assessment of post-wildfire hydrological response in semi-arid basins: The effects of varying rainfall representations in the KINEROS2/AGWA model. International Journal of Wildland Fire. 1-11.

Yatheendradas, S., Wagener, T., Gupta, H., Unkrich, C.L., Goodrich, D.C., Schaffner, M., Stewart, A. 2008. Understanding uncertainty in distributed flash flood forecasting for semiarid regions. Water Resources Research, Vol. 44, W05S19.

---

## Editor Comment (EC1) · M. Sivapalan (Editor) · 3 Jan 2017

There have been 5 review comments already. The last review comment does the extra job of synthesizing the comments of the previous reviewers.

The authors should respond to each of the review comments separately - if questions or concerns are repeated by the reviewers you only need to respond once.

In any case there should be a rebuttal of the review comments, to allow the reviewers to respond as well - this is the HESS discussion process.

At the end of this discussion process, I will come back and summarize my recommendations to the authors, including the decision to publish, decline or accept with minor or major revisions.

---

## Author Comment (AC1) · 17 Feb 2017

**RC#1**

**Gal – Modeling the paradoxical evolution of runoff in pastoral Sahel. The case of the Agoufou watershed, Mali**

> We thank reviewer 1 for reviewing our manuscript and providing valuable feedbacks. We have addressed all of his comments and discuss them below.

**General observations**

This paper deals with the Sahelian paradox: despite the decline of the annual precipitation, the Sahelian region is paradoxically subject to an increase in runoff associated with an increase of the runoff coefficient. The causes of this phenomenon, commonly known as the "Sahelian paradox", are not yet clear. Based on an event-based and physical-based hydrological model, Kineros2, the authors model the runoff on a small basin located in the Gourma under the Niger River loop. The model allows them to prioritize the different factors that lead to this paradox. The title of their article is moreover incomplete since this last part of prioritization does not appear in the title.

> Thank you for the suggestion
>
> We will change the title accordingly (also suggested by other reviewers).

This question of the Sahelian paradox questioned some researchers but many hypotheses have not been validated.

My first observation concerns the approach: a model is not the reality, a model is only an impoverished image of the reality, even a physical-based model: there is always a process of calibration of parameters to be launched; a model parameter have never a physical meaning. Therefore, we can not rely exclusively on a model, however excellent it may be, to determine all processes involved in rainfall-runoff transformation, and even to prioritize them. Now, this paper gives the impression that the authors seek to validate their knowledge they have of the problematic by means of a model. I preach for my part for incessant back and forth between observation and simulation ... I'm therefore a bit dissatisfied ...

> We agree, a model is not the reality. We will change the manuscript to make clear that explanations are "according to the model" and that the results are subject to uncertainties. We believe that this study, even if model-based, has shed new lights on the Sahelian paradox phenomenon and that it provides an important contribution to the debate on the man-made versus natural causes.

My second observation concerns the very numerous approximations made by the authors: we do not know what are their simulation impacts, because the authors did not discuss the subject. They present mean or median results that ultimately smooth the response of the basin.

> We have added 2 figures to document modeling result in more details (maps of MAN, Ks, runoff over the watershed, for the Past and Present cases, as well as Precipitation / Runoff plots for the Past and present cases) which illustrate simulation results and add spatial and temporal information (see specific observations section for the Figures).
>
> There is one point that deserves more detailed explanations: the need to lump events and to look at runoff in a statistical way. It is directly linked to the temporal disaggregation of rainfall. We have physical reasons to use a short time step (namely the importance of Hortonian runoff). For each daily precipitation amount, we use ten events with a 5-min resolution. Since they are taken from a 5-min look-up-table, it ensures that on average, we have a good distribution of 5-min intensities (see the figure included on the response to Technical Comment p. 6, below). At the scale of a single event, though, we have no guarantees that the 5-min intensities correspond to the reality. For this reason, we look at annual means and 15 years averages (which are based on a large number of events, so that the statistical distribution makes sense). We also show the variability induced by temporal disaggregation and seasonal results, as an illustration (grey envelops in Figure 6). We will explain that in more detail in the revised manuscript.

However, I congratulate the authors for all the data that they were able to collect and process (it is not

simple in these environments) and which was the basis of this work.

Thank you for this remark. We agree that important scientific questions arise in less observed areas.

**Specific observations**

The study material is very simple: a single watershed, which does not make it possible to give a universal character to the results obtained.

Due to the limited data availability, no studies of this kind have been carried out up to now in pastoral areas of the Sahel, which makes our study original. The Agoufou watershed is a great case study given the unique long-term environmental monitoring, thanks to the AMMA-CATCH observatory and older programs, starting in the 1980s-90s (Boudet, 1972; Hiernaux and Turner, 1996). The watershed displays a spectacular increase in runoff, which has been quantified in a previous study (see Gal et al., 2016).

In addition, the strong evolution of surface water observed at Agoufou has also been observed in the Gourma region (91 lakes, Gardelle et al, 2010) and elsewhere in the Sahel (Niger and Mauritania, see Gal et al., 2016).

We believe that the mechanisms highlighted here for the Agoufou basin may be at play in other regions of pastoral Sahel. Moreover we cannot exclude that these mechanisms may also play a role in other areas where land use change was considered the major cause for the observed hydrological changes. This of course calls for additional studies.

We will explain in the revised section "study area" the reasons for this choice, which also responds to comments by other reviewers

Boudet, G., 1972. Désertification de l'Afrique tropicale sèche. Adansonia 12, 505–524.

Hiernaux, P., Turner, M.D., 1996. The effect of clipping on growth and nutrient uptake of Sahelian annual rangelands. J. Appl. Ecol. 33, 387–399. doi:10.2307/2404760

I would appreciate that the authors use at least one other model and compare the results of these different models and compare them to their observations and their knowledge of the terrain.

A previous study, not detailed here (found in Gal L., 2016, "Modélisation de l'évolution paradoxale de l'hydrologie Sahélienne. Application au basin d'Agoufou", PhD thesis, Université de Toulouse) based on a literature review was carried out with 20 different hydrological models (global, distributed and semi-distributed) in order to select the model best suited to the zone and the objectives of the study. KINEROS2 was found to be suited for the study purpose.

In addition a model/data intercomparison project, called ALMIP2 for AMMA Land Surface Model Intercomparison phase 2, has been carried out in this area to assess the capability of land surface models (LSMs), vegetation models and hydrological models to describe hydrological processes in this area: 20 different LSMs were analyzed (Grippa et al, in press in JHM, available as early release on line at http://journals.ametsoc.org/doi/pdf/10.1175/JHM-D-16-0170.1 or upon request to M. Grippa). The results highlight the difficulty of models to distinguish between shallow or silty soils generating the runoff ending up in ponds and no-runoff areas, like the sandy deeper soils, which infiltrate all rainfall. LSMs have been found to be too sensitive to rain and not enough to soil properties. We hope that our results will stimulate the scientific community to undertake further studies in different basins and with different models to validate or invalidate our findings. The data are being put on the AMMA-CATCH database in that purpose.

Please, give ranges of uncertainties of your treatments/process

For observational data, the uncertainties are detailed in Gal et al. (2016) and will be recalled in the revised manuscript.

For the uncertainties on the planes parameters, we will provide additional results in the revised

manuscript. A sensitivity study has been carried out to highlight the robustness of the model in ranking the factors responsible for the increase of surface runoff. To that end, Ks of all planes was multiplied by 2.5, and MAN by 1.75. This corresponds to the interval given by Casenave and Valentin (1989) for many Sahelian soils. Both changes (Ks and MAN) tend to decrease runoff, therefore the combination of KS and MAN decrease total runoff by a factor of 3. The ranking of the different factors is however the same as with the original planes parameter. This test illustrates the robustness of the results.

[Figure]

I'm not native-english, so I cannot evaluate the quality of the English.

**Technical observations**

Page 3, lines 30 and after : It also means that as a result of important rainfall events, these ponds may be temporarily interconnected for a more or less long period. Is this type of interconnections possible at Agoufou pond?

According to the satellite images and regular field survey, no visible connection between the Agoufou lake and the eastern pond has been observed during the whole study period. However, with the dramatic increase in the surface water and precipitation recovery, it is possible that in the future these two lakes will be connected.

Page 4, lines 18 and after: the problem of such a model (event-based model) is to fix the initial conditions for each simulation: how do you proceed?

This is explained later in the article (page 7 line 1 and 30). The initial properties of the soil have to be prescribed to run KINEROS2. We have calculated the time required for the top soil to return to an initial state (dry soil over the first few centimeters) using soil moisture profiles available for different soils via the AMMA-CATCH observatory (described by de Rosnay et al., 2009). This time is rather short (of the order of 48h, depending on soil type) and it is used to separate the different rainfall events. This justifies to reset the soil moisture to initial condition before each event, especially in an area where Hortonian runoff dominates.

Page 5, "Precipitation and meteorological data…": you need to more detailed your data. Some analyses are needed

We have added the location of the Hombori station in the revised Figure 1 and we give the time scales of the field data (15 minutes for meteorological data used for the STEP model input). In addition, we will add the references to Guichard et al. (2009), Frappart et al. (2009), Timouk et al. (2009) and Gal et al. (2016) who have already analyzed and detailed the in-situ data used here. In addition, we have included new figures, one of them figuring runoff/precipitations for all events.

[Figure]

Page 6, "landscape…" : did you discuss of your results with the local population? Did they validate your maps of landscape/.drainage network evolution?

The site has been visited by our team during several field campaigns per year until 2012, and we are working with locals since then (security issues prevent site access for French scientists). One local village chief has been involved in the project since the beginning, and different other persons have been involved in data collection. One of us (Pierre Hiernaux) is extremely familiar with the site, his first measurements in this area started in 1984, and he has lead the work done on the landscapes map (L. Kergoat and M. Grippa also spent time in Agoufou each year in 2004-2009). We are in regular contact with people living in Hombori and Agoufou, who provide us with valuable information on this region and field data (including regular photographs of the lake, water height, and the vegetation development at the long term monitoring sites).

Page 6, "Rainfall…" : astute approach but you have to validate it. That's why I asked before to more

analyse your climatic data.

We have compared the histograms of rainfall intensity (5-min) obtained by the rainfall disaggregation of daily data from the Hombori synop station (figure below, in grey) to 5-minutes data from different rain gauges around the Agoufou watershed available during the 2005-2011 period (black). The figure below shows that the histograms are quite comparable, particularly for the high intensities, which are the most important for runoff production.

[Figure]

Page 6, lines 35 and after : can we have an idea of how many times you have to widen your intervals?

We will give the statistics corresponding to the figure below in the revised manuscript. Most of the time, intervals are less than 5mm wide (76%).

[Figure]

Page 7, line 4 and after : can't you validate this assumption with the synop station and the stations network of Amma-Catch program?

Indeed AMMA-CATCH stations have been used to address this question. The figure below shows an example of the rainfall PDF derived for different AMMA-CATCH stations for an average precipitation year (There are no others stations than those identified in Fig.1). To further investigate the question, we have also looked at the cloud top temperature, derived by MSG remote sensing data, during this year. This analysis allowed us to conclude that the rainfall cells in the area are generally larger than the watershed area.

[Figure]

Page 8, "model calibration..." : why don't use an automatic calibration? Why these intervals : you tell later that some values found in the literature are higher? Can we have the dispersion of your ten simulation for an event?

Assessment of the channels parameters is not fully automated and requires a large number of simulations and post-processing. This is why we choose to sample a reasonable interval with a limited number of parameter values. The accuracy obtained appears to be sufficient for our objectives. Indeed, given to the compensation of MAN and Ks, different combinations of these two parameters (in the neighborhood of the retained solution) give close values of runoff at the outlet. So we do not think that too much precision is meaningful. It is interesting to note that, once the plane parameters (Ks, MAN etc.) are prescribed according to local map and survey for texture and built-in FAO soil K2 classification, consistent channel values (compared to the literature) are obtained through the calibration. MAN values of about of 0.03 s.m$^{-1/3}$ are commonly reported for Sahelian channels with Ks being more variable depending on the material being eroded on the basin (here mainly silt and clay).

We do not work at the scale of the event so we did not calibrate at that time scale. However, we also calculated the bias at the intra-annual scale (with all observation data) and we get an average bias of -8% (RMSE = 4.6x10$^5$), which is also acceptable.

Page 9, "reference ..." : it is a too simplistic assumption which have an impact on your results... Isn't it possible to use "interpolated situations"?

We are not sure that we completely understand the question but if it is about providing intermediate steps between the "present" case and the "past" case, it is not possible because we have no data (runoff and land cover evolution) available over this intermediate period. LANDSAT satellite data are rare in the 90ies in this region do to the unavailability of a ground reception station to record acquisitions over West Africa.

Page 9, "soil land..." : give ranges of the uncertainties of your process. How can you say that in 2011, the western area of the basin contribute to the Agoufou pond? In the Inner Delta of Niger, there is the same phenomenon of interconnected lakes during some strong events; these interconnections are not necessarily permanent ant can disappear for a while; It's certainly the same here. How can you be sure that it does not happen before, between 1960 and 1975?

On the aerial photographs of 1956 it is clear that the western part is not connected to the main network unlike in 2011. This connection is mainly due to an increase and increased concentration of surface runoff given that the heaviest rains before 1956 were not sufficient to connect the western part.

As specified in the article (4.3.2), even if the connection occurred prior to 1975 it should not change the

results significantly, since the western part contributes only weakly to the outlet (annual volume contribution = $3.3 \times 10^4$ m$^3$).

The maps below, which will be added to the revised paper, shows the spatial distribution of runoff over the watershed. It highlights the important changes in the northern part of the watershed, which more than doubled the contributing area.

[Figure]

Page 13, line 37 : Pierre at al., 2016 is not referenced

Thank you for highlighting this, it will be corrected

 Page 14, line 32 : "erreur…" ????

Corrected. Apologies for this error

Page 15, line 34 : what are the "stocking rates"?

It is "Livestock stoking rate" («pression de pâture" in French)

---

## Author Comment (AC2) · 17 Feb 2017

We thank reviewer 2 for reviewing our manuscript and providing valuable feedbacks. We have now addressed all of his/her comments and discuss them in the following.

The paper focuses on a very important topic of runoff generation processes and their changes through time and in identifying the main causes of such changes in an area of sparse data. The objectives and the general approach that was taken in the study to achieve these objectives are scientifically sound and a good and insightful data analysis is presented.

I have however main concerns of the modeling strategy, assumptions and application. They include: a large gap between the model complexity and the data used for its application; sensitivity analyses are essential to justify many of the modeling decisions made, such as which model parameters to calibrate (leaving out probably the most sensitive parameters); the uniform rainfall assumption is very problematic and should be justified; and, model calibration was made against a single number of mean annual runoff volume (although annual volumes estimation do available). See more details below:

1)        The authors rightly present the need of high temporal resolution rainfall and apply a disaggregation procedure. However, in space they assume a uniform distribution of rainfall over the catchment. This assumption is very problematic. Even if the storm cell is more or less at the same size of the basin or somewhat larger, the cell location in space will be often thus that only a partial coverage is achieved. Given the very high sensitivity of runoff to the partial coverage the exclusion of this factor from analysis might add a large uncertainty. Sensitivity analysis of this of catchment runoff to partial storm coverage should at least be examined.

This is an important point. The figure below shows an example of the rainfall PDF derived for different AMMA-CATCH stations for an average precipitation year (There are no others stations than those identified in Fig.1). Similar rainfall intensity distributions are observed for the different stations, especially for intense precipitation, which contributes to runoff.

[Figure]

To further investigate the question, we had also looked at the cloud top temperature derived by MSG remote sensing data, during this year. This analysis allowed us to conclude that the rainfall cells in the area are generally larger than the watershed area. In addition, the figure below (that will be added to the revised manuscript) shows that the contributing part is located to the north of the watershed. Therefore, only one third of the watershed is concerned.

[Figure]

The analysis of remote sensing data revealed an average time lag of 15 minutes between the east and west of the watershed (squall lines usually propagate westward in the Sahel). We agree with the reviewers that this could have a significant effect on the runoff. In our simulation setup, the impact of a time lag would only affect the timing of water flow entering channels, given that planes are too small (average 1ha) to be affected by the rainfall spatial variability. Thus, flows in channel would be impacted (in fact peak flow should be decreased and flow duration increased, leading to increased infiltration in channels). Such an effect however is compensated for by channel calibration (indeed this is also a reason to calibrate channel parameters). In addition, even if the absolute runoff values would be different in the case of heterogeneous rainfall, there is no reason that this would change the difference between the past and present case runoff, which is the main focus of the paper.

2)      The change in the channel network density between the two periods was presented by a change in planes aspect ratio rather than the CSA elements. Altering the aspect ratio generates a more or a less elongated catchment shape but the drainage density is changed only a little. Why not to utilize the derived channel network maps and identify for each period its own CSA configuration?

It was probably not clear in the first manuscript, but changing the aspect ratio reproduces exactly the elongation of the channel network (since plane's width corresponds to channel length). Changing planes elements was considered also but there was a risk of changing plane properties in an uncontrolled way (changing planes is not straightforward in KINEROS2). That would have made more difficult interpreting changes between the past and present cases. The density factor estimated at 1.3 is not very important at the watershed scale, but when it is computed for the contributing sub-basins, it doubles between the past and the present.

3)      Kineros2 has many parameters. The authors have chosen to calibrate the channel Ks and Manning coefficient, while the other parameters were determined using from data and using different functions. This is a very problematic decision – why these parameters were selected for calibration? What do we know about the accuracy of the other model parameters that are not calibrated? There are two necessary steps that are essential to justify the authors decision: 1) sensitivity analysis that will show the parameters that are most important for calibration (i.e., that model output most sensitive to them), 2) for the pre-defined parameters, assess their uncertainty and examine how this is translated into small uncertainty in model output.

4)      I believe that total runoff is much more sensitive parameters associated with the infiltration process in the plans, e.g., to plans Ks rather than to channel Manning coefficient and probably to channel Ks. The decision to calibrate the two channel parameters, MAN and Ks is not clear and must be supported.

5)      Furthermore, a main impact on annual runoff was found to be the modification of soil properties and vegetation cover, but the model parameters associated with the hydraulic properties of these units were not determined in such a way we have a high certainty in those parameters. Obviously, modeled runoff is very sensitive to these parameters, but they were not calibrated or even examined for their sensitivity.

Based on feedback from other reviewers and reviewer 2, it appears that the objectives of our study and the methodology we employed were not well explained. Therefore, we will better explain this in the

revised manuscript.

The objective of this work is to estimate the impact of observed landscape changes on surface runoff. In that purpose, calibrating the plane parameters would enforce a strong constrain on the model, which it is susceptible to mask out the impact of differences between the past and present case (it is not possible to calibrate each type of plane separately). Our approach was therefore to prescribe the plane parameters from maps of observed landscape changes, using indications on texture (field survey, soil map) and FAO classification soil types. In doing so, we accept that some uncertainty comes from approximated Ks and MAN over planes, but we do not influence the differences between Past and Present.

We fully agree with reviewer 2 that parameters on planes are important in generating runoff. Our paper provides a ranking of the different changes that impacted runoff changes. We have performed a sensitivity analysis using significantly larger Ks (x2.5) and larger MAN (x1.75) for all planes, which shows that the absolute runoff does change but the ranking of the different scenario does not change, as it is shown by the following figures. This sensitivity test is based on data compiled by Casenave and Valentin (1989) for Sahelian soil, and represents the variability for different types of soils. Both parameters changes increase the runoff, so the total effect of changing both Ks and MAN that way is a rather strong test (as it can be seen on the total runoff), which provides some robustness to our ranking results.

[Figure]

Ranking and impact of the different changes observed over time, with all planes Ks multiplied by 2.5 and MAN multiplied by 1.75. The results of this sensitivity test mirror what is found in the article, with a total runoff divided by more than 3.

The calibration of channels parameter plays a minor role, ensuring only that the plane description results in simulated runoff which can match observations with plausible channel parameters. The resolution of the satellite and aerial photographs used to analyze the past and the present does not allow an identification of channel properties and their possible changes over time. The philosophy adopted for this paper was therefore to calibrate the less known parameters (channels) rather than the most sensitive ones. Another reason for calibrating channel properties is to account for the rainfall heterogeneity and time lags in the precipitation events over the watershed, as discussed above

With the default values of the parameters on the planes, we have obtained, via the calibration, values of MAN and Ks in the channels that are coherent with the literature.

We have explain the calibration approach in more details in the revised manuscript.

6)      The calibration strategy seems to me not appropriate. The authors use the bias of the annual runoff as the objective function for calibration; however, Bias does not account for the year by year variations but integrates all year data into a single value, so possibly a large overestimation of modeled runoff in one year can be compensated by a large underestimation in another year. Instead, an objective

function that accounts for the yearly residuals, such as the most popular RMSD objective function is much preferred. It should be emphasized that using the Bias ignores the annual runoff values estimated in the authors previous work and just uses their integration.

7)      Furthermore, a calibration strategy that is based on Bias of runoff volumes, implies that a single number is used for calibration (one data!). This seems not reasonable given the very complex model used and the hard work done to produce very high resolution data.

As explained above, calibration is not critical for the main conclusions of our paper. We fully agree with reviewer 2 that different criteria could have been used, and that RMSE could even ameliorate inter-annual variability. The RMSE values for the calibration simulations is specified in the results. We will specify this in the revised method section.

We have tested K2 sensitivity to our calibration approach by running the model using other channel parameters values (Ks = 40 mm.hr$^{-1}$ and MAN = 0.02 s.m$^{1/3}$). This yields results that are similar to those obtained using the calibration results (For present period: 3.3.10$^6$ m$^3$ for Ks = 30 mm.hr$^{-1}$ and MAN = 0.03 s.m$^{1/3}$ against 3.6.10$^6$ m$^3$ for Ks = 40 mm.hr$^{-1}$ and MAN = 0.02 s.m$^{1/3}$). If the RMSE criterion is chosen, the sets Ks = 40 mm.hr$^{-1}$ and MAN = 0.03 s.m$^{1/3}$ is obtained  We fully agree with reviewer 2 that different criteria could have been used, and that RMSE could even ameliorate inter-annual variability but we chose to keep the notion of bias as referents. In addition, there is not only one bias data but one per calibration year (n = 5) and for ten rainfall event (n = 10).

8)      The authors utilize a very detailed and high resolution hydrological model (which I am not sure is the most appropriate given the very limited data they have), but they do not really take advantage of the detailed simulations. For example, they could try to understand why the change of soil and vegetation properties increased runoff, for which type of rain events it is more pronounced? Are the change manifested in higher peak discharge or in more streamflow events, etc.

Thank you for the suggestion. We have added figures in the revised manuscript. Two of them show the spatial patterns of parameters and results and their changes over time, taking advantage of the distributed nature of the model. The other figure is a runoff/precipitation, which takes advantage of the event-based simulations.

The first figure shows the impact of the landscape changes between present and past on the Manning roughness coefficient and the saturated hydraulic conductivity for the whole watershed. These modifications have led to doubling the contributing part of the watershed.

[Figure]

The second figure represents discharge vs precipitation for all events in the 15 years period in the past and the present. Two main conclusions can be derived from this figure: 1) for the same precipitation intensity, we have twice as much discharge for the Present case. 2) For the present period, rainfall events of 18.8 mm on average, contribute to the discharge whereas in the past rain events of 30 mm are required.

[Figure]

As far as the model choice is concerned, a preliminary study based on a literature review, (not detailed here but found in Gal 2016, PhD thesis), was carried out on 20 models (global, distributed and semi-distributed) in order to choose the hydrological model best suited to the zone and the objectives of the study. KINEROS2 was found to be the well suited (more details on this can be found in the response to reviewer 1)

9)     As rainfall is so highly variable, conclusions about the effect of its possible change should be done with a caution. For example, the authors state that "The results show that changes in daily precipitation regime do not explain runoff changes between the past and the present." (P. 15, L. 31), but even if such changes do occur in reality it is most likely that they are not statistically detectable due to the high natural variability.

We agree with that comment. We will rewrite this sentence, thanks for the suggestion.

**Specific comments**

Thank you for the specific comments and suggestions, they will be taken into account in the revised version of the manuscript

1)     A climatic description of the area is lacking: mean annual rainfall, potential/actual ET, etc.

OK

2)     Only one station is used for rainfall data (and few others are used for the temporal disaggregation); clearly a poor coverage of the catchment, as seen in Figure 1. Have the authors examined the option of remotely sensed precipitation? At least to examine the storm coverage area (which is assumed here to fully cover the catchment.

Yes, we have investigated this issue, which we agree is important. We have also looked at the cloud top temperature, derived by MSG remote sensing data, during this year. This analysis allowed us to conclude that the rainfall cells in the area are generally larger than the watershed area (and especially the contributing part, which is located in the northernmost part of the watershed).

3)     Assumption of soil recovers its initial conditions in two days – this assumption can be reasonable for arid regions. What about the deep soils in the south?

We have calculated the time required for the soil to return to an initial state (dry soil over the first few centimeters, which controls Hortonian runoff) using soil moisture data available via the AMMA-CATCH observatory described by de Rosnay et al. (2009). This time is rather short (48 h depending on soil type).This justified to reset the soil moisture to its initial condition after each event. In Sandy (deep) soils do not contribute to surface runoff.

4)      Add the "absolute value" sign to Eq. 3

OK. Thank you.

5)      The optimization of MAN and KS should be at a higher resolution in my opinion

We do not want to overemphasize the calibration of channels parameters, as it is detailed in response to comment 3, 4 and 5. In addition, assessment of the channels parameters is not fully automated and requires a large number of simulations and post-processing. This is why we choose to sample a reasonable interval with a limited number of parameter values. The accuracy obtained appears to be sufficient for our objectives. Indeed given to the compensation of MAN and Ks, different combination of these two parameters (in the neighborhood of the retained solution) give close values of runoff at the outlet. So we do not think that high precision would be meaningful.

6)      Please clarify how you identified "Isolated dunes (S1) are found at the same location for both periods, but have been eroded and partially encrusted (P. 10).

The images below show that isolated dunes were partially crusted, modifying the hydrodynamic properties of the soil and the growth of the vegetation. This evolution was confirmed by field work (Pierre Hiernaux).

[Figure]

7)      Please represent the RMSE value also in percent from the mean (P. 10 L. 22).

OK.

8)      I recommend to present runoff ratios for each year and to show an example of flood hydrograph.

We have added rainfall/runoff for all events (as well as maps of runoff per plane), which brings information on how the watershed behaves in the Past and Present periods. The observed runoff coefficient have been presented and discussed in Gal et al. 2016, so that we would prefer not to duplicate.

---

## Author Comment (AC3) · 17 Feb 2017

We thank reviewer 3 for reviewing our manuscript and providing valuable feedbacks. We have addressed all of his/her comments and discuss them in the following.

**General comments**

This manuscript presents a modelling exercise made for investigating the causes for the so- called 'Sahelian paradox' that consists of a runoff increase in the last decades after the catastrophical drought of the seventies, in spite of a decrease in annual precipitation. The subject is of interest for the HESS readers, uses a known rainfall-runoff and erosion model as well as rainfall input data derived from networks and new information on land cover, soil types and catchment runoff derived from remote sensing and photointerpretation. The paper is mostly well written and its overall formal quality is good.

Nevertheless, the manuscript handles the issues related to temporal and spatial scales as well as parameterisation in too simplistic ways for the results being sufficiently sound to 'explain' or to 'understand' the Sahelian paradox.

Using a 5-minute step event model designed for small agricultural catchments for simulating the annual discharges of a 250 $Km^2$ basin, assuming that the relationship between daily and 5-minute rainfall intensity did not change on time is not a conventional research approach.

The reasons why we use this kind of model with the data we have (daily precipitation, annual or infrequent runoff) were probably not well explained in the manuscript. We simulate the different hydrological processes at fine spatial and temporal scales. We believe it is necessary given that hydrological processes in the Sahel, as in some other semi-arid areas, are driven by rain events at the sub-hourly time scale (so high time resolution makes sense) and runoff is generated by shallow and impermeable soils occupying a small portion of the landscape (so high spatial resolution also makes sense) by Hortonian runoff. Even if the final objective is to investigate changes between the present and the past 15-year periods, we believe it is critical to use a model that can address this scales. Indeed results from a land surface model intercomparison (Grippa et al, in press in J. of HydroMeteorology, available as early release on line at http://journals.ametsoc.org/doi/pdf/10.1175/JHM-D-16-0170.1 and upon request to M. Grippa) have shown that global land surface models are unable to represent surface hydrology in this area.

In the literature, the size of watersheds studied with the KINEROS2 model varies widely according to the authors. Al-Qurashi et al. (2008) tested the model on a catchment whose area was 734 km² and obtained good results at the annual time scale.

Concerning the rain, we agree that assuming a distribution of rain at 5 minutes which is similar between the present and the past yields some uncertainty. Research on rainfall intensity is currently carried out (including studies by some colleagues of ours) to investigate that, but up to now, no study shows that 5-min intensity has changed between the Past and the Present period.

Al-Qurashi, A., McIntyre, N., Wheater, H., Unkrich, C., 2008. Application of the Kineros2 rainfall-runoff model to an arid catchment in Oman. J. Hydrol. 355, 91–105. doi:10.1016/j.jhydrol.2008.03.022

More essentially, as it is well known among the hydrological community that any hydrological model can give good results for the wrong reasons, when the purpose is not to obtain good results but to investigate the reasons, the researcher must be particularly cautious to take into account the likely equifinality of diverse possible parameterizations.

We agree with reviewer 3 statement. Indeed we have carefully selected the hydrological model to use in this study based on its capability of representing the hydrological processes that characterize the study region (model choice is explained in response to reviewer 1 and 2). In addition, we have only calibrated the parameters of the channels to check that the planes parameters produce reasonable runoff (= runoff that fit observation with plausible channels parameters), as explained in response to rev 1 and 2. The overall philosophy was not to calibrate and optimize the most important parameters (planes KS, MAN) and to constrain the model as less as possible. A sensitivity test which multiplies planes KS and MAN by 2.5 and 1.75 leads to the same ranking in terms of Past/Present changes, which provides robustness for our main results. Of course, further studies with different models would be interesting and the data are being put on the AMMA-CATCH database in that purpose.

The paper might be accepted for publication if both the model parameters and results were more investigated. The analysis of the contribution of the diverse factors to the change of the catchment response is a strength of the paper, but it assumes just a 'correct' parameterisation; an uncertainty analysis should be made or, at least, a sensitivity analysis of the model response to parameter variation.

As said above, we have performed a sensitivity analysis using significantly larger Ks (x2.5) and larger MAN (x1.75) for all planes, which shows that the absolute runoff does change but the ranking of the different scenario does not change, as it is shown by the following figure. This sensitivity test is based on data compiled by Cazenave and Valentin (1989) for the Sahel and represents the variability for different types of soils. Both parameters changes decrease the runoff, so the total effect of changing both Ks and MAN that way is a rather strong test (as it can be seen on the total runoff), which provides some robustness to our ranking results.

[Figure]

In the same vein, simulations performed with different values of Ks and MAN for the channels give similar results.

Annual catchment discharges should not be the unique focus of model output, but some model results at the event scale (extreme events, annual number of runoff producing events, rainfall threshold values...) and at the landscape unit scale (identification of contribution areas for diverse type of events...) should also be shown and discussed.

Following this suggestion, we have added 2 figures in the revised manuscript, and we agree that they provide very interesting information, thank you for this suggestion. The first figure shows the impact of the landscape changes between present and past on the Manning roughness coefficient and the saturated hydraulic conductivity for the whole watershed. These modifications have led to a doubling of the contributing part of the watershed.

[Figure]

The second figure represents discharge vs precipitation for all events in the 15 years period in the past and the present. Two main conclusions can be derived from this figure: 1) for the same precipitation intensity, we have twice as much discharge for the Present case. 2) For the present period, rainfall events of 18.8 mm on average, contribute to the discharge whereas in the past rain events of 30 mm are required (intercepts of linear fits for non-zero discharges).

[Figure]

Yet, the authors must make a rigorous distinction between model and reality, simulations and observations, as well as more clearly separate drivers, processes and model parameters.

We agree, we will modify our writing in the new version of the manuscript, adding 'according to the model' and similar sentences.

**Detailed comments:**

Page 1, line 16: KINEROS is not a water balance model but a rainfall-excess runoff model. Water balance, which is the main challenge in the Sahelian paradox, is therefore indirectly simulated, as runoff water is subtracted from infiltration and subsequent evapotranspiration. This is a relevant aspect to be stated in order to assist the understanding of the paper by readers not familiar with KINEROS.

OK. Thanks for the suggestion.

Page 1, line 17: it is necessary to state the catchment area

> OK. It will be done.

Page 1, line 23: "shallow soil being eroded and giving place to impervious soils" please rewrite in a less literary way

> OK, will be done.

Page 1, line 24: the converse is more rigorous: "The KINEROS-2 model was parameterized in order to simulate these changes in combination or independently".

> OK

Page 1, line 26: Catchment flows shown in volume ($m^3$) cannot be related to rainfall rate and this is not usual in the hydrological literature because volume depends on catchment area, it is more adequate to show them in runoff units (mm per year). Showing only the simulated results is not informative at all (are those simulated with KINEROS2?).

> We will add the correspondence between volume in m3 and mm/year and use mm/year when possible as in the rainfall/discharge figure above. (From a water resource point of view volume in m3 is also informative).

Page 1, line 29: "Modification" refers to the action of changing something and should not be used for a natural change. This is unclear that vegetation cover was modified in the parameterisation. Please describe more precisely the modification of parameters made for simulating the landscape changes.

> OK

Page 2, Line5: During or after the drought?

> OK

Page 2, line 25: Reduction of vegetation cover and topsoil crusting are factors of too different nature to be cited together. Reduction of vegetation cover can (directly) decrease rainfall interception and plant transpiration, or decrease the sol protection against raindrop energy, so (indirectly) favour soil crusting. Soil crusting usually decreases infiltration rates and favours rainfall-excess overland flow. Please, describe more precisely the drivers and mechanisms that have been pointed out for explanation of the 'Sahelian paradox'.

> OK, thank you for the suggestion.

Page 4, line 2: "and runoff is frequently generated over them."

> OK

Page 4, line 21: KINEROS was not designed explicitly for arid and semiarid areas.

> Agree, but most of K2 applications concern semi-arid zones so far. The sentence will be changed accordingly.

Page5, lines 18-22. These are not soil and land cover data but just images.

> OK

Page 6, line 13 (and below): "Erosion surface" seems to refer to a geomorphic unit (land form), but here is not a good denomination for a landscape unit because it is not clear to most readers and it is equivocal with the soil erosion processes. "Pediment" is a geomorphic English term equivalent to the French "glacis" term that could be used instead.

> OK. We will change the text accordingly, thank you.

Page 6, line 30 and subsequent: This is one of the main weak points of the paper, as the method used assumes that there is no change in the fine temporal structure of rainfall events. In the lack of data to improve the approach, some sensitivity exercise should be made to test the role of changing this structure on runoff generation. This may be made using a range of 'ensemble' 5- minutes series with higher, average and lower 5-minute intensity within reasonable bounds.

> This is certainly an important topic. However, we have little information so far to provide reasonable bounds for 5-min intensities. There seems to be a change in the frequency of high rainfall events

(Frappart et al. 2009, Panthou et al. 2014), starting around 2000. This is a rather weak signal though. To our knowledge, trends for the very scarce 5-min long times series (ex. Niamey station, Léauthaud et al. 2016) have not been evidenced. We feel like such a sensitivity test would be a little bit speculative for this paper and we prefer to stick to the conclusion that daily rainfall regime change (which has been observed with large dataset) does not contribute to increase runoff in our case.

Leauthaud, C., Cappelaere, B., Demarty, J., Guichard, F., Velluet, C., Kergoat, L., ... & Mainassara, I. (2016). A 60-year reconstructed high-resolution local meteorological data set in Central Sahel (1950–2009): evaluation, analysis and application to land surface modelling. International Journal of Climatology.

Page 7, line 19 and subsequent: In fact, there is a terminological confusion in the paper respect to the changes in the drainage network: the changes observed are really changes in the stream channel network; in the old period runoff was too slight or infrequent in the thalwegs to form distinguishable channels that were cut after the drought period (see another comment below). The subsequent paragraph describes how the DEM derived drainage network was adapted to the network observed in 2011, but not clearly how the 'old' network was parameterized.

OK, we agree that additional details are needed here. This will be explained in more details

Page 7, line 34: Thickets

Page 8, line 1: Low LAI is a reason but high winds favour the evaporation of intercepted rainfall. Check the rainfall interception literature in semiarid areas (e.g. Llorens & Domingo Journal of Hydrology, 2007)

We have been checking the literature before assuming no interception. There are few data for similar biomes and similar climate (strong convective event with strong gusts, low vegetation cover).The nearest case studies are for semi-desert sites or desert sites, for instance the sites documented by Carlyle-Moses, D. E. (2004). Throughfall, stemflow, and canopy interception loss fluxes in a semi-arid Sierra Madre Oriental matorral community. *Journal of Arid Environments*, *58*(2), 181-202. (Review by Llorens and Domingo is mainly for trees, and include 3 bush sites with Mediterranean climate).
Studies in arid/semi-arid environment point to small interception losses (e.g. less than 10%) with high throughfall and significant stemflow.
In our case, the rainfall rates that produce runoff are the highest convective rates, for which interception can be neglected. We agree that interception can occur for instance at the very end of a convective event, when 'stratiform' precipitation sometimes occur, but these do not produce runoff. Also, the area contributing to runoff has an extremely low vegetation cover. Deep soils with important grass cover and scattered trees don't produce runoff, although interception losses are set to zero.
Therefore, we believe it is reasonable to neglect interception when the objective is runoff simulation.
In terms of evaporation during convective events, the high winds (gusts) come with high relative humidity (close to 100%), cold air, and large cloud cover (and the diurnal cycle of convection and squall line produces maximum rainfall from late afternoon to early morning). Information can be found in Frappart et al. 2009, Guichard et al. 2009, Samain et al. 2008, Largeron et al. 2015, for rainfall and associated meteorological data.

Largeron, Y., Guichard, F., Bouniol, D., Couvreux, F., Kergoat, L., & Marticorena, B. (2015). Can we use surface wind fields from meteorological reanalyses for Sahelian dust emission simulations?. *Geophysical Research Letters*, *42*(7), 2490-2499.

Samain, O., Kergoat, L., Hiernaux, P., Guichard, F., Mougin, E., Timouk, F., & Lavenu, F. (2008). Analysis of the in situ and MODIS albedo variability at multiple timescales in the Sahel. *Journal of Geophysical Research: Atmospheres*, *113*(D14).

Page 9, line 22 and subsequent: In the writing of the following paragraphs there is sometimes confusion between the changes of the extension of the mapped landscape units, the changes of the properties of these units and the related hydrological processes.

OK, this will be rewritten. Thank you.

Page 10, line 9 and subsequent: Please, change the "Erosion surface" term.

OK

Page 10, line 11: "an important erosion of the underlying soil has occurred": do you have evidences of this phenomenon? Where are the eroded soils deposited? "Impervious bare soils have replaced most of these areas": this is not a rigorous description of a landscape change. In all this paragraph there is confusion between changes in the map units, the characteristics of these units and the processes related to these changes (as causes or consequences).

This will be rewritten more clearly. Erosion is both wind driven (particles are exported/imported) and water driven (particles are deposited in ponds and channels) and both processes interact (wind driven particles can be washed out by water erosion for instance, and dry banks can be eroded by the wind).

Page 10, line 15: The development of a drainage network (in fact this seems to mean that new channels are observed in previously unchannelled thalwegs) may be attributed to the increase of overland flow, but not necessarily to the change from sheet runoff to concentrated runoff on the hillslopes, unless new rills and shallow gullies are observed throughout. The entrenchment of channels in semiarid conditions has been attributed to increased runoff or the decay of valley bottom vegetation (e.g. Nogueras et al. Catena 2000 and cited references).

The two factors can play a role even if it is not easy to distinguish between them (an example of changes in the drainage network is shown below)

[Figure]

[Figure]

Runoff concentration was pointed out by several studies carried out in the Sahel also. A typical case is the transformation of a tiger bush (e.g. site 8 of the long term survey, Dardel et al. 2014a ), with vegetation bands perpendicular to the flow, which is replaced by bare soil with scattered trees (mostly Acacia ehrenbergiana)  that grow along the newly created rills, parallel to the slope. Field survey provide many example of conversion of sheet to concentrated runoff in this area. Of course there is an interplay between increased runoff, concentration, vegetation decay.

Page 11 line 5 and subsequent. This sub-section is not well written. The parameterisation of the channels is not a result. Please, describe changes in precipitation before changes in discharge and use a chronological order of the periods when possible. Reporting discharges in volume is really difficult to follow, please use runoff units (mm/year). Please, state observations before simulations throughout.

This will be done.

Page 11, lines 16-18: this paragraph is unnecessary unless the behaviour of the catchment is better described, as proposed above.

OK

Page 11, line 24: Please, include a sentence recalling that the reference run is the recent period and that changes are evaluated using equation (4).

OK

Page 11, line 26 and subsequent: "... present characteristics except dune crusting ...". "... has two effects on the parameterisation of the land surface ...". "... dune crust on the simulated annual  discharge...". Reporting volumes for the sub-basins is confusing, please report percentages of the total runoff and clearly state that these are simulated values for the recent/older periods.

OK

Page 12, lines 3 and subsequent: Please, state the (indirect) effect of the vegetation changes on model parameters, this is to help understanding the runoff slowing and infiltration increase. Please, report discharge in mm.

OK

Page 12, lines 14 and subsequent: "Modification" and "erosion surfaces" are not appropriate terms here, as discussed above. The "increase in erosion surfaces" is contradictory respect to the small changes in these units as described in Page 10 line 9. Here there is confusion between soil properties and landscape units, please be more explicit.

OK thank you for the suggestion.

Page 12, lines 20 and subsequent: " the result is an increase of  xxx mm/year of the discharge for the past period..."

OK

Page 12, line 28: mind the confusion between soil and landscape unit

OK

Page 14, line 36: "... and soil properties may largely explain..."

OK

Page 15, lines 3-4: "The lack...concentrated runoff": There is a melange of causes and consequences, yet, the change from sheet to concentrated runoff must be demonstrated.

OK. Runoff concentration is best observed during field survey (new rills, sometimes cutting through sand / silt bars, and vegetation changes from tiger bush tickets perpendicular to the slope to scattered trees like Acacia ehrenbergiana growing along rills parallel the slope. Hiernaux et al. 2009, Dardel et al. 2014a)

Page 15, lines 13-14: See the note above on channel entrenchment.

OK

Page 15, line 14 and subsequent: "Our work has shown that enhanced and concentrated runoff results in an increase in both the number and the length of channels, therefore increasing the drainage density and diminishing the travel time for water to reach the drainage network" : This is not shown in the results above.

See comment above about this. We will rephrase and stick to drainage network development.

Page 15, line 19: "Our results suggest that..."

OK

Page 15, line 28: "are simulated as part of vegetation..."

OK

Page 15, line 33: "surface runoff is observed and simulated to decrease..."

OK

Page 15, lines 36-27: a preliminary test should be made changing the fine temporal structure of rainfall, as suggested above.

See comment above.

Conclusions: this section should be rewritten after the revision of the manuscript, but it is important to bear in mind that in this case the model approach may be useful to "investigate" or to "shed some light on" the paradoxical evolution in the Sahel, but not to "understand" it.

OK, we agree with this statement. This will be done. Thank you for the suggestion.

---

## Author Comment (AC4) · 17 Feb 2017

> We thank the reviewer for reviewing our manuscript and providing his valuable feedbacks. We have addressed all of his/her comments and discussed them in the following.

The paper deals with the "Sahelian paradox" phenomenon where despite a decrease of the precipitation in the Sahel during the last 50 years, an increase of runoff was observed. The Agoufou catchment (245 km$^2$) is taken as a study case, and the spatially distributed model KINEROS2 used to prioritize the different factors (dune crusting, drainage network development, vegetation changes, modification of soil properties, or a combination of some of these factors) that lead to this paradox.

My first feeling is that the title of the paper does not reflect its real content because the paper remains an application of a model on a catchment. Neither the catchment nor the model was chosen to demonstrate a hypothesis related to the "Sahelian paradox".

Moreover the use of the model to prioritize the factors causing the increase of runoff remains a numerical modelling exercise without any validation using hydrological data. The model can give good results for the bad reasons. Consequently the conclusions of the paper on the main factors causing the "Sahelian paradox" may be correct, but may also be not correct.

> We will change the title of the paper according to this comment and comments by the other reviewers. We agree that our conclusions are based on a model, and as such call for future work with other models for instance.
>
> The study site (the Agoufou watershed), has been chosen to address the Sahelian paradox. See below for detailed answers.

My comments concern:

i ) The choice of the studied basin: The paper doesn't justify the basin choice in com- parison to other catchments in the same region. Does the land use change and the consequences on the "Sahelian Paradox" observed on the neighboring catchments. In order to demonstrate (or not) the hypotheses and quantify the role of the different factors which lead to the "Sahelian Paradox", it would be preferable to choose the- catchments with internal stream-gauges and piezometers.

> The study site was chosen for several reasons:
>
> This site is instrumented by the SO AMMA-CATCH which allows to have field data (soil moisture, LAI, observed discharge, etc.) in the long term (starting in 1984 for the long-term ecological survey), as well as good field knowledge by co-authors.
>
> Gal et al. (2016) show the spectacular evolution of the volume of water entering the Agoufou lake (outlet of the watershed) over the past 60 years despite the decrease in precipitation. This increase in the ponds level is a very good example of the Sahelian paradox that has also been demonstrated in other Sahelian watersheds in Northern Mali (see Gardelle et al. 2010).
>
> This site is a pastoral catchment area where agricultural activity is almost non-existent and cannot therefore be an explanation for the Sahelian paradox (the land use change hypothesis has been put forward in other places). It is therefore in these areas that the debate is open.
>
> These three main reasons explain the choice of this study site, although it will be certainly interesting to extend the analysis carried out on this study site to other Sahelian watersheds.

An increase of runoff will be accompanied with a modification of the other hydrological processes especially the water table level and extension, and evapotranspiration. However, the paper doesn't deal with these two main hydrological processes due to the lack of data. Consequently, the available data are not sufficient to validate or not a given hypothesis.

> Evapotranspiration has been monitored with a network of flux stations (up to 5) deployed over different soil units over 2005-2010 (see Timouk et al. 2009, and more data unpublished). It has also been

modelled with different LSM or SVATS (Grippa et al. 2017 in press, Garcia et al. RSE 2013, Bateni et al., 2014 among others). We propose to add a sentence stating that the expansion of rocky soils, silty layers or iron pan likely yields a slight decrease of evapotranspiration over the watershed, coupled to an increase in lake evaporation. Indeed, over deep sandy soils, evapotranspiration is close to rainfall (95% or more, with some uncertainty due to the eddy covariance technique) whereas is it much lower on pediments (< 50%, Timouk et al. 2009). The change in evapotranspiration however is really small compared to the change in runoff and we prefer not to overemphasize it.

There is ongoing work on the water table, which is not facilitated by the security issues in Northern Mali. Geology in the Gourma is such that water tables are variable is size and depth (completely different from the "Continental Terminal" In Niger for instance). Local people do not report systematic evolution of well levels (note that this may be related to the point that water tables are variable and complex in the Gourma, and are not the main water resource used by people there).

Last, in a system dominated by Hortonian runoff, the main players are not evapotranspiration and water tables but rainfall and land surface (the water table is well below the lake bottom, so that it does not feed the lake).

Bateni, S. M., Entekhabi, D., Margulis, S., Castelli, F., & Kergoat, L. (2014). Coupled estimation of surface heat fluxes and vegetation dynamics from remotely sensed land surface temperature and fraction of photosynthetically active radiation. *Water Resources Research*, *50*(11), 8420-8440.

García, M., Sandholt, I., Ceccato, P., Ridler, M., Mougin, E., Kergoat, L., ... & Domingo, F. (2013). Actual evapotranspiration in drylands derived from in-situ and satellite data: Assessing biophysical constraints. *Remote Sensing of Environment*, *131*, 103-118.

ii) What we learn from data: The paper doesn't present a detailed analysis of the spatio-temporal data and do not discuss the evolution of the components of the water balance. The authors must first discuss what we learn from the data only, and in a second time what is the added value using the model.

The data used for modeling are the hydrodynamic soil parameters are derived from the landscape maps described in this paper. We use measured runoff data (Lake Agoufou water balance), which are detailed in Gal et al. (2016) and we prefer not to duplicate what is already written in this first paper. We will make it clear in the revised manuscript.

iii) The available data: The main problem is that only "annual" water outflow is available, reconstructed by the author for some years (see Table 1)! Moreover, one rain-gauge is available on the catchment, and data at a fine time step (5 min) are only available for given periods. The lack of analysis of the spatio-temporal structure of rainfall at 5-min time step, and the use of an empirical method for temporal disaggregation is a weak point of the study. Moreover, the paper limits the analysis at the annual water balance and no information is given on flood events characteristics on 5-min time step: evolution from the 50th until now of the rainfall intensities, runoff coefficient, peakflows, lag time, etc. The data are not coherent: a fine DEM resolution (30m) vs annual flow and daily rainfall! I'm not sure that the available hydrological data are sufficient to give responses to the important questions raised in the introduction!

This is an important point, and we think we need to give more explanations and information in the revised paper.

First of all, we propose to add additional figures (one with maps of Ks and MAN for Past and Present, and one figuring runoff versus rainfall for all events of the Past and Present, see below). Thank you for suggesting adding information on the spatial and temporal features of hydrology and changes over time.

Then, we will explain why looking at yearly or 15-year runoff comes from the fact that we want to use 5-min rainfall input. Indeed, given that we need to perform a temporal disaggregation of daily data, which creates noise and variability in the 5-min precipitation forcing, we need to consider ensemble-mean and we need to average over as many events as we can. Using a look-up table of 5-min events

preclude from looking at a particular event, since it will not provide the real 5-min intensity for this event, but will on average provide the right distribution of 5-min intensity (see the histogram of intensities, below, for a comparison). That was probably not clear enough in the manuscript. To document the dispersion caused by temporal disaggregation, we have shown the envelopes of the ten ensemble members (Figure 6). We believe it is really important use 5-min rainfall, to be able to implement changes in land surface in a physical way (Hortonian runoff in a climate with short and intense convective precipitation from squall lines)

New figures: The figure below shows the impact of the changing landscape on the Manning roughness coefficient and the saturated hydraulic conductivity in the northern part of the watershed. These modifications have led to doubling the contributing area.

[Figure]

The second additional figure represents discharge for all events as a function of event precipitation amount. Two conclusions can be drawn from this figure: 1) for the same precipitation amount, we have twice as much discharge for the present case. 2) For the present period, rainfall events of 18.8 mm and larger average contribute to the discharge whereas in the past rain events larger than 30 mm were required.

[Figure]

Last, the figure below shows a comparison between histogram of rainfall intensity of two different sources of rainfall (disaggregated and 5-min raingauges). There are quite comparable, particularly for the high intensities which are the most important for runoff production.

[Figure]

iv)      The uncertainty on data: The authors must discuss the uncertainty on the hydrological data (e.g. spatial distribution of rainfall, the annual runoff reconstructed) and the consequences on modelling results.

We agree we can provide more information on uncertainty, which is also an important point. For observational data, a full analysis of the uncertainties is detailed in Gal et al., 2016 Journal of Hydrology). We will give more details about this in the revised manuscript.

For the uncertainties on the planes parameters, we will also provide some additional results in the revised manuscript. A sensitivity study has been carried out to highlight the robustness of the model in ranking the factors responsible for the increase of surface runoff. To that end, Ks of all planes was multiplied by 2.5, and MAN by 1.75. This corresponds to the interval given by Casenave and Valentin (1989) for many Sahelian soils. Both changes (Ks and MAN) tend to decrease runoff, therefore the combination of KS and MAN decrease total runoff by a factor of 3.

[Figure]

Ranking and impact of the different changes observed over time are similar to what is found with the original planes parameters.

Last point: Uncertainty due to the use of homogeneous rainfall (one station used): This is an important point. The figure below shows an example of the rainfall PDF derived for different AMMA-CATCH stations for an average precipitation year. Similar rainfall intensity distributions are observed for the different stations, especially for intense precipitation, which contributes to runoff.

[Figure]

To further investigate the question, we had also looked at the cloud top temperature derived by MSG remote sensing data, during this year. This analysis allowed us to conclude that the rainfall cells in the area are generally larger than the watershed area (the contributing part is located to the north of the catchment. Therefore, only one third of the watershed is concerned.) The analysis of remote sensing data revealed an average time lag of 15 minutes between the east and west of the watershed (squall lines usually propagate westward in the Sahel). We agree that this could have a notable effect on runoff. In our simulation setup, the impact of a time lag would only affect the timing of water flow entering channels, given that planes are too small to be affected by the rainfall spatial variability. Thus, flows in channel would be impacted (in fact peak flow should be decreased and flow duration increased, leading to increased infiltration in channels). Such an effect however is compensated for by channel calibration. In addition, even if the absolute runoff values would be different in the case of heterogeneous rainfall, there is no obvious reason that this would change over time and impact the difference between the past and present case runoff, which is the main focus of the paper.

v)      The choice of the hydrological model: The spatially distributed model KINEROS2 is used without any justification. I'm not sure that this model is the most appropriate for the available data (only one rain-gauge, and total annual runoff). The paper doesn't demonstrate that the prioritization of factors causing the "Sahelian paradox" are inde- pendent of the model choice. Comparing different models will give more arguments for the discussion.

A previous study, not detailed here (found in Gal L., 2016, "Modélisation de l'évolution paradoxale de l'hydrologie Sahélienne. Application au basin d'Agoufou", PhD thesis, Université de Toulouse) based on a literature review was carried out with 20 different hydrological models (global, distributed and semi-distributed) in order to select the model best suited to the zone and the objectives of the study. KINEROS2 was found to be suited for the study purpose.

In addition a model/data intercomparison project, called ALMIP2 for AMMA Land Surface Model Intercomparison phase 2, has been carried out in this area to assess the capability of land surface models (LSMs), vegetation models and hydrological models to describe hydrological processes in this area: 20 different LSMs  were analyzed (Grippa et al, in press in JHM, available as early release on line at http://journals.ametsoc.org/doi/pdf/10.1175/JHM-D-16-0170.1 or upon request to M. Grippa). The results highlight the difficulty of models to distinguish between shallow or silty soils generating the runoff ending up in ponds and no-runoff areas, like the sandy deeper soils, which infiltrate all rainfall. LSMs have been found to be too sensitive to rain and not enough to soil properties. We hope that our results will stimulate the scientific community to undertake further studies in different basins and with different models to validate or invalidate our findings. The data are being put on the AMMA-CATCH database in that purpose.

vi)      The calibration procedure: an important number of the model parameters were arbitrarily fixed and only two parameters were calibrated. The values of the calibrated parameters will depend on the values chosen for the fixed ones.  The authors must justify the choice of the parameters to be calibrated, and discuss how a modification of the fixed parameters will impact the conclusions of the paper.

The calibration approach was probably not clearly explained in the first manuscript.
The plane parameters values were derived from FAO codes and soil texture data from field studies. These parameters have not been adjusted because we want to investigate how their changes impact surface runoff (note that we have performed a sensitivity study that provides robustness to our results). Calibration was performed on channels parameters, which are not well constrained by observations. The calibration mostly show that the runoff simulated over the planes yields observed total watershed runoff with plausible values of channel parameters. This is satisfying, of course, but our results on ranking factors does not depend on the calibration.

vii)      The criteria function: Only the "Bias" (Eq 3), at the annual time step, was chosen as a criteria function.  The paper doesn't present any simulated hydrographs, neither other values of the criteria function (especially criteria related to peakflows) in order to evaluate the performance of the model. Different criteria functions must be used.

OK. RMSE is added to the Bias in the calibration results. We will include this in the "method" section of the revised article. We cannot use the usual criteria in hydrology (Nash, KGE) because we have little data at the intra-annual scale. We have added also a discharge/rainfall plots with all events (see above)

viii)      In order to study the "Sahelian paradox", it will be interesting to compare the com- ponents of the water balance on a large number of basins (and more especially embed- ded ones). Before undertaken a modelling exercise, an analysis of data is necessary in order to link (or not) the evolution of "hydrological" processes to the evolution of land use.

We completely agree that looking at the water balance of many watersheds is highly desirable. There is now ample evidence for the Sahelian paradox (see review by Descroix et al. 2009, and there is an ongoing review paper, by Descroix et al. also, that will update the state of the art on that subject). We contribute to this scientific question in adding information on pastoral area (no or very few crops), and endorheic areas, as well as on ecohydrology processes (Gardelle et al. 2010, Dardel et al. 2014a, Gal et al. 2016, Sighommou et al. 2012, Gal et al. this study, Descroix et al. in prep). It is not easy however

to decipher the drivers of the paradox, since many factors do change over time (hydrology, but also climate, land use, demography, crop management, etc…). We believe modelling is also important in highlighting possible causations and new or possibly overlooked factors, until a clear picture emerges. Of course, the fact that few data exist (even on land use) is an issue. We believe our study points a number of important questions on this debated subject.

**Other comments:**

All comments below will be taken into account in the revised version of the manuscript with the exception of the comment on fig 5 (see below) . Thank you for these suggestions.

- Abstract: please indicate the catchment area in the abstract.

- P3, L11-15: The objectives of the paper are reduced to an application of a model on a given basin, and don't give responses to the main question related to the "Sahelian paradox".

- The title of the paper must be in accordance with the objectives announced.

- Section 2.1 "Study site": The hydrological data 'rainfall, runoff) and the spatial data (land use, soil, geology, DEM, etc.) must be presented in this section and nor as input to the KINEROS2 model. The authors must discuss what we learn from data before undertaking a modelling exercise.

- What is the uncertainty on the delimitation of units on maps (from Table 2) and consequently on the area of each class of land use in space and in time (Table 6)?

- How the drainage network was defined on Figures 3 and 4? How the channel network was interpolated in time between 1956 and 2011?

- The Manning coefficient MAN has a unit (($m^{1/3}$ / s)

- Table 3: How these parameters were fixed? What is the sensitivity of the mode l results of these values are taken different?

- Figure 5: The grid used must be refined?

We believe the accuracy we obtain is sufficient for our objectives. Indeed, given that there is some compensation between MAN and Ks, the different combinations of these two parameters close to the one we retained give quite similar values of runoff at the outlet. (For the present period: $3.3.10^6$ m$^3$ for Ks = 30 mm.hr-1 and MAN = 0.03 s.m$^{-1/3}$ against 3.6.106 m$^3$ for Ks = 40 mm.hr-1 and MAN = 0.02 s.m$^{-1/3}$). Basically, it is interesting to note that the map of planes parameters we use combined with channel values consistent with the literature are able to match observed runoff. The main conclusions of our paper (ranking) are not sensitive to this calibration values (see also the sensitivity test for planes parameters). Literature, that often reports MAN values of about of 0.03 s.m$^{-1/3}$ for Sahelian channels and variable Ks depending on the material being eroded on the basin (here mainly silt and clay).

---

## Author Comment (AC5) · 17 Feb 2017

**Referee comment on "Modeling the paradoxical evolution of runoff in pastoral Sahel. The case of the Agoufou watershed, Mali" by Laetitia Gal et al. (Hydrol. Earth Syst. Sci. Discuss., doi: 10.5194/hess-2016-623, 2016.)**

This reviewer largely agrees with many of the comments already expressed by reviewers RC2 and RC4. Given the numerous issues expressed I feel the paper should be reframed and possibility retitled along the lines expressed by reviewer RC3 whose last suggestion was "Conclusions: this section should be rewritten after the revision of the manuscript, but it is important to bear in mind that in this case the model approach may be useful to "investigate" or to "shed some light on" the paradoxical evolution in the Sahel, but not to "understand" it." A new title might be something like "Exploration of the paradoxical evolution of runoff in the pastoral Sahel - Agoufou Watershed using available data and a watershed model."

> We thank reviewer 5 for providing valuable comments and remarks on the first manuscript, as well as on the comments/suggestions of the other reviewers. We appreciate the suggestions based on a deep knowledge of the K2 model.
>
> The title will be modified following your suggestions, thank you. Also, we will moderate the terms used in the conclusion and in the manuscript to emphasize we present modelling-based conclusions.

The author could then stress that the model selected could be one of many for this investigation, but K2 was selected for x, y, and z reasons as a tool to investigate possible reasons for the paradoxical evolution of runoff in the Agoufou watershed. Within the constructs of the model, its structure, and the assumptions inherent in the model it was felt it could be used to conduct a relative ranking of various factors and watershed attribute changes contributing to the paradox. Using other models one might come to different conclusions or attributions but the authors could encourage others to conduct comparable "detective" investigations to better understand factor contributing to the paradox.

> A preliminary study, not detailed here (found in Gal L., 2016, "Modélisation de l'évolution paradoxale de l'hydrologie Sahélienne. Application au basin d'Agoufou", PhD thesis, Université de Toulouse) based on a literature review was carried out with 20 different hydrological models (global, distributed and semi-distributed) in order to select the model best suited to the zone and the objectives of the study. KINEROS2 was found to be suited for the study objectives. In addition a model/data intercomparison project, called ALMIP2 for AMMA Land Surface Model Intercomparison phase 2, was carried out in this area to assess the capability of land surface models (LSMs), vegetation models and hydrological models to describe hydrological processes in this area: 20 different LSMs were analyzed (Grippa et al, in press in JHM, available as early release on line at http://journals.ametsoc.org/doi/pdf/10.1175/JHM-D-16-0170.1 and upon request to M. Grippa). The results highlighted the difficulty of models to distinguish between shallow or silty soils generating the runoff ending up in ponds and no-runoff areas, like sandy deeper soils, which infiltrate all rainfall. LSMs have been found to be too sensitive to rain and not enough to soil properties. These various arguments explain why we have chosen the Kineros2 model.

As pointed out by the other reviewers the uniform precipitation assumption for a basin this size constitutes a major simplification and calls into question the ability to carry out a defensible model calibration and validation. Al-Qurashi et al. (2008) applied K2 to a 734 $km^2$ arid watershed with 7 rain gauges where a "parameter set which gave best calibration performance over any combination of 26 events did not generally produce acceptable performance (defined as within 30% of observed) when used to predict the 27th event". In this and similar situations, the authors noted that "data sets typically used for distributed (or semi-distributed) rainfall-runoff modeling in arid regions cannot provide an accuracy which justifies the effort and expense of this (K2) modeling approach. The limitations imposed by relatively sparse observations of rainfall are of particular concern" (Al-Qurashi et al., 2008, p. 104).

The uniform precipitation hypothesis is an important point and has required additional analyzes not detailed in the manuscript. The figure below shows an example of the rainfall PDF derived for different AMMA-CATCH stations for an average precipitation year (There are no others stations than those identified in Fig.1, so we have no station in the north of the Agoufou watershed). Similar rainfall intensity distributions are observed for the different stations, especially for intense precipitation, which contributes to runoff.

[Figure]

To further investigate the question, we had also looked at the cloud top temperature derived by MSG remote sensing data, during this year. This analysis allowed us to conclude that the rainfall cells in the area are generally larger than the watershed area. In addition, the figure below (that will be added to the revised manuscript) shows that the contributing part is located to the north of the catchment. Therefore, only one third of the watershed is concerned.

[Figure]

The analysis of remote sensing data revealed an average time lag of 15 minutes between the east and west of the watershed (squall lines usually propagate westward in the Sahel). In our simulation setup, the impact of a time lag would only affect the timing of water flow entering channels, given that planes are too small to be affected by the rainfall spatial variability. Thus, flows in channel would be impacted (in fact peak flow should be decreased and flow duration increased, leading to increased infiltration in channels). Such an effect however is compensated for by channel calibration. In addition, even if the absolute runoff

values would be different in the case of heterogeneous rainfall, there is no reason that this would change the difference between the past and present case runoff, which is the main focus of the paper.

Qurashi et al. (2008) analyzed their results at the event scale unlike we do. We analyze the results on an annual scale and 15-year average scale. The point is indeed to analyze the results in a statistical way in the light of uncertainties related to the precipitation.

To remove this major limitation and use K2 as a tool to explore causes of the runoff increase this reviewer suggests taking a relative change approach as advanced by Goodrich et al. (2012) and Sidman et al. (2015) for post-fire watershed assessment in watersheds that often do not have any rainfall-runoff data available for calibration and validation. In this approach a pre-fire land cover map is used to parameterize the watershed and conduct a simulation with a spatially uniform design storm. The burn severity map is then used to alter model parameters based on prior research and analysis of the effects of burns on cover and soil hydraulic properties. A second post-fire simulation is then conducted using the same rain storm. The results can then be spatially differenced to analyze changes. For the author's study the present and past model parameterizations based on analysis of historic and current land cover and soils data are analogous to the pre- and post-fire conditions. The authors could pick one of their most trusted rainfall data sets (perhaps when they had high temporal resolution measurements) and use that rainfall input data set for both the past and present watershed model parameterizations. Given one of their conclusions (last paragraph) was that climatic and precipitation changes from past to present appeared too little or no impact on the findings this would further justify the approach noted above. By doing so the authors would isolate the analysis on watershed changes as they would be using identical input drivers. This would still be directly in line with the stated objectives of their study.

In fact, if we understand correctly, using 5-min data from a look-up-table (i.e. well documented storms) is similar to what you propose, but done in a more systematic way since all rain events are considered. This holds true for the Present compared to C, D, V and S simulations. Only the P (precipitation) simulation uses different 5-min evens, corresponding to Past daily rainfall.

Technical Comments:

The authors have confused the meaning of the CSA or contributing source area. This is the drainage area required it initiate the head of a first order channel and effectively defines the level of geometric complexity of the watershed with a smaller CSA (percent of drainage area or absolute area) resulting in more watershed modeling elements. The channel source area modeling elements are those that drain to the head of a first order element. The remaining upland or hillslope modeling elements (planes – they can be curvilinear as well) contribute laterally to channel modeling elements.

That's right, CSA was used for all planes. This has been corrected.

Regarding the questions by other reviewers of K2 model sensitivity the author's should review and cite Yatheendradas et al. (2008) who conducted a thorough analysis of variance. In their analysis, model prediction uncertainties are dominated by precipitation input uncertainties (another reason in suggesting the approach noted above). For K2 model parameters a multiplier on the Ksat of overland flow model elements and the Manning's roughness multiplier on overland flow model elements were the most sensitive parameters while the channel roughness multiplier was also quite important. Given this information it is odd that the authors selected the Ksat of the channels and not of the overland flow planes for calibration. Note that Ksat of channels and Ksat of hillslope elements do interact. The relatively low calibrated Ksat channel value that the authors found could easily be the result of a higher Ksat on the hillslope elements resulting in lower lateral runoff inflow into the channels.

We agree with the high sensitivity or runoff to planes parameters. Indeed, it is one conclusion of our study (Soils and Vegetation changes on planes ranking first and second).

We will explain the rationales of our approach in a more precise way in the revised manuscript: Not to calibrate the planes parameters (which are constrained by our land surface maps). Calibrate channels only,

which are less known, with the only objective to check than plane runoff produces a total flow consistent with observation, with plausible values of the channels parameters.

[Figure]

To make it clearer, we have performed a sensitivity test to planes parameters (see figure above), multiplying planes Ks by 2.5 and planes MAN by 1.75 (based on literature review by Casenave and Valentin, 1989). The absolute values of total runoff changes as expected, but the ranking of the factors is the same. This gives robustness to our findings.

(Technically, we did not use the multipliers because they modify the parameters of the planes AND the channels. Adding this feature, either adjust planes or channels, is in discussion for implementation in KINEROS2, Shea Burns, pers. com).

The calibration of channels parameter plays a minor role, ensuring only that the planes description results in simulated runoff which can match observations with plausible channel parameters. The resolution of the satellite and aerial photographs used to analyze the past and the present does not allow an identification of channel properties and their possible changes over time.

Given the author's finding of the importance in the change drainage density and channel characteristics two items are suggested:
1.      Did the author's use the default values for channel cross-sectional geometry? If so these value were derived from regressions relating X-S measurements to easily derived variables from GIS operation on watershed data as obtained at the Walnut Gulch Experimental Watershed in SE Arizona, USA (Miller et al., 2000). The Walnut Gulch relationships are likely to be a poor representation of the channel cross sectional characteristics of the Agoufou watershed. It is suggested that the authors gather some field measurements from the Agoufou watershed. At least from several stream orders so they might be scaled across all the channels in the study watershed.

We only changed channels width (10 ans 11 m) to fit observations and checked that channel depth was correct (.4 to .7m).

2.      Instead of altering the aspect ratio of the overland flow (plane) hillslope elements a watershed discretization can be derived from for each (past and present) channel network (contact Shea Burns for details).

It was probably not clear in the first manuscript, but changing the planes aspect ratio reproduces exactly the elongation of the channel network (since lateral plane's width corresponds to channel length). Changing CSA was considered also but there was a risk of changing plane properties in an uncontrolled way (changing

CSA with DEM doesn't seem straightforward in K2). That would have complicated the interpretion changes between the past and present cases.

*References*

Al-Qurashi, A., McIntyre, N., Wheater, H., Unkrich, C.L. 2008. Application of the Kineros2 rainfall---runoff model to an arid catchment in Oman. J. of Hydrology.
355:91---105.

Goodrich, D.C., Burns, I.S., Unkrich, C.L., Semmens, D., Guertin, D.P., Hernandez, M., Yatheendradas, S., Kennedy, J.R., Levick, L. 2012. KINEROS2/AGWA: Model Use, Calibration, and Validation. Transactions of the ASABE. 55(4): 1561---1574.

Miller, S.N., Youberg, A., Guertin, D.P., Goodrich, D.C. 2000. Channel morphology investigations using geographic information systems and field research. Proc. Land Stewardship in the 21st Century: The Contributions of Watershed Manage., USDA Forest Service RMRS---P---13, pp. 415---419.

Sidman, G., Guertin, D.P., Goodrich, D.C., Unkrich, C.L., Burns, I.S. 2015. Risk--- assessment of post---wildfire hydrological response in semi---arid basins: The effects of varying rainfall representations in the KINEROS2/AGWA model. International Journal of Wildland Fire. 1---11.

Yatheendradas, S., Wagener, T., Gupta, H., Unkrich, C.L., Goodrich, D.C., Schaffner, M., Stewart, A. 2008. Understanding uncertainty in distributed flash flood forecasting for semiarid regions. Water Resources Research, Vol. 44, W05S19.

---

## Author Comment (AC6) · 17 Feb 2017

Dear Editor,

We have now provided detailed answers to all comments.

Sincerely yours

Laetitia Gal

---

## Referee Report (RR1)

Referee comment on "The paradoxical evolution of runoff in pastoral Sahel: Analysis of the hydrological changes over the Agoufou watershed (Mali) using the KINEROS-2 model."

The manuscript clearly improved in clarity and quality after the revision and now most of the objections for its publication do not apply any more. Nevertheless, there are some technical and scientific aspects that should be modified before it being acceptable for publication in HESS.

The main question was already sated in the first revision and the corresponding changes introduced are not still satisfactory to this reviewer. The sentence *shallow soil being eroded and replaced by impervious soils* in the abstract and the corresponding one *with shallow soils being eroded and being replaced by impervious soils* in the conclusion sound very catastrophic, so the reader expects finding the description of erosion-deposition features such as new gullies and recently formed alluvial fans or sediment filled channels (phenomena not easily understood in such a short time and under deep droughts). But these phenomena are not described in the paper; if well understood, the only description of similar facts appears in Table 2, P4: *degradation of the tiger bush results in eroded and crusted soils which are largely impervious and produce important surface runoff*. Consequently, it seems that large parts of a landscape unit identified as "P3: Hard pan surface with tiger bush " in the old surveys were identified as "degraded tiger bush" in the new ones, but the main difference found seems to be the degradation of the vegetation cover but not the erosion of soils. Similar facts seem to be applied to the changes between P1 and P1v (herbaceous vegetation layer indicating the occurrence of shallow sandy soils <30 cm).
In other words: the authors seem to have over enhanced the role of erosion processes in the abstract and conclusions in comparison with the observed facts. Soil erosion cannot be frivolously inferred or claimed if there are no clear observations.

Other comments:

Flooded areas and floodplains are sometimes used a synonymous, but it is important to make clear which are the changes: floodplains are geomorphic units that do not change easily in centuries whereas flooded areas can vary for every event.

Page 1, line 16: "...quantify and rank different processes..." because not all the processes are analysed

Page 2, line 28: "...reduction of soil hydraulic conductivity..."

Page 4, line 10: "...which favour the frequent generation of runoff..."

Page 9, line 8: "... from soil textures throughout the world..."

Page 9, line 10: 0.1% or 0.001 $m^3 m^{-3}$?

Page 14, line 4: "... summarized in Fig. 8 and Table 8..."

Page 14, line 15: "... sand dunes interrupting the water flow..."

Page 14, lines 18-21: It is unclear which is the mechanism how the expansion of the drainage network in the northern area causes an increase of flow at the outlet. Is it because in the lack of this drainage network runoff arrived along planes where infiltration was higher than in the channels?. Please, explain the mechanisms behind this change.

Page 18, lines 24-26: (*Our study implies that enhanced and concentrated runoff and/or increase surface runoff results in an increase in both the number and the length of channel*s) this is not demonstrated in the study. The study shows the role of changing drainage density in the total simulated runoff, but not which is the cause of channel expansion. I suggest (as done in my previous review) to take into account the classical literature about channel entrenchment in semiarid areas.

In several expressions throughout the paper there is an improper or doubtful use of plurals, for instance "geometrics parameters" (pag 5, lin 33); "fields measurements" (pag 4, lin 29); channels parameters (pag 11, line20)...

There are many typing errors with lacking spaces throughout the paper

---

## Author Response (AR3)

The manuscript was improved and it is more clear now. I am also quite satisfied with the author response to my comments. There is however one point which I do not think was addressed adequately and this is the assumption of rain cells covering the entire catchment. I have some doubts about this. The authors provide comparison of rainfall distribution between different locations, but this is not relevant. Two locations can have a similar climatology in terms of rain intensity distribution, but still at a given moment different rain intensity are measured. To support the assumption that rain cells cover the entire catchment the authors should present the scatter of observed rain intensity at the same time from two gauges (for example the one near the outlet and the one within the catchment).
Thank you.

Thank you. We agree with the reviewer comment about the heterogeneity of precipitation. Indeed even if we have shown that climatology and pdf distribution are quite similar for different locations within the watershed, instantaneous precipitation intensity can be different.

The analysis of top cloud temperature, used as a proxy of the rainfall rate, revealed a time lag of 15 minutes between the eastern and the western part of the watershed, which is consistent with cells known to propagate westward at about 30-40 km/h during the African monsoon.

Given that planes are quite small and that we can reasonably assume rain intensity to be homogenous on a given plane, such a time lag would mostly result in a delayed arrival of water into the river network, which would reduce the water depth into channels at a given time, and therefore water flow.

This phenomenon is however compensated for because channels characteristics (Man and KS) are calibrated. The effect of possible time lags in rain intensity therefore probably translates into an overestimation of channel conductivity by calibration. This would not change our results, since we have shown in the paper that the ranking of factors responsible for the increase of surface runoff since 1950 does not depend on channel parameters as shown be our sensitivity analysis

The authors also refer to Gal's thesis, but I could not read it since it is in French. Looking at the MSG images presented in the thesis it is hard to tell what is the "cell" size since cloud top temperatures are shown and how they are translated to rain rates is not clear from the images. There are rainfall products at high resolution (~4 km, 30-min) that could be used to demonstrate this point in a better way. Also, the authors refer t a possible variability of rainfall but the write in page 8 line 18: "This variability is significantly smoothed out at the annual time step but still persists and constitutes an uncertainty". But runoff response is not at the annual scale but on the event scale, and surely it is sensitive to rainfall variability.

In my opinion, it is better not to claim for a uniform coverage if the authors are not sure about it. Other studies have shown rain cells are often in the size of few tens of square km, so even if we take the relevant third of the catchment, the location of the rain cell totally change the rain intensity over this area.

I think the way to go is to introduce somehow (in a random way or assuming a certain structure) the intra-catchment rainfall variability into the simulations and show their conclusions are not very sensitive to this variability. I assume the computed runoff for each event will be quite different if some variability will be introduced comparing to the uniform rain assumption, but the final mean volume for the two periods will still have a large difference and the main causes of this difference will not change.

We have been investigating this precipitation heterogeneity issue in several ways, including satellite imagery and Lagrangian kriging. First, we agree on the difficulty to translate top cloud temperature into rainfall rates, especially when going down to small spatial and temporal resolution. Indeed, according to the recent analysis carried out by Guilloteau et al. (2016), satellite products with a fine spatial and temporal scale (finer than 40km and 2h) are poorly correlated to radar measurements and do not represent the physical variability of rainfall pattern in the Sahel accurately.

To follow the reviewer suggestion to look at rainfall at higher resolution, we have used rainfall products (5km, 30min) derived for from in-situ measurements using a Lagrangian kriging method (Vischel et al. 2011). This method

accounts for the propagation of rainy systems derived from raingauge data. The figure below shows cumulative rainfall in 2008 for 6 adjacent pixels of the Agoufou watershed (covering the contributing area) obtained in this way. We can see that, at the scale of 30 minutes, there is not any time lag during the significant rainfall events, which confirms the analysis carried out using top cloud temperature, and that differences in the total rainfall amounts are of about 10%. This dataset however has a 30 min time step which is not what we want for our study, and it was produced for 2008 only, the year when raingauge number was at its maximum. Adding a sensitivity study with this dataset would not be consistent with the rest of the study (time step). We therefore preferred to make it clear that heterogeneity probably exists, mainly affects the channels, and is compensated for during channels parameters setup.

We hope that this has been better explained in the revised manuscript.

[Figure]

Figure 1: Cumulative rainfall for 2008 in the northern part of the Agoufou watershed (6 pixels located on the map at the right hand side).

The revised text now reads:

*Section 3.2.1: "In addition, the cloud top temperature, derived by MSG remote sensing data, was also analyzed, confirming that the rainfall cells in this region are generally larger than the watershed area (see Gal, 2016 for more details). However, a small temporal time lag (<30min) can occur between the eastern and western parts of the watershed since the cells are propagating westward. Planes are small enough to be considered as homogeneous in terms of precipitation but such a time lag impacts the water level and infiltration in channels. However, this effect is compensated for during the simulation setup and the derivation of channels parameters (see below). Last, the spatial variability in Sahelian rain fields at the event scale, as observed by Le Barbe et al. (2002), tends to smoothed out at the annual time step even if it still persists and constitutes a source of uncertainty."*

*Section 5.2: "As highlighted in section 3.2.1, a possible time lag in precipitation falling within the watershed would have an impact on the instantaneous amount of water ending up into channels. In our simulation setup, channels are calibrated, so a possible time lag in precipitation would have as consequence an overestimation of channel conductivity during calibration. However this would not sensibly impact our results, which have been shown to be robust to changes in channel characteristics. "*

Minor comments: add "potential" before evaporation in page 3 line 37

Done, thank you

References

C. Guilloteau, R. Roca and M. Gosset (2016): "A Multiscale Evaluation of the Detection Capabilities of High-Resolution Satellite Precipitation Products in West Africa", JHM, 17, 2041-2059, DOI: 10.1175/JHM-D-15-0148.1

T. Vischel, G. Quantin, T. Lebel, J. Viarre, M. Gosset, F. Cazenave and G. Panthou (2011) "Generation of High-Resolution Rain Fields in West Africa: Evaluation of Dynamic Interpolation Methods", JHM, 12, 1465-1482, DOI: 10.1175/JHM-D-10-05015.1

**Referee comment on "The paradoxical evolution of runoff in pastoral Sahel: Analysis of the hydrological changes over the Agoufou watershed (Mali) using the KINEROS-2 model."**

The manuscript clearly improved in clarity and quality after the revision and now most of the objections for its publication do not apply any more. Nevertheless, there are some technical and scientific aspects that should be modified before it being acceptable for publication in HESS.

The main question was already sated in the first revision and the corresponding changes introduced are not still satisfactory to this reviewer. The sentence shallow soil being eroded and replaced by impervious soils in the abstract and the corresponding one with shallow soils being eroded and being replaced by impervious soils in the conclusion sound very catastrophic, so the reader expects finding the description of erosion-deposition features such as new gullies and recently formed alluvial fans or sediment filled channels (phenomena not easily understood in such a short time and under deep droughts). But these phenomena are not described in the paper; if well understood, the only description of similar facts appears in Table 2, P4: degradation of the tiger bush results in eroded and crusted soils which are largely impervious and produce important surface runoff. Consequently, it seems that large parts of a landscape unit identified as "P3: Hard pan surface with tiger bush " in the old surveys were identified as "degraded tiger bush" in the new ones, but the main difference found seems to be the degradation of the vegetation cover but not the erosion of soils.

Similar facts seem to be applied to the changes between P1 and P1v (herbaceous vegetation layer indicating the occurrence of shallow sandy soils <30 cm).

In other words: the authors seem to have over enhanced the role of erosion processes in the abstract and conclusions in comparison with the observed facts. Soil erosion cannot be frivolously inferred or claimed if there are no clear observations.

Many thanks to the editor for his comments and suggestions.

We agree with the fact that we do not give direct evidence of the erosion process and we do not want to make it appear dramatic. We therefore changed our sentences in order to smooth out our conclusion and suggest that the erosion phenomenon is most likely responsible for the landscape changes observed according to observations and literature.

There are however signs that suggest erosion, in addition to gully development: Two examples are reported in the Figure below that shows that the impermeable zones (O; the dark brown on the right hand side images) were more restricted in 1956 (images on the left) with shallow sandy soil with vegetation (SSS; gray area with black dots) as well as some silt layer (Si; bright) being present in 1956 and replaced by rocky outcrops (O). These areas have been shaped by important erosion, either water or wind erosion, following the degradation of the vegetation.

[Figure]

Figure 1: Landscape evolution between 1956 (left) and 2011 (right) for the Agoufou watershed. With SSS: Shallow Sandy Soil; Sa: Sandy soil; Si: Silty soil and O: Outcrop.

This erosion process has been put forward for the Tin Adjar watershed (100 km north of our study site) by Kergoat et al. (2015), with strong erosion of shallow sandy soils observed in the upper part of the watershed and a marked alluvial deposit close to the outlet (sorry it is in French, a translation of the book is in process).

In the case of Agoufou, increased runoff involves regression of vegetation cover, runoff concentration and erosion that created rocky outcrops or pediments that are not suitable for plant growth and probably interplay of these factors (not demonstrated though).

Erosion is also in line with the processes described by Descroix et al. (2007) in other locations in the Sahel over a similar period of time and also by Poesen et al. (2003) and Valentin (2005) concerning the drainage network development.

[Figure]

[Figure]

**Figure 2: Landscape evolution between 1954 (left) and 2007(right) for the Tin Adjar watershed (Kergoat et al., 2015).**

Other comments:

Flooded areas and floodplains are sometimes used a synonymous, but it is important to make clear which are the changes: floodplains are geomorphic units that do not change easily in centuries whereas flooded areas can vary for every event.

Floodplain corresponds to the alluvial plains whereas flooded zone corresponds to the landscape unit which includes the floodplain and the open water. It is a generic term use for a landscape unit.

Page 1, line 16: "...quantify and rank different processes..." because not all the processes are analyzed
Page 2, line 28: "...reduction of soil hydraulic conductivity..."
Page 4, line 10: "...which favour the frequent generation of runoff..."
Page 9, line 8: "... from soil textures throughout the world..."
Page 9, line 10: 0.1% or 0.001 $m^3m^{-3}$?
Page 14, line 4: "... summarized in Fig. 8 and Table 8..."
Page 14, line 15: "... sand dunes interrupting the water flow..."

All these remarks have been taken into account. Thank you.

Page 14, lines 18-21: It is unclear which is the mechanism how the expansion of the drainage network in the northern area causes an increase of flow at the outlet. Is it because in the lack of this drainage network runoff arrived along planes where infiltration was higher than in the channels?. Please, explain the mechanisms behind this change.

Yes exactly, the infiltration was higher on average over the planes than in the channels. Anyway, when we changed the channel parameters according to the literature (testing more infiltrating channels), the increase of drainage density also induced an increase of surface runoff but to a lesser extent.
This has been added on the revised manuscript line 38 page 18.

Page 18, lines 24-26: (*Our study implies that enhanced and concentrated runoff and/or increase surface runoff results in an increase in both the number and the length of channels*) this is not demonstrated in the study. The study shows the role of changing drainage density in the total simulated runoff, but not which is the cause of channel

expansion. I suggest (as done in my previous review) to take into account the classical literature about channel entrenchment in semiarid areas.

Thank you for this suggestion, we have added in the revised manuscript more information about gullies entrenchment causes based on the literature (Poesen et al., 2003; Valentin et al., 2005; Marzolff et al., 2011).

In several expressions throughout the paper there is an improper or doubtful use of plurals, for instance "geometrics parameters" (pag 5, lin 33); "fields measurements" (pag 4, lin 29); channels parameters (pag 11, line20)...
There are many typing errors with lacking spaces throughout the paper.

Thank you, this has been checked throughout the manuscript

[revised manuscript text omitted]

Ks (mm hr$^{-1}$)

**Table 7: Percent bias and RMSE on annual discharge over 2011-2015 for 25 sets of channels Ks and Man parameters. The minimum value for the percent bias and RMSE is indicated by the square box (red and black respectively).**

|  | Ref* | | | S-PL** | | | S-CH*** | | |
|------|------|------|------|------|------|------|------|------|------|
|  | mQ( (10⁶ m³) | mQ (mm) | Ex (%) | mQ( (10⁶ m³) | mQ (mm) | Ex (%) | mQ( (10⁶ m³) | mQ (mm) | Ex (%) |
| **PRES** | 3.3 | 13.4 | 0.0 | 0.8 | 3.3 | 0.0 | 3.0 | 12.3 | 0.0 |
| **C** | 3.2 | 13.3 | 2.7 | 0.7 | 2.8 | 14.9 | 3.0 | 12.2 | 1.0 |
| **D** | 2.7 | 10.9 | 22.1 | 0.7 | 2.7 | 19.6 | 2.8 | 11.4 | 8.9 |
| **V** | 2.1 | 8.7 | 41.9 | 0.3 | 1.3 | 63.7 | 1.9 | 7.9 | 43.3 |
| **S** | 0.7 | 2.6 | 94.0 | 0.1 | 0.2 | 97.2 | 0.5 | 2.2 | 98.1 |
| **P** | 4.1 | 16.7 | -28.6 | 1.0 | 3.9 | -21.5 | 3.7 | 15.3 | -29.3 |
| **PAST** | 0.5 | 2.1 | 100.0 | 0.0 | 0.1 | 100.0 | 0.5 | 2.0 | 120.0 |

*: Initial simulation with default parameters presented in **Fig.8**
**: Simulation with Ks (×2.5) and MAN (×1.75) modified for all planes
***: Simulation with Ks (40 mm hr$^{-1}$) and MAN (0.03 s m$^{-1/3}$) modified for all channels

**Table 8: Mean annual discharge (mQ) and *Ex* values obtained for the initial simulation with default parameters presented in Fig.8 (Ref*) and the sensitivity test for plane (S-PL**) and channel (S-CH***) parameters.**

[Figure]

**Fig. 1: The Agoufou watershed (245 km²) located in western Sahel (Mali) with drainage network and available rain gauges (map source: http://www.diva-gis.org/gdata).**

[Figure]

5     **Fig. 2: Planes for the Agoufou watershed with the DEM-derived drainage network.**

[Figure]

**Fig. 3: Land cover maps of the Agoufou watershed for (a) 1956, (b) 2011 and (c) gives the surface difference (in km²) for each landscape unit, between 2011 and 1956.**

[Figure]

**Fig. 4: Identification of the four sub-watersheds (grey shades) which display the largest changes between the drainage network in 1956 (red line) and in 2011 (black line).**

[Figure]

**Fig. 5: Cumulative discharge (CQ) for years with observation data over 2000−2015. For each year, black dots are for the observations and the gray-shaded envelop represents the maximum and minimum of the ten members of the ensemble simulations. (a) is for the validation period and (b) is for the channel set-up period.**

[Figure]

**Fig. 6: Evolution of annual discharge (AQ in m³) between 1960 and 2015: simulations with standard deviation of the ten members (black dots with error bar) and observations (red dots) together with annual precipitation (AP in mm, blue bars).**

[Figure]

**Fig. 7: Relation between precipitation and discharge for all events in the 15 years period in the past (red points) and the present (black points).**

[Figure]

**Fig. 8: Mean discharge (mQ) for present and past reference cases and for the attribution simulations described in Table 5, with either independent or combined factors. For the references cases, observation data are added (red points). Errors bars indicate the standard deviation of the ten member and the fifteen years of simulation.**

[Figure]

**Fig. 9: Spatial patterns of the (a) saturated hydraulic conductivity (Ks), (b) Manning's roughness coefficient (Man) and (c) Discharge (Q) between the past and the period for the monsoon season (JAS).**